# Limitless stability for Graph Convolutional Networks

## Abstract

This work establishes rigorous, novel and widely applicable stability guarantees and transferability bounds for graph convolutional networks – without reference to any underlying limit object or statistical distribution. Crucially, utilized graph-shift operators (GSOs) are not necessarily assumed to be normal, allowing for the treatment of networks on both directed- and for the first time also undirected graphs. Stability to node-level perturbations is related to an 'adequate (spectral) covering' property of the filters in each layer. Stability to edge-level perturbations is related to Lipschitz constants and newly introduced semi-norms of filters. Results on stability to topological perturbations are obtained through recently developed mathematical-physics based tools. As an important and novel example, it is showcased that graph convolutional networks are stable under graph-coarse-graining procedures (replacing strongly-connected sub-graphs by single nodes) precisely if the GSO is the graph Laplacian and filters are regular at infinity. These new theoretical results are supported by corresponding numerical investigations.

## 1 Introduction

Graph Convolutional Networks (GCNs) (Kipf & Welling, 2017; Hammond et al., 2011; Defferrard et al., 2016) generalize Euclidean convolutional networks to the graph setting by replacing convolutional filters by functional calculus filters; i.e. scalar functions applied to a suitably chosen graph-shift-oprator capturing the geometry of the underlying graph. A key concept in trying to understand the underlying reasons for the superior numerical performance of such networks on graph learning tasks (as well as a guiding principle for the design of new architectures) is the concept of **stability**. In the Euclidean setting, investigating stability essentially amounts to exploring the variation of the output of a network under non-trivial changes of its input (Mallat, 2012; Wiatowski & Bölcskei, 2018). In the graph-setting, additional complications are introduced: Not only input signals, but now also the graph shift operators facilitating the convolutions on the graphs may vary. Even worse, there might also occur changes in the topology or vertex sets of the investigated graphs – e.g. when two dissimilar graphs describe the same underlying phenomenon – under which graph convolutional networks should also remain stable. This last stability property is often also referred to as **transferability** (Levie et al., 2019a). Previous works investigated stability under changes in graph-shift operators for specific filters (Levie et al., 2019b; Gama et al., 2020) or the effect of graph-rewiring when choosing a specific graph shift operator (Kenlay et al., 2021). Stability to topological perturbations has been established for (large) graphs discretising the same underlying topological space (Levie et al., 2019a), the same graphon (Ruiz et al., 2020; Maskey et al., 2021) or for graphs drawn from the same statistical distribution (Keriven et al., 2020; Gao et al., 2021).

Common among all these previous works are two themes limiting practical applicability: First and foremost, the class of filters to which results are applicable is often severely restricted. The same is true for the class of considered graph shift operators; with non-normal operators (describing directed graphs) either explicitly or implicitly excluded. Furthermore – when investigating transferability properties – results are almost exclusively available under the assumption that graphs are large and either discretize the same underlying 'continuous' limit object sufficiently well, or are drawn from the same statistical distributions. While these are of course relevant regimes, they do not allow to draw conclusions beyond such asymptotic settings, and are for example unable to deal with certain spatial graphs, inapplicable to small-to-medium sized social networks and incapable of capturing the inherent multi-scale nature of molecular graphs (as further discussed below). Finally, hardly any

work has been done on relating the stability to input-signal perturbations to network properties such as the interplay of utilized filters or employed non-linearities. The main focus of this work is to provide alleviation in this situation and develop a 'general theory of stability' for GCNs – agnostic to the types of utilized filters, graph shift operators and non-linearities; with practically relevant transferability guarantees not contingent on potentially underlying limit objects. To this end, Section 2 recapitulates the fundamentals of GCNs in a language adapted to our endeavour. Sections 3 and 4 discuss stability to node- and edge-level perturbations. Section 5 discusses stability to structural perturbations. Section 6 discusses feature aggregation and Section 7 provides numerical evidence.

## 2 GCNs via Complex Analysis and Operator Theory

Throughout this work, we will use the label $G$ to denote both a graph and its associated vertex set. Taking a signal processing approach, we consider signals on graphs as opposed to graph embeddings:

**Node-Signals:** Node-signals on a graph are then functions from $G$ to the complex numbers; i.e. elements of $\mathbb{C}^{|G|}$ (with $|G|$ the cardinality of $G$). We allow nodes $i \in G$ in a given graph to have weights $\mu_i$ not necessarily equal to one and equip the space $\mathbb{C}^{|G|}$ with an inner product according to $\langle f, g \rangle = \sum_{i \in G} \overline{f}(i)g(i)\mu_i$ to account for this. We denote the hence created Hilbert space by $\ell^2(G)$.

**Characteristic Operators:** Fixing an indexing of the vertices, information about connectivity within the graph is encapsulated into the set of edge weights, collected into the adjacency matrix $W$ and (diagonal) degree matrix $D$. Together with the weight matrix $M := \text{diag}\left(\{\mu_i\}_{i=1}^{|G|}\right)$, various standard geometry capturing characteristic operators – such as weighted adjacency matrix $M^{-1}W$, graph Laplacian $\Delta := M^{-1}(D-W)$ and normalized graph Laplacian $\mathscr{L} := M^{-1}D^{-\frac{1}{2}}(D-W)D^{-\frac{1}{2}}$ can then be constructed. For undirected graphs, all of these operators are self-adjoint. On directed graphs, they need not even be normal ($T^*T = TT^*$). We shall remain agnostic to the choice of characteristic operator; differentiating only between normal and general operators in our results.

**Functional Calculus Filters:** A crucial component of GCNs are functional calculus filters, which arise from applying a function $g$ to an underlying characteristic operator $T$; creating a new operator $g(T)$. Various methods of implementations exist, all of which agree if multiple are applicable:

Generic Filters: If (and only if) $T$ is normal, we may apply generic complex valued functions $g$ to $T$: Writing normalized eigenvalue-eigenvector pairs of $T$ as $(\lambda_i, \phi_i)_{i=1}^{|G|}$ one defines $g(T)\psi = \sum_{i=1}^{|G|} g(\lambda_i)\langle \phi_i, \psi \rangle_{\ell^2(G)} \phi_i$ for any $\psi \in \ell^2(G)$. One has $\|g(T)\|_{op} = \sup_{\lambda \in \sigma(T)} |g(\lambda)|$, with $\sigma(T)$ denoting the spectrum of $T$. If $g$ is bounded, one may obtain the $T$-independent bound $\|g(T)\|_{op} \leqslant \|g\|_\infty$. Keeping in mind that $g$ being defined on all of $\sigma(T)$ (as opposed to all of $\mathbb{C}$) is clearly sufficient, we define a space of filters which will harmonize well with our concept of transferability discussed in Section 5. The introduced semi-norm will quantify the stability to perturbations in coming sections.

**Definition 2.1.** Fix $\omega \in \mathbb{C}$ and $C > 0$. Define the space $\mathscr{F}_{\omega,C}^{cont}$ of continuous filters on $\mathbb{C}\backslash\{\omega, \overline{\omega}\}$, to be the space of multilinear power-series' $g(z) = \sum_{\mu,\nu=0}^{\infty} a_{\mu\nu} (\omega - z)^{-\mu} (\overline{\omega} - \overline{z})^{-\mu}$ for which the semi-norm $\|g\|_{\mathscr{F}_{\omega,C}^{cont}} := \sum_{\mu,\nu>0}^{\infty} |\mu + \nu| C^{\mu+\nu-1} |a_{\mu\nu}|$ is finite.

Denoting by $B_\epsilon(\omega) \subseteq \mathbb{C}$ the open ball of radius $\epsilon$ around $\omega$, one can show that for arbitrary $\delta > 0$ and every continuous function $g$ defined on $\mathbb{C}\backslash(B_\epsilon(\omega) \cup B_\epsilon(\overline{\omega}))$ which is regular at infinity – i.e. satisfies $\lim_{r\to+\infty} g(rz) = c \in \mathbb{C}$ independent of which $z \neq 0$ is chosen – there is a function $f \in \mathscr{F}_{\omega,C}^{cont}$ so that $|f(z) - g(z)| \leqslant \delta$ for all $z \in \mathbb{C}\backslash(B_\epsilon(\omega) \cup B_\epsilon(\overline{\omega}))$. In other words, functions in $\mathscr{F}_{\omega,C}^{cont}$ can approximate a wide class of filters to arbitrary precision. More details are presented in Appendix B.

Entire Filters: If $T$ is not necessarily normal, one might still consistently apply entire (i.e. everywhere complex differentiable) functions to $T$. Detail details on the mathematical background are given in Appendix C. Here we simply note that such a function $g$ is representable as an (everywhere convergent) power series $g(z) := \sum_{k=0}^{\infty} a_k^g z^k$ so that we may simply set $g(T) = \sum_{k=0}^{\infty} a_k^g \cdot T^k$. For the norm of the derived operator one easily finds $\|g(T)\|_{op} \leqslant \sum_{k=0}^{\infty} |a_k^g|\|T\|_{op}^k$ using the triangle

inequality. While entire filters have the advantage that they are easily and efficiently implementable – making use only of matrix multiplication and addition – they suffer from the fact that it is impossible to give a $\|T\|_{op}$-independent bound for $\|g(T)\|_{op}$ as for continuous filters. This behaviour can be traced back to the fact that no non-constant bounded entire function exists (Bak & Newman, 2017).

HOLOMORPHIC FILTERS: To define functional calculus filters that are both applicable to non-normal $T$ and boundable somewhat more controlably in terms of $T$, one may relax the condition that $g$ be entire to demanding that $g$ be complex differentiable (i.e. **holomorphic**) only on an open subset $U \subseteq \mathbb{C}$ of the complex plane. Here we assume that $U$ extends to infinity in each direction (i.e. is the complement of a closed and bounded subset of $\mathbb{C}$). For any $g$ holomorphic on $U$ and regular at infinity we set (with $(zId - T)^{-1}$ the so called reolvent of $T$ at $z$)

$$g(T) := g(\infty) \cdot Id + \frac{1}{2\pi i} \oint_{\partial D} g(z) \cdot (zId - T)^{-1} dz, \qquad (1)$$

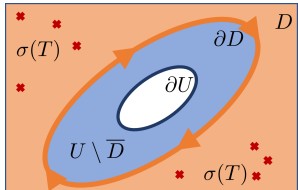

for any $T$ whose spectrum $\sigma(T)$ is completely contained in $U$. Here we have used the notation $g(\infty) = \lim_{r \to +\infty} g(rz)$ and taken $D$ to an open set with nicely behaved boundary $\partial D$ (more precisely a Cauchy domain; c.f. Appendix C). We assume that $D$ completely contains $\sigma(T)$ and that its closure $\overline{D}$ is completely contained in $U$. The orientation

Figure 1: Set-Visualisations

of the boundary $\partial D$ is the usual positive orientation on $D$ (such that $D$ 'is on the left' of $\partial D$; cf. Fig. 1). Using elementary facts from complex analysis it can be shown that the resulting operator $g(T)$ in (1) is independent of the specific choice of $D$ (Gindler, 1966). While we will present results below in terms of this general definition – remaining agnostic to numerical implementation methods for the most part – it is instructive to consider a specific exemplary setting with definite and simple numerical implementation of such filters: To this end, chose an arbitrary point $\omega \in \mathbb{C}$ and set $U = \mathbb{C}\backslash\{\omega\}$ in the definitions above. Any function $g$ that is holomorphic on $U$ and regular at $\infty$ may then be represented by its Laurent series, which is of the form $g(z) = \sum_{k=0}^{\infty} b_k^g (z - \omega)^{-k}$ (Bak & Newman, 2017). For any $T$ with $\sigma(T) \subseteq U$ (i.e. $\omega \notin \sigma(T)$) evaluating the integral in (1) yields (c.f. Appendix C):

$$g(T) = \sum_{k=0}^{\infty} b_k^g \cdot (T - \omega Id)^{-k} \qquad (2)$$

Such filters have already been employed successfully, e.g. in the guise of Cayley filters (Levie et al., 2019c), which are polynomials in $\frac{z+i}{z-i} = 1 + \frac{2i}{z-i}$. We collect them into a designated filter space:

**Definition 2.2.** For a function $g(z) = \sum_{k=0}^{\infty} b_k^g (z - \omega)^{-k}$ on $U := \mathbb{C}\backslash\{\omega\}$ define the semi-norm $\|g\|_{\mathscr{F}_{\omega,C}^{hol}} := \sum_{k=1}^{\infty} |b_k^g| k C^{k-1}$ for $C > 0$. Denote the set of such $g$ for which $\|g\|_{\mathscr{F}_{\omega,C}^{hol}} < \infty$ by $\mathscr{F}_{\omega,C}^{hol}$.

In order to derive $\|T\|_{op}$-independent bounds for $\|g(T)\|_{op}$, we will need to norm-bound the resolvents appearing in (1) and (2). If $T$ is normal, we simply have $\|(zId - T)^{-1}\|_{op} = 1/\text{dist}(z, \sigma(T))$. In the general setting, following Post (2012), we call any positive function $\gamma_T$ satisfying $\|(zId - T)^{-1}\|_{op} \leqslant \gamma_T(z)$ on $\mathbb{C}\backslash\sigma(T)$ a **resolvent profile** of $T$. Various methods (e.g. Szehr (2014); MichaelGil (2012)) to find resolvent profiles. Most notably Bandtlow (2004b) gives a resolvent profile solely in terms of $1/\text{dist}(z, \sigma(T))$ and the departure from normality of $T$. We then find the following result:

**Lemma 2.3.** For holomorphic $g$ and generic $T$ we have $\|g(T)\|_{op} \leqslant |g(\infty)| + \frac{1}{2\pi} \oint_{\partial D} |g(z)| \gamma_T(z) d|z|$. Furthermore we have for any $T$ with $\gamma_T(\omega) \leqslant C$, that $\|g(T)\|_{op} \leqslant \|g\|_{\mathscr{F}_{\omega,C}^{hol}}$ as long as $g \in \mathscr{F}_{\omega,C}^{hol}$.

Lemma 2.3 (proved in Appendix D) finally bounds $\|g(T)\|_{op}$ independently of $T$, as long as appearing resolvents are suitably bounded; which – importantly – does not force $\|T\|_{op}$ to be bounded.

**Non-Linearities & Connecting Operators:** To each layer of our GCN, we associate a (possibly) non-linear and $L_n$-Lipschitz-continuous function $\rho_n : \mathbb{C} \to \mathbb{C}$ satisfying $\rho_n(0) = 0$ which acts point-wise on signals in $\ell^2(G_n)$. This definition allows to choose $\rho_n = |\cdot|, ReLu, Id$ or any sigmoid function shifted to preserve zero. To account for recently proposed networks where input- and 'processing' graphs are decoupled (Alon & Yahav, 2021; Topping et al., 2021), and graph pooling layers (Lee et al., 2019), we also allow signal representations in the hidden network layers $n$ to live in varying graph signal spaces $\ell^2(G_n)$. Connecting operators are then (not necessarily linear) operators

$P_n : \ell^2(G_{n-1}) \to \ell^2(G_n)$ connecting the signal utilized of subsequent layers. We assume them to be $R_n$-Lipschitz-continuous ($\|P_n(f) - P_n(g)\|_{\ell^2(G_{n-1})} \leqslant R_n \|f - g\|_{\ell^2(G_n)}$) and triviality preserving ($P_n(0) = 0$). For our original node-signal space we also write $\ell^2(G) \equiv \ell^2(G_0)$.

**Graph Convolutional Networks:** A GCN with $N$ layers is then constructed as follows: Let us denote the width of the network at layer $n$ by $K_n$. The collection of hidden signals in this layer can then be thought of a single element of

$$\mathscr{L}_n := \bigoplus_{i \in K_n} \ell^2(G_n). \qquad (3)$$

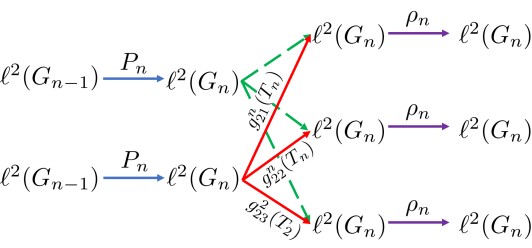

Figure 2: Update Rule for a GCN

Further let us write the collection of functional calculus filters utilized to generate the representation of this layer by $\{g_{ij}^n(\cdot) : 1 \leqslant j \leqslant K_{n-1}; 1 \leqslant i \leqslant K_n\}$. Further denoting the characteristic operator of this layer by $T_n$, the update rule (c.f. also Fig. 2) from the representation in $\mathscr{L}_{n-1}$ to $\mathscr{L}_n$ is then defined on each constituent in the direct sum $\mathscr{L}_n$ as

$$f_i^{n+1} = \rho_{n+1} \left( \sum_{j=1}^{K_n} g_{ij}^{n+1}(T_{n+1}) P_{n+1}(f_j^n) \right), \quad \forall 1 \leqslant i \leqslant K_n.$$

We also denote the initial signal space by $\mathscr{L}_{\text{in}} := \mathscr{L}_0$ and the final one by $\mathscr{L}_{\text{out}} := \mathscr{L}_N$. The hence constructed map from the initial to the final space is denoted by $\Phi : \mathscr{L}_{\text{in}} \to \mathscr{L}_{\text{out}}$.

## 3 STABILITY TO INPUT SIGNAL PERTURBATIONS

In order to produce meaningful signal representations, a small input signal change should produce only a small variation in the output of our GCN. This property is quantified by the Lipschitz constant of the map $\Phi$ associated to the network, which is estimated by our first result below.

**Theorem 3.1.** With the notation of Section 2 let $\Phi_N : \mathscr{L}_{\text{in}} \to \mathscr{L}_{\text{out}}$ be the map associated to an $N$-layer GCN. We have with $B_n := \sqrt{\sup_{\lambda \in \sigma(T_n)} \sum_{j \in K_{n-1}} \sum_{i \in K_n} |g_{ij}^n(\lambda)|^2}$ for all $f, h \in \mathscr{L}_{\text{in}}$ that

$$\|\Phi_N(f) - \Phi_N(h)\|_{\mathscr{L}_{\text{out}}} \leqslant \left( \prod_{n=1}^N L_n R_n B_n \right) \cdot \|f - h\|_{\mathscr{L}_{\text{in}}}$$

if $T_n$ is normal. For general $T_n$ we have for all $\{g_{ij}\}$ entire, holomorphic and in $\mathscr{F}_{\omega,C}^{hol}$ respectively:

$$B_n := \begin{cases} \sum\limits_{k=0}^{\infty} \sqrt{\sum_{j \in K_{n-1}} \sum_{i \in K_n} |(a_{ij}^{g_n})_k|^2} \cdot \|T_n\|_{op}^k \\ \sqrt{\sum_{j \in K_{n-1}} \sum_{i \in K_n} \|g_{ij}^n(\infty)\|^2} + \frac{1}{2\pi} \oint_{\partial D} \gamma_T(z) \sqrt{\sum_{j \in K_{n-1}} \sum_{i \in K_n} |g_{ij}^n(z)|^2} d|z| \\ \sqrt{\sum_{j \in K_{n-1}} \sum_{i \in K_n} \|g_{ij}^n\|_{\mathscr{F}_{\omega,C}^{hol}}^2} \end{cases}$$

Appendix E contains the corresponding proof and discusses how the derived bound are not necessarily tight for sparsely connected layers. After Lipschitz constants of connecting operators and non-linearities are fixed, the stability constant of the network is completely controlled by the $\{B_n\}$; which for normal $T_n$ in turn are controlled by the interplay of the utilized filters on the spectrum of $T_n$. This allows to combine filters with $\sup_{\lambda \in \sigma(T_n)} |g_{ij}^n(\lambda)| = \mathcal{O}(1)$ but supported on complimentary parts of the spectrum of $T_n$ while still maintaining $B_n = \mathcal{O}(1)$ instead of $\mathcal{O}(\sqrt{K_n \cdot K_{n-1}})$. In practice one might thus penalize a 'multiple covering' of the spectrum by more than one filter at a time during training in order to increase stability to input signal perturbations. If $T_n$ is not normal but filters are holomorphic, an interplay persists – with filters now evaluated on a curve and at infinity.

## 4 STABILITY TO EDGE PERTURBATIONS

Operators capturing graph-geometries might only be known approximately in real world tasks; e.g. if edge weights are only known to a certain level of precision. Hence it is important that graph

convolutional networks be insensitive to small changes in the characteristic operators $\{T_n\}$. Since we consider graphs with arbitrary vertex weights $\{\mu_g\}_{g \in G}$, we also have to consider the possibility that these weights are only known to a certain level of precision. In this case, not only do the characteristic operators $T_n$, $\widetilde{T}_n$ differ, but also the the spaces $\ell^2(G)$, $\ell^2(\widetilde{G})$ on which they act. To capture this setting mathematically, we assume in this section that there is a linear operator $J : \ell^2(G) \to \ell^2(\widetilde{G})$ facilitating contact between signal spaces (of not-necessarily the same dimension). We then measure closeness of characteristic operators in the respective spaces by considering the generalized norm-difference $\|(JT - \widetilde{T}J)\|$; with $J$ translating between the respective spaces. Before investigating the stability of entire networks we first comment on single-filter stability. For normal operators we then find the following result, proved in Appendix A building on ideas first developed in (Wihler, 2009).

**Lemma 4.1.** Denote by $\| \cdot \|_F$ the Frobenius norm and let $T$ and $\widetilde{T}$ be normal on $\ell^2(G)$ and $\ell^2(\widetilde{G})$ respectively. Let $g$ be Lipschitz continuous with Lipschitz constant $D_g$. For any linear $J : \ell^2(G) \to \ell^2(\widetilde{G})$ we have $\|g(\widetilde{T})J - Jg(T)\|_F \leqslant D_g \|\widetilde{T}J - JT\|_F$.

Unfortunately, scalar Lipschitz continuity only directly translates to operator functions if they are applied to normal operators and when using Frobenius norm (as opposed to e.g. spectral norm). For general operators we have the following somewhat weaker result, proved in Appendix F:

**Lemma 4.2.** Let $T, \widetilde{T}$ be operators on on $\ell^2(G)$ , $\ell^2(\widetilde{G})$ with $\|T\|_{op}, \|\widetilde{T}\|_{op} \leqslant C$. Let $J : \ell^2(G) \to \ell^2(\widetilde{G})$ be linear. With $K_g = \frac{1}{2\pi} \oint_{\partial D} \frac{1}{|z|} \gamma_T(z) \gamma_{\widetilde{T}}(z) |g(z)| d|z|$ for $g$ holomorphic and $K_g = \sum_{k=1}^{\infty} |a_k^g| k C^{k-1}$ for $g$ entire, we have $\|g(T)J - Jg(\widetilde{T})\|_{op} \leqslant K_g \cdot \|JT - \widetilde{T}J\|_{op}$.

Each $K_g$ itself is interpretable as a semi-norm. For GCNs we find the following (c.f. Appendix F):

**Theorem 4.3.** Let $\Phi_N$, $\widetilde{\Phi}_N$ be the maps associated to $N$-layer graph convolutional networks with the same non-linearities and filters, but based on different graph signal spaces $\ell^2(G)$, $\ell^2(\widetilde{G})$, characteristic operators $T_n, \widetilde{T}_n$ and connecting operators $P_n, \widetilde{P}_n$. Assume $B_n, \widetilde{B}_n \leqslant B$ as well as $R_n, \widetilde{R}_n \leqslant R$ and $L_n \leqslant L$ for some $B, R, L > 0$ and all $n \geqslant 0$. Assume that there are identification operators $J_n : \ell^2(G_n) \to \ell^2(\widetilde{G}_n)$ $(0 \leqslant n \leqslant N)$ commuting with non-linearities and connecting operators in the sense of $\|\widetilde{P}_n J_{n-1} f - J_n P_n f\|_{\ell^2(\widetilde{G}_n)} = 0$ and $\|\rho_n(J_n f) - J_n \rho_n(f)\|_{\ell^2(\widetilde{G}_n)} = 0$. Depending on whether normal or arbitrary characteristic operators are used, define $D_n^2 := \sum_{j \in K_{n-1}} \sum_{i \in K_n} D_{g_{ij}^n}^2$ or $D_n^2 := \sum_{j \in K_{n-1}} \sum_{i \in K_n} K_{g_{ij}^n}^2$. Choose $D$ such that $D_n \leqslant D$ for all $n$. Finally assume that $\|J_n T_n - \widetilde{T}_n J_n\|_* \leqslant \delta$ and with $* = F$ if both operators are normal and $* = op$ otherwise. Then we have for all $f \in \mathcal{L}_{\text{in}}$ and with $\mathscr{J}_n$ the operator that the $K_n$ copies of $J_n$ induce through concatenation that $\|\widetilde{\Phi}(\mathscr{J}_0 f) - \mathscr{J}_N \Phi(f)\|_{\widetilde{\mathcal{L}}_{\text{out}}} \leqslant N \cdot DRL \cdot (BRL)^{N-1} \cdot \|f\|_{\mathcal{L}_{\text{in}}} \cdot \delta$.

The result persists with slightly altered constants, if identification operators only *almost* commute with non-linearities and/or connecting operators, as Appendix G further elucidates. Since we estimated various constants $(B_n, D_n, ...)$ of the individual layers by global ones, the derived stability constant is clearly not tight. However it portrays requirements for stability to edge level perturbations well: While the (spectral) interplay of Section 3 remains important, it is now especially large single-filter stability constants in the sense of Lemmata 4.1 and 4.2 that should be penalized during training.

## 5 STABILITY TO STRUCTURAL PERTURBATIONS: TRANSFERABILITY

While the demand that $\|\widetilde{T}J - JT\|$ be small in some norm is well adapted to capture some notions of closeness of graphs and characteristic operators, it is too stringent to capture others. As an illustrative example, further developed in Section 5.2 and numerically investigated in Section 7 below, suppose we are given a connected undirected graph with all edge weights of order $\mathcal{O}(1/\delta)$. With the Laplacian as characteristic operator (governing heat-flow in Physics (Cole, 2011)), we may think of this graph as modelling an array of coupled heat reservoirs with edge weights corresponding to heat-conductivities. As $1/\delta \to \infty$, the conductivities between respective nodes tend to infinity, heat exchange is instantaneous and all nodes act as if they are fused together into a single large entity – with the graph together with its characteristic operator behaving as an effective one-dimensional system. This 'convergent' behaviour is however not reflected in our characteristic operator, the graph Laplacian

$\Delta_\delta$: Clearly $\|\Delta_\delta\|_{op} = 1/\delta \cdot \|\Delta_1\|_{op} \to \infty$ as $1/\delta \to \infty$. Moreover, we would also expect a Cauchy-like behaviour from a 'convergent system', in the sense that if we for example keep $1/\delta_a - 1/\delta_b = 1$ constant but let $(1/\delta_a), (1/\delta_b) \to \infty$ we would expect $\|\Delta_{\delta_a} - \Delta_{\delta_b}\|_{op} \to 0$ by a triangle-inequality argument. However, we clearly have $\|\Delta_{\delta_a} - \Delta_{\delta_b}\|_{op} = |1/\delta_a - 1/\delta_b| \cdot \|\Delta_1\|_{op} = \|\Delta_1\|_{op}$, which does not decay. The situation is different however, when considering **resolvents** of the graph Laplacian. An easy calculation (c.f. Appendix H) yields $\|(\omega Id - \Delta_{\delta_b})^{-1} - (\omega Id - \Delta_{\delta_a})^{-1}\|_{op} = \mathcal{O}(\delta_a \cdot \delta_b)$ so that we recover the expected Cauchy behaviour. What is more, we also find the convergence $(\omega Id - \Delta_\delta)^{-1} \to P_0 \cdot (\omega - 0)^{-1}$; where $P_0$ denotes the projection onto the one-dimensional lowest lying eigenspace of the $\Delta_\delta$s (spanned by the vectors with constant entries). We may interpret $(\omega - 0)^{-1}$ as the resolvent of the graph Laplacian of a singleton (since such a Laplacian is identically zero) and thus now indeed find our physical intuition about convergence to a one-dimensional system reflected in our formulae. Motivated by this example, Section 5.1 develops a general theory for the difference in outputs of networks evaluated on graphs for which the **resolvents** $R_\omega := (\omega Id - T)^{-1}$ and $\widetilde{R}_\omega := (\omega Id - \widetilde{T})^{-1}$ of the respective characteristic operators are close in some sense. Subsequently, Section 5.2 then further develops our initial example while also considering an additional setting.

## 5.1 GENERAL THEORY

Throughout this section we fix a complex number $\omega \in \mathbb{C}$ and for each operator $T$ assume $\omega, \overline{\omega} \notin \sigma(T)$. This is always true for $\omega$ with $|\omega| \geqslant \|T\|_{op}$, but if $T$ is additionally self adjoint one could set $\omega = i$. If $T$ is non-negative one might choose $\omega = (-1)$). As a first step, we then note that the conclusion of Lemma 4.1 can always be satisfied if we chose $J \equiv 0$. To exclude this case – where the application of $J$ corresponds to losing too much information – we follow Post (2012) in making the following definition:

**Definition 5.1.** Let $J : \ell^2(G) \to \ell^2(\widetilde{G})$ and $\widetilde{J} : \ell^2(\widetilde{G}) \to \ell^2(G)$ be linear, and let $T$ $(\widetilde{T})$ be operators on $(\ell^2(G))$ $(\ell^2(\widetilde{G}))$. We say that $J$ and $\widetilde{J}$ are $\epsilon$-quasi-unitary with respect to $T, \widetilde{T}$ and $\omega$ if

$$\|Jf\|_{\ell^2(\widetilde{G})} \leqslant 2\|f\|_{\ell^2(G)}, \quad \|(J - \widetilde{J}^*)f\|_{\ell^2(\widetilde{G})} \leqslant \epsilon\|f\|_{\ell^2(G)},$$

$$\|(Id - \widetilde{J}J)R_\omega f\|_{\ell^2(G)} \leqslant \epsilon\|f\|_{\ell^2(G)}, \quad \|(Id - J\widetilde{J})\widetilde{R}_\omega u\|_{\ell^2(\widetilde{G})} \leqslant \epsilon\|u\|_{\ell^2(\widetilde{G})}. \tag{4}$$

The motivation to include the resolvents in the norm estimates (4) comes from the setting where $T = \Delta$ is the graph Laplacian and $\omega = (-1)$. In that case, the left equation in (4 is for example automatically fulfilled when demanding $\|(Id - \widetilde{J}J)f\|_{\ell^2(G)}^2 \leqslant \epsilon(\|f\|^2 + \mathcal{E}_\Delta(f))^{\frac{1}{2}}$, with $\mathcal{E}_\Delta(\cdot) = \langle \cdot, \Delta \cdot \rangle_{\ell^2(G)}$ the (positive) energy form induced by the Laplacian $\Delta$ (Post, 2012). This can thus be interpreted as a relaxation of the standard demand $\|(Id - \widetilde{J}J)\|_{op} \leqslant \epsilon$. Relaxing the demands of Section 4, we now demand closeness of resolvents instead of closeness of operators:

**Definition 5.2.** If, for $\omega \in \mathbb{C}$ and linear $J : \ell^2(G) \to \ell^2(\widetilde{G})$ the resolvents $R_\omega$ and $\widetilde{R}_\omega$ satisfy $\|(\widetilde{R}_\omega J - JR_\omega)f\|_{\ell^2(\widetilde{G})} \leqslant \epsilon\|f\|_{\ell^2(G)}$ for all $f \in \ell^2(G)$, $T$ and $\widetilde{T}$ are called $\omega$-$\epsilon$-**close** with identification operator $J$. If additonally $\|(\widetilde{R}_\omega^* J - JR_\omega^*)f\|_{\ell^2(\widetilde{G})} \leqslant \epsilon\|f\|_{\ell^2(G)}$, they are **doubly $\omega$-$\epsilon$-close**.

Our first result establishes that operators being (doubly-)$\omega$-$\epsilon$-close indeed has useful consequences:

**Lemma 5.3.** Let $T$ $(\widetilde{T})$ be operators on $\ell^2(G)$ $(\ell^2(\widetilde{G}))$. If these operators are $\omega$-$\epsilon$-close with identification operator $J$, and $\|R_\omega\|_{op}, \|\widetilde{R}_\omega\|_{op} \leqslant C$ we have $\|Jg(T) - g(\widetilde{T})J\|_{op} \leqslant K_g \cdot \|(\widetilde{R}_\omega J - JR_\omega)\|_{op}$ with $K_g = \frac{1}{2\pi}\oint_{\partial D}(1 + |z - \omega|\gamma_T(z))(1 + |z - \omega|\gamma_{\widetilde{T}}(z))|g(z)||dz|$ for holomorphic $g$, $K_g = \|g\|_{\mathscr{F}_{\omega,C}^{hol}}$ if $g \in \mathscr{F}_{\omega,C}^{hol}$ and $K_g = \|g\|_{\mathscr{F}_{\omega,C}^{cont}}$ for $T, \widetilde{T}$ normal and doubly $\omega$-$\epsilon$-close.

This result may then be extended to entire networks, as detailed in Theorem 5.4 below whose statement persists with slightly altered stability constants, if identification operators only *almost* commute with non-linearities and/or connecting operators. Proofs are contained in Appendix I.

**Theorem 5.4.** Let $\Phi_N, \widetilde{\Phi}_N$ be the maps associated to $N$-layer graph convolutional networks with the same non-linearities and functional calculus filters, but based on different graph signal spaces $\ell^2(G_n), \ell^2(\widetilde{G}_n)$, characteristic operators $T_n, \widetilde{T}_n$ and connecting operators $P_n, \widetilde{P}_n$. Assume $B_n, \widetilde{B}_n \leqslant B$ as well as $R_n, \widetilde{R}_n \leqslant R$ and $L_n \leqslant L$ for some $B, R, L > 0$ and all $n \geqslant 0$. Assume that

there are identification operators $J_n : \ell^2(G_n) \to \ell^2(\widetilde{G}_n)$ $(0 \leqslant n \leqslant N)$ commuting with non-linearities and connecting operators in the sense of $\|\widetilde{P}_n J_{n-1} f - J_n P_n f\|_{\ell^2(\widetilde{G}_n)} = 0$ and $\|\rho_n(J_n f) - J_n \rho_n(f)\|_{\ell^2(\widetilde{G}_n)} = 0$. define $D_n^2 := \sum_{j \in K_{n-1}} \sum_{i \in K_n} K_{g_{ij}^n}^2$ with $K_{g_{ij}^n}$ as in Lemma 5.3. Choose $D$ such that $D_n \leqslant D$ for all $n$. Finally assume that $\|J_n(\omega Id - T_n)^{-1} - (\omega Id - \widetilde{T}_n)^{-1} J_n\|_{op} \leqslant \epsilon$. If filters in $\mathscr{F}_{\omega,C}^{cont}$ are used, assume additionally that $\|J_n((\omega Id - T_n)^{-1})^* - ((\omega Id - \widetilde{T}_n)^{-1})^* J_n\|_{op} \leqslant \epsilon$. Then we have for all $f \in \mathscr{L}_{\mathrm{in}}$ and with $\mathscr{J}_n$ the operator that the $K_n$ copies of $J_n$ induce through concatenation that $\|\widetilde{\Phi}_N(\mathscr{J}_0 f) - \mathscr{J}_N \Phi_N(f)\|_{\widetilde{\mathscr{L}}_{out}} \leqslant N \cdot DRL \cdot (BRL)^{N-1} \cdot \|f\|_{\mathscr{L}_{in}} \cdot \epsilon$.

## 5.2 EXEMPLARY APPLICATIONS

**Collapsing Strong Edges:** We first pick our example from the beginning of section 5 up again and generalize it significantly: We now consider the graph that we collapse to a single node to be a sub-graph (of strong edges) embedded into a larger graph. Apart from coupled heat reservoirs, this setting also e.g. captures the grouping of close knit communities within social networks into single entities, the scale-transition of changing the description of (the graph of) a molecule from individual atoms interacting via the coulomb potential $Z_1 Z_2 / R$ (with $R$ the distance and $Z_1, Z_2$ atomic charges) to the interaction of (functional) groups comprised of closely co-located atoms, or spatial networks if weights are set to e.g. inverse distances. In what follows, we shall consider two graphs with vertex sets $G$ and $\widetilde{G}$. We consider $G$ to be a subset of the vertex set $\widetilde{G}$ and think of the graph corresponding to $G$ as arising in a collapsing procedure from the 'larger' graph $\widetilde{G}$.

More precisely, we assume that the vertex set $\widetilde{G}$ can be split into three disjoint subsets $\widetilde{G} = \widetilde{G}_{Latin} \bigcup \widetilde{G}_{Greek} \bigcup \{\star\}$ (c.f. also Fig. 3). We assume that the adjacency matrix $\widetilde{W}$ when restricted to Latin vertices or a Latin vertex and the exceptional node '$\star$' is of order unity ($\widetilde{W}_{ab}, \widetilde{W}_{a\star} = \mathcal{O}(1), \forall a, b \in \widetilde{G}_{Latin}$). For Greek indices, we assume that we may write $\widetilde{W}_{\alpha\beta} = \frac{\omega_{\alpha\beta}}{\delta}$ and $\widetilde{W}_{\alpha\star} = \frac{\omega_{\alpha\star}}{\delta}$ such that ($\omega_{\alpha\beta}, \omega_{\alpha\star} = \mathcal{O}(1)$ for all $\alpha, \beta \in \widetilde{G}_{Greek}$. We also assume that the sub-graph corresponding to vertices in $\widetilde{G}_{Greek} \bigcup \{\star\}$ is connected. We then take $G = \widetilde{G}_{Latin} \bigcup \{\star\}$ (c.f. again Fig. 3). The adjacency matrix $W$ on this graph is constructed by defining $W_{ab} = \widetilde{W}_{ab}, \forall a, b \in \widetilde{G}_{Latin}$ and setting (with $W_{a\star} \equiv W_{\star a}$)

$$W_{\star a} := \widetilde{W}_{a\star} + \sum_{\beta \in \widetilde{G}_{Greek}} \widetilde{W}_{a\beta} \quad \left( \forall a \in \widetilde{G}_{Latin} \right).$$

$W_{d\star} = \widetilde{W}_{d\alpha} + \widetilde{W}_{d\star}$
$W_{b\star} = \widetilde{W}_{b\star}$
$W_{ac} = \widetilde{W}_{ac}$

■ = edge-weight $\mathcal{O}(1/\delta)$
— = edge-weight $\mathcal{O}(1)$

Figure 3: Collapsed (left) and original (right) Graphs

We also allow our graph $\widetilde{G}$ to posses node-weights $\{\widetilde{\mu}_{\widetilde{g}}\}_{\widetilde{g} \in \widetilde{G}}$ that are not necessarily equal to one. The Laplace operator $\Delta_{\widetilde{G}}$ acting on the graph signal space $\ell^2(\widetilde{G})$ induces a positive semi-definite and convex energy form on this signal space via $E_{\widetilde{G}}(u) := \langle u, \Delta_{\widetilde{G}} u \rangle_{\ell^2(\widetilde{G})} = \sum_{g,h \in \widetilde{G}} \widetilde{W}_{gh} |u(g) - u(h)|^2$. Using this energy form, we now define a set comprised of $|G|$ signals, all of which live in $\ell^2(\widetilde{G})$. These signals are used to facilitate contact between the respective graph signal spaces $\ell^2(G)$ and $\ell^2(\widetilde{G})$.

**Definition 5.5.** For each $g \in G$, define the signal $\psi_g^\delta \in \ell^2(\widetilde{G})$ as the unique solution to the convex optimization program

$$\min E_{\widetilde{G}}(u) \quad subject \ to \ u(h) = \delta_{hg} \ for \ all \ h \in \widetilde{G}_{Latin} \bigcup \{\star\}. \tag{5}$$

Given the boundary conditions, what is left to determine in the above optimization program are the 'Greek entries' $\psi_g^\delta(\alpha)$ of each $\psi_g^\delta$. As Appendix J further elucidates, these can be calculated explicitly and purely in terms of the inverse of $\Delta_{\widetilde{G}}$ restricted to Greek indices as well as (sub-)columns of the adjacency matrix $\widetilde{W}$. Node-weights on $G$ are then defined as $\mu_g^\delta := \sum_{h \in \widetilde{G}} \psi_g^\delta(h) \cdot \widetilde{\mu}_h$. We denote the corresponding signal space by $\ell^2(G)$. Importantly, one has $\mu_a^\delta \to \widetilde{\mu}_a$ for any Latin index and $\mu_\star^\delta \to \widetilde{\mu}_\star + \sum_{\alpha \in \widetilde{G}_{Greek}} \widetilde{\mu}_\alpha$ as $\delta \to 0$; which recovers our physical intuition about heat reservoirs. To

translate signals from $\ell^2(G)$ to $\ell^2(\widetilde{G})$ and back, we define two identification operators $J : \ell^2(G) \rightarrow \ell^2(\widetilde{G})$ and $\widetilde{J} : \ell^2(\widetilde{G}) \rightarrow \ell^2(G)$ via $Jf := \sum_{g \in G} f(g) \cdot \psi_g^\delta$ and $(\widetilde{J}u)(g) := \langle u, \psi_g^\delta \rangle_{\ell^2(\widetilde{G})}/\mu_g^\delta$ for all $f \in \ell^2(G)$, $u \in \ell^2(\widetilde{G})$ and $g \in G$. Our main theorem then states the following:

**Theorem 5.6.** With definitions and notation as above, there are constants $K_1, K_2 \geqslant 0$ such that the operators $J$ and $\widetilde{J}$ are $(K_1\sqrt{\delta})$-quasi-unitary with respect to $\Delta_{\widetilde{G}}$, $\Delta_G$ and $\omega = (-1)$. Furthermore, the operators $\Delta_{\widetilde{G}}$ and $\Delta_G$ are $(-1)$-$(K_2\sqrt{\delta})$ close. with identification operator $J$.

Appendix J presents the (fairly involved) proof of this result. Importantly, the size of the constants $K_1, K_2$ is independent of the cardinality (or more precisely the total weight) of $\widetilde{G}_{Latin}$, implying that Theorem 5.6 also remains applicable in the realm of large graphs. Finally we note, that this stability result is contingent on the use of the (un-normalized) graph Laplacian (c.f. Appendix K):

**Theorem 5.7.** In the setting of Theorem 5.6 denote by $T$ $(\widetilde{T})$ adjacency matrices or normalized graph Laplacians on $\ell^2(G)$ $(\ell^2(G))$. There are no functions $\eta_1, \eta_2 : [0, 1] \rightarrow \mathbb{R}_{\geqslant 0}$ with $\eta_i(\delta) \rightarrow 0$ as $\delta \rightarrow 0$ $(i = 1, 2)$, families of identification operators $J^\delta, \widetilde{J}^\delta$ and $\omega \in \mathbb{C}$ so that $J^\delta$ and $\widetilde{J}^\delta$ are $\eta_1(\delta)$-quasi-unitary with respect to $\widetilde{T}$, $T$ and $\omega$ while the operators $\widetilde{T}$ and $T$ remain $\omega$-$\eta_2(\delta)$ close.

**The Realm of Large Graphs:** In order to relate our transferability framework to the literature, we consider an 'increasing' sequence of graphs $(G_n \subseteq G_{n+1})$ approximating a limit object, so that the transferability framework of Levie et al. (2019a) is also applicable. We choose the limit object to be the circle of circumference $2\pi$ and our approximating graphs to be the closed path-graph on $N$ vertices equidistantly embedded into the circle (c.f. Fig 4). With $h = 2\pi/N$ the node-distance, we set weights to $1/h^2$; ensuring consistency with the 'continuous' Laplacian in the limit $N \rightarrow \infty$. More details are presented in Appendix L, which also contains the proof of the corresponding transferability result:

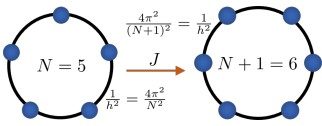

Figure 4: Closed Path-Graphs

**Theorem 5.8.** In the above setting choose all node-weights equal to one and $N$ to be odd for definiteness. There exists constants $K_1, K_2 = \mathcal{O}(1)$ so that for each $N \geqslant 1$, there exist identification operators $J, \widetilde{J}$ mapping between $\ell^2(G_N)$ and $\ell^2(G_{N+1})$ so that $J$ and $\widetilde{J}$ are $(K_1/N)$-quasi-unitary with respect to $\Delta_{G_N}$, $\Delta_{G_{N+1}}$ and $\omega = (-1)$. Furthermore, the operators $\Delta_{G_N}$ and $\Delta_{G_{N+1}}$ are $(-1)$-$(K_2/N)$ close with identification operator $J$.

Lemma 5.3 then implies an $\mathcal{O}(\frac{1}{N})$-decay of $\|g(T)J - Jg(\widetilde{T})\|_{op}$ for fixed $g$. This reduces to an $\mathcal{O}(\frac{\sqrt{N}}{N})$-decay for Levie et al. (2019a) (ibid. Theorem 5, pt. 3) assuming a similar decay of operator-distances. Our framework might this capture transferability properties other approaches could miss.

## 6 GRAPH LEVEL STABILITY

To solve tasks such as graph classification or regression over multiple graphs, graphs of varying sizes need to be represented in a common feature space. Here we show that aggregating node-level features into such graph level features via $p$-norms ($\|f\|_{\ell^p(G)} := (\sum_{g \in G} |f_g|^p \mu_g)^{1/p}$) preserves stability. To this end, let $\mathscr{L}_{out}$ be a target space of a GCN in the sense of (3). On each of the (in total $K_{out}$) $\ell^2(G_{out})$ summands of $\mathscr{L}_{out}$, we may apply the map $f_i \mapsto \|f_i\|_{\ell^p(G_{out})}$. Stacking these maps, we build a map from $\mathscr{L}_{out}$ to $\mathbb{R}^{K_{out}}$. Concatenating the map $\Phi_N$ associated to an $N$-layer GCN with this map yields a map from $\mathscr{L}_{in}$ to $\mathbb{R}^{K_{out}}$. We denote it by $\Psi_N^p$ and find:

Figure 5: Graph Level Aggregation

**Theorem 6.1.** For $p \geqslant 2$ we have in the setting of Theorem 3.1 that $\|\Psi_N^p(f) - \Psi_N^p(h)\|_{\mathbb{R}^{K_{out}}} \leqslant \left( \prod_{n=1}^N L_n R_n B_n \right) \cdot \|f - h\|_{\mathscr{L}_{in}}$. In the setting of Theorem 4.3 or 5.4 and under the additional assumption that the 'final' identification operator $J_N$ satisfies $\left| \|J_N f_i\|_{\ell^k(\widetilde{G}_N)} - \|f_i\|_{\ell^k(G_N)} \right| \leqslant \delta \cdot K \cdot \|f_i\|_{\ell^2(G_N)}$ for all $f_i \in \ell^2(G_N)$, we have $\|\Psi_N^p(f) - \widetilde{\Psi}_N^p(\mathscr{J}_0 f)\|_{\mathbb{R}^{K_{out}}} \leqslant (N \cdot DRL + K \cdot (BRL)) \cdot (BRL)^{N-1} \cdot \|f\|_{\mathscr{L}_{in}} \cdot \delta$.

Derived stability results thus persist (under mild assumptions) if graph level features are aggregated via $p$-norms. Appendix M contains the corresponding proof.

## 7 NUMERICAL RESULTS

We focus on investigating structural perturbations, as corresponding results are most involved and novel:

We first consider a graph on 5 nodes with an adjacency matrix $A$ with $\mathcal{O}(1)$-entries (c.f. 30 in Appendix N). We then scale $A$ by $1/\delta_a$ and $1/\delta_b$ (with $\frac{1}{\delta_a} - \frac{1}{\delta_b} = 1$) respectively and consider the norm-difference between associated Laplacians and resolvents. Fig. 6 (a) then illustrate the theoretical result (c.f. Section 5) that resolvent- instead of Laplacian-differences capture the convergence behaviour. Embedding the considered graph into a larger graph ($\widetilde{W} \in \mathbb{R}^{8 \times 8}$; c.f. (31) in Appendix N), we consider the collapsing edge setting of Section 5.2 in Fig. 6 (b). As expected, the corresponding resolvents do approach each other as $\delta \to 0$. Contrary to the theoretical bound in Lemma 5.3, differences of resolvent-monomials decrease as their power $k$ increases.

Beyond small graphs – inaccessible to traditional asymptotic methods – our method is also applicable to the large-graph setting: Fig. 7 picks up the example of an 'increasing' graph sequence 'approximating' the circle again. As predicted in Section 5.2, the difference in resolvents decays ($\propto \frac{1}{N}$). Fig. 10 in Appendix N shows how the difference in Laplacians diverges instead. Hence

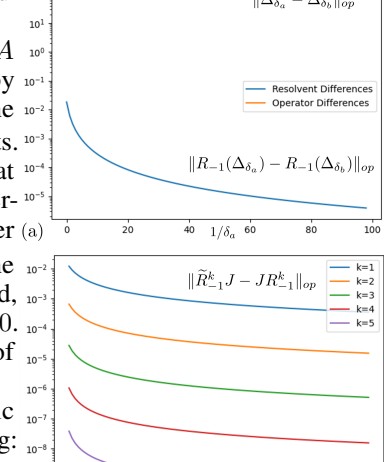

(a)

(b)

Figure 6: Edge-Collapse Stability

our framework might capture stability properties traditional approaches could miss.

Finally, we investigate the transferability of a two-layer GCN with 16 nodes per hidden Layer combined with the aggregation method of Section 6 into a graph-level map $\Psi_2^p$. Filters are of the form (2) up to order $k = 11$. Coefficients $\{b_k^g\}$ are sampled uniformly from $[-100, 100]$. Feature vectors are generated on the QM7 dataset. There each graph represents a molecule; nodes correspond to individual atoms. Adjacency matrices are given by $\widetilde{W}_{ij} = Z_i Z_j / \|x_i - x_j\|$ with $Z_i$ ($x_i$) the atomic charge (equilibrium position) of atom $i$. We choose node-weights as $\widetilde{\mu}_i = Z_i$ and the Laplacian as characteristic operator. Leading up to Fig. 8

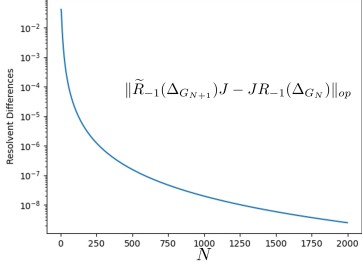

Figure 7: The Large-$N$ Regime

we consider the graph of methane (5 Nodes; one Carbon ($Z_1 = 6$) and four Hydrogen nodes ($Z_{i>1} = 1$)) and deflect one of the Hydrogen atoms ($i = 2$) out of equilibrium and along a straight line towards the Carbon atom. We then consider the transferability of the entire GCN between the resulting graph and an effective graph combining Carbon and deflected Hydrogen into a single node "$\star$" with weight $\mu_\star = Z_1 + Z_2 = 7$ located at the equilibrium position of Carbon. With $J$ translating from effective to original description, we consider $\|\Psi_2^p(f) - \Psi_2^p(Jf)\|_{\mathbb{R}^{16}}$ (averaged over 100 random unit-norm choices of $f$) as a function of $\|x_1 - x_2\|^{-1}$. At equilibrium the transferability error is $\mathcal{O}(1)$. It decreases fast with decreasing Carbon-Hydrogen distance, with the choice of representation (effective vs. original) quickly becoming insignificant for generated feature vectors.

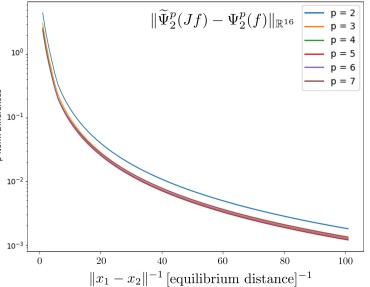

Figure 8: GCN Transferability

## 8 DISCUSSION

A theoretically well founded framework capturing stability properties of GCNs was developed. We related node-level stability to (spectral) covering properties and edge-level stability to introduced semi-norms of employed filters. For non-normal characteristic operators, tools from complex analysis provided grounds for derived stability properties. We introduced a new notion of stability to structural perturbations, highlighted the importance of the resolvent and detailed how the developed line of thought captures relevant settings of structural changes such as the collapse of a strongly connected sub-graph to a node. There – precisely if the graph Laplacian was employed – the transferability error could be bounded in terms of the inverse characteristic coupling strength on the sub-graph.

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

## A  SOME CONCEPTS IN LINEAR ALGEBRA

In the interest of self-containedness, we provide a brief review of some concepts from linear algebra utilized in this work that might potentially be considered more advanced. Presented results are all standard; a very thorough reference is Michael Reed (1981).

**Hilbert Spaces:** To us, a Hilbert space — often denoted by $\mathcal{H}$ — is a vector space over the complex numbers which also has an inner product — often denoted by $\langle \cdot, \cdot \rangle_{\mathcal{H}}$. Prototypical examples are given by the Euclidean spaces $\mathbb{C}^d$ with inner product $\langle x, y \rangle_{\mathbb{C}^d} := \sum_{i=1}^d \overline{x}_i y_i$. Associated to an inner product is a norm, denoted by $\| \cdot \|_{\mathcal{H}}$ and defined by $\|x\|_{\mathcal{H}} := \sqrt{\langle x, x \rangle_{\mathcal{H}}}$ for $x \in \mathcal{H}$.

**Direct Sums of Spaces:** Given two potentially different Hilbert spaces $\mathcal{H}$ and $\widehat{\mathcal{H}}$, one can form their direct sum $\mathcal{H} \oplus \widehat{\mathcal{H}}$. Elements of $\mathcal{H} \oplus \widehat{\mathcal{H}}$ are vectors of the form $(a, b)$, with $a \in \mathcal{H}$ and $b \in \widehat{\mathcal{H}}$. Addition and scalar multiplication are defined in the obvious way by

$$(a, b) + \lambda(c, d) := (a + \lambda c, b + \lambda d)$$

for $a, c \in \mathcal{H}$, $b, d \in \widehat{\mathcal{H}}$ and $\lambda \in \mathbb{C}$. The inner product on the direct sum is defined by

$$\langle (a, b), (c, d) \rangle_{\mathcal{H} \oplus \widehat{\mathcal{H}}} := \langle a, c \rangle_{\mathcal{H}} + \langle b, d \rangle_{\widehat{\mathcal{H}}}.$$

As is readily checked, this implies that the norm $\| \cdot \|_{\mathcal{H} \oplus \widehat{\mathcal{H}}}$ on the direct sum is given by

$$\|(a, b)\|^2_{\mathcal{H} \oplus \widehat{\mathcal{H}}} := \|a\|^2_{\mathcal{H}} + \|b\|^2_{\widehat{\mathcal{H}}}.$$

Standard examples of direct sums are again the Euclidean spaces, where one has $\mathbb{C}^d = \mathbb{C}^n \oplus \mathbb{C}^m$ if $m + n = d$, as is easily checked. One might also consider direct sums with more than two summands, writing $\mathbb{C}^d = \oplus_{i=1}^d \mathbb{C}$ for example. In fact, one might also consider infinite sums of Hilbert spaces: The space $\oplus_{i=1}^\infty \mathcal{H}_i$ is made up of those elements $a = (a_1, a_2, a_3, ...)$ with $a_i \in \mathcal{H}_i$ for which the norm

$$\|a\|^2_{\oplus_{i=1}^\infty \mathcal{H}_i} := \sum_{i=1}^\infty \|a_i\|^2_{\mathcal{H}_i}$$

is finite. This means for example that the vector $(1, 0, 0, 0, ...)$ is in $\oplus_{i=1}^\infty \mathbb{C}$, while $(1, 1, 1, 1, ...)$ is not.

**Direct Sums of Maps:** Suppose we have two collections of Hilbert spaces $\{\mathcal{H}_i\}_{i=1}^\Gamma$, $\{\widetilde{\mathcal{H}}_i\}_{i=1}^\Gamma$ with $\Gamma \in \mathbb{N}$ or $\Gamma = \infty$. Suppose further that for each $i \leqslant \Gamma$ (resp. $i < \Gamma$) we have a (not necessarily linear) map $J_i : \mathcal{H}_i \to \widetilde{\mathcal{H}}_i$. Then the collection $\{J_i\}_{i=1}^\Gamma$ of these 'component' maps induce a 'composite' map

$$\mathscr{J} : \oplus_{i=1}^\Gamma \mathcal{H}_i \longrightarrow \oplus_{i=1}^\Gamma \widetilde{\mathcal{H}}_i$$

between the direct sums. Its value on an element $a = (a_1, a_2, a_3, ...) \in \oplus_{i=1}^\Gamma \mathcal{H}_i$ is defined by

$$\mathscr{J}(a) = (J_1(a_1), J_2(a_2), J_3(a_3), ...) \in \oplus_{i=1}^\Gamma \widetilde{\mathcal{H}}_i.$$

Strictly speaking, one has to be a bit more careful in the case where $\Gamma = \infty$ to ensure that $\|\mathscr{J}(a)\|_{\oplus_{i=1}^\infty \widetilde{\mathcal{H}}_i} \neq \infty$. This can however be ensured if we have $\|J_i(a_i)\|_{\widetilde{\mathcal{H}}_i} \leqslant C\|a_i\|_{\mathcal{H}_i}$ for all $1 \leqslant i$ and some $C$ independent of all $i$, since then $\|\mathscr{J}(a)\|_{\oplus_{i=1}^\infty \widetilde{\mathcal{H}}_i} \leqslant C\|a\|_{\oplus_{i=1}^\infty \mathcal{H}_i} \leqslant \infty$. If each $J_i$ is a linear operator, such a $C$ exists precisely if the operator norms (defined below) of all $J_i$ are smaller than some constant.

**Operator Norm:** Let $J : \mathcal{H} \to \widetilde{\mathcal{H}}$ be a linear operator between Hilbert spaces. We measure its 'size' by what is called the operator norm, denoted by $\| \cdot \|_{op}$ and defined by

$$\|J\|_{op} := \sup_{\psi \in \mathcal{H}, \|\psi\|_{\mathcal{H}} = 1} \frac{\|A\psi\|_{\widetilde{\mathcal{H}}}}{\|\psi\|_{\mathcal{H}}}.$$

**Adjoint Operators** Let $J : \mathcal{H} \to \widetilde{\mathcal{H}}$ be a linear operator from the Hilbert space $\mathcal{H}$ to the Hilbert space $\widetilde{\mathcal{H}}$. Its adjoint $J^* : \widetilde{\mathcal{H}} \to \mathcal{H}$ is an operator mapping in the opposite direction. It is uniquely determined by demanding that

$$\langle Jf, u \rangle_{\widetilde{\mathcal{H}}} = \langle f, J^*u \rangle_{\mathcal{H}}$$

holds true for arbitrary $f \in \mathcal{H}$ and $u \in \widetilde{\mathcal{H}}$.

**Normal Operators:** If a linear operator $\Delta : \mathcal{H} \to \mathcal{H}$ maps from and to the same Hilbert space, we can compare it directly with its adjoint. If $\Delta\Delta^* = \Delta^*\Delta$, we say that the operator $\Delta$ is normal. Special instances of normal operators are self-adjoint operators, for which we have the stronger property $\Delta = \Delta^*$. If an operator is normal, there are unitary maps $U : \mathcal{H} \to \mathcal{H}$ diagonalizing $\Delta$ as

$$U^*\Delta U = \mathrm{diag}(\lambda_1, ...\lambda_n),$$

with eigenvalues in $\mathbb{C}$. We call the collection of eigenvalues the spectrum $\sigma(\Delta)$ of $\Delta$. If $\dim \mathcal{H} = d$, we may write $\sigma(\Delta) = \{\lambda\}_{i=1}^d$. It is a standard exercise to verify that each eigenvalue satisfies $|\lambda_i| \leqslant \|\Delta\|_{op}$. Associated to each eigenvalue is an eigenvector $\phi_i$. The collection of all (normalized) eigenvectors forms an orthonormal basis of $\mathcal{H}$. We may then write

$$\Delta f = \sum_{i=1}^d \lambda_i \langle \phi_i, f \rangle_{\mathcal{H}} \phi_i.$$

**Resolvent of an Operator:** Given an operator $T$ on some Hilbert space $\mathcal{H}$, we have by definition that the operator $(T - z) : \mathcal{H} \to \mathcal{H}$ is invertible precisely if $z \neq \sigma(T)$. In this case we write

$$R_z(T) = (zId - T)^{-1}$$

and call this operator the **resolvent** of $T$ at $z$.

If $T$ is normal it can be proved that the norm of the resolvent satisfies

$$\|R_z(T)\|_{op} = \frac{1}{dist(z, \sigma(\Delta))},$$

where $dist(z, \sigma(\Delta))$ denotes the minimal distance between $z$ and any eigenvalue of $\Delta$. For non-normal operators, one can prove

$$\|R_z(T)\|_{op} \leqslant \gamma_T(z)$$

with

$$\gamma_T(z) = \exp\left[2\|T\|_1/d(z, \sigma(T))\right]/d(z, \sigma(T))$$

as is proved in Bandtlow (2004a).

**Frobenius Norm:** Given two finite dimensional Hilbert spaces $\mathcal{H}_1$ and $\mathcal{H}_2$ with orthonormal bases $\{\phi_i^1\}_{i=1}^{d_1}$ and $\{\phi_i^1\}_{i=1}^{d_1}$, the Frobenius norm $\|\cdot\|_F$ of an operator $A : \mathcal{H}_1 \to \mathcal{H}_2$ may be defined as

$$\|A\|_2^2 := \sum_{i=1}^{d_2} \sum_{j=1}^{d_1} |A_{ij}|^2$$

with $A_{ij}$ the matrix representation of $A$ with respect to the bases $\{\phi_i^1\}_{i=1}^{d_1}$ and $\{\phi_i^1\}_{i=1}^{d_1}$. It is a standard exercise to verify that this norm is indeed independent of any choice of basis and hence invariant under multiplying $A$ with a unitary on either the left or the right side. More precisely, if $U : \mathcal{H}_2 \to \mathcal{H}_2$ and $V : \mathcal{H}_1 \to \mathcal{H}_1$ are unitary, we have

$$\|UAV\|_F^2 = \|A\|_F^2.$$

Frobenius norms can be used to transfer Lipschitz continuity properties of complex functions to the setting of functions applied to normal operators:

**Lemma A.1.** Let $g : \mathbb{C} \to \mathbb{C}$ be Lipschitz continuous with Lipschitz constant $D_g$. This implies

$$\|g(X)J - Jg(Y)\|_F \leqslant D_g \cdot \|X - Y\|_F.$$

for normal operators $X$ on $\mathcal{H}_2$, $Y$ on $\mathcal{H}_1$ and any linear map $J : \mathcal{H}_1 \to \mathcal{H}_2$.

*Proof.* This proof is a modified version of the proof in Wihler (2009). Let $U, W$ be unitary (with respect to the inner product $\langle \cdot, \cdot \rangle_{\mathcal{H}}$) operators diagonalizing the normal operators $X$ and $Y$ as

$$V^*XV = \mathrm{diag}(\lambda_1, ...\lambda_{d_2}) =: D(X)$$
$$W^*YW = \mathrm{diag}(\mu_1, ...\mu_{d_1}) =: D(Y).$$

Since the Frobenius norm is invariant under unitary transformations we find

$$
\begin{aligned}
\|g(X)J - Jg(Y)\|_F^2 &= \|g(VD(X)V^*) - g(WD(Y)W^*)\|_F^2 \\
&= \|Vg(D(X))V^*J - JWg(D(Y))W^*\|_F^2 \\
&= \|g(D(X))V^*JW - V^*JWg(D(Y))\|_F^2 \\
&= \sum_{i,j} |(g(D(X))V^*JW - V^*JWg(D(Y)))_{ij}|^2 \\
&= \sum_{i,j} \left| \sum_k [g(D(X))]_{ik}[V^*JW]_{kj} - [V^*JW]_{ik}[g(D(Y))]_{kj} \right|^2 \\
&= \sum_{i,j} |[V^*W]_{ij}|^2 |g(\lambda_j) - g(\mu_i)|^2 \\
&\leqslant \sum_{i,j} |[V^*W]_{ij}|^2 D_g^2 |\lambda_j - \mu_i|^2 \\
&= D_g^2 \|X - Y\|_F^2.
\end{aligned}
$$

$\square$

## B  APPROXIMATING BOUNDED CONTINUOUS FILTERS

Let us recall Definition 2.1:

**Definition B.1.** Fix $\omega \in \mathbb{C}$ and $C > 0$. Define the space $\mathscr{F}_{\omega,C}^{cont}$ of continuous filters on $\mathbb{C}\backslash\{\omega,\overline{\omega}\}$, to be the space of multilinear power-series' $g(z) = \sum_{\mu,\nu=0}^{\infty} a_{\mu\nu} (\omega - z)^{-\mu} (\overline{\omega} - \overline{z})^{-\mu}$ for which the norm $\|g\|_{\mathscr{F}_{\omega,C}^{cont}} := \sum_{\mu,\nu=0}^{\infty} |\mu + \nu| C^{\mu+\nu} |a_{\mu\nu}|$ is finite.

We now prove that upon denoting by $B_\epsilon(\omega) \subseteq \mathbb{C}$ the open ball of radius $\epsilon$ around $\omega$, one can show that for arbitrary $\delta > 0$ and every continuous function $g$ defined on $\mathbb{C}\backslash(B_\epsilon(\omega) \cup B_\epsilon(\overline{\omega}))$ which is regular at infinity – i.e. satisfies $\lim_{r\to+\infty} g(rz) = c \in \mathbb{C}$ independent of which $z \neq 0$ is chosen – there is a function $f \in \mathscr{F}_{\omega,C}^{cont}$ so that $|f(z) - g(z)| \leqslant \delta$ for all $z \in \mathbb{C}\backslash(B_\epsilon(\omega) \cup B_\epsilon(\overline{\omega}))$.
Making use of the Stone-Weierstrass theorem for complex functions, it suffices to prove that for every point $z$ in $\mathbb{C}\backslash(B_\epsilon(\omega) \cup B_\epsilon(\overline{\omega}))$ there are functions $f$ and $g$ in $\mathscr{F}_{\omega,C}^{cont}$ for which

$$ f(z) \neq g(z). $$

But this is obvious since $(\omega - z)^{-1}$ is injective on $\mathbb{C}\backslash(B_\epsilon(\omega) \cup B_\epsilon(\overline{\omega}))$.

## C  COMPLEX ANALYSIS

A general reference for topics discussed in this section is Bak & Newman (2017).
For a complex valued function $f$ of a single complex variable, the derivative of $f$ at a point $z_0 \in \mathbb{C}$ in its domain of definition is defined as the limit

$$ f'(z_0) := \lim_{z \to z_0} \frac{f(z) - f(z_0)}{z - z_0}. $$

For this limit to exist, it needs to be independent of the 'direction' in which $z$ approaches $z_0$, which is a stronger requirement than being real-differentiable. A function is called holomorphic on an open set $U$ if it is complex differentiable at every point in $U$. It is called entire if it is complex differentiable at every point in $\mathbb{C}$. Every entire function has an everywhere convergent power series representation

$$ g(z) = \sum_{k=0}^{\infty} a^g z^k. \tag{6} $$

If a function $g$ is analytic (i.e. can be expanded into a power series), we have

$$ g(\lambda) = -\frac{1}{2\pi i} \oint_S \frac{g(z)}{\lambda - z} dz \tag{7} $$

for any circle $S \subseteq \mathbb{C}$ encircling $\lambda$ by Cauchy's integral formula.

In fact, the integration contour need not be a circle $S$, but may be the boundary of any so called Cauchy domain containing $\lambda$:

**Definition C.1.** A subset $D$ of the complex plane $\mathbb{C}$ is called a Cauchy domain if $D$ is open, has a finite number of components (the closure of two of which are disjoint) and the boundary of $\partial D$ of $D$ is composed of a finite number of closed rectifiable Jordan curves, no two of which intersect.

Equation (7) forms the backbone of complex analysis. Since the integral

$$I := -\frac{1}{2\pi i} \oint_{\partial D} g(z)(zId - T)^{-1}dz \tag{8}$$

is well defined for holomorphic $g(\cdot)$ and any operator $T$ for which $\sigma(T)$ and $\partial D$ are disjoint (c.f. e.g. Post (2012) for details), we can essentially take (8) as a defining equation through which one might apply holomorphic functions to operators.

While functions that are everywhere complex differentiable have a series representation according to (6), complex functions that are holomorphic only on $\mathbb{C}\backslash\{\omega\}$ have a series representation (called Laurent series) according to

$$g(z) = \sum_{k=-\infty}^{\infty} a_k(z - \omega)^k.$$

If these functions are assumed to be regular at infinity, no terms with positive exponent are permitted and (changing the indexing) we may thus write

$$g(z) = \sum_{k=0}^{\infty} a_k(z - \omega)^{-k}.$$

Motivated by this, we now prove the following consistency result:

**Lemma C.2.** With the notation of Section 2 we have for any $k \geqslant 1$ and $\omega \notin \sigma(T)$ that

$$(\omega \cdot Id - T)^{-k} := \frac{1}{2\pi i} \oint_{\partial D} (\omega - z)^{-k} \cdot (zId - T)^{-1}dz,$$

where we interpret the left hand side of the equation in terms of inversion and matrix powers.

*Proof.* We first note that we may write

$$R_\lambda(T) = \sum_{n=0}^{\infty} (\lambda - \omega)^n(-1)^n R_\omega(t)^{n+1}$$

for $|\lambda - \omega| \leqslant \|R_\omega(T)\|$ using standard results in matrix analysis (namely the 'Neumann Characterisation of the Resolvent' which is obtained by repeated application of a resolvent identity; c.f. Post (2012) for more details). We thus find

$$\frac{1}{2\pi i} \oint_{\partial D} \left(\frac{1}{\omega - z}\right)^k \frac{1}{zId - T}dz = \frac{1}{2\pi i} \oint_{\partial D} \left(\frac{1}{\omega - z}\right)^k \sum_{n=0}^{\infty} (\omega - z)^n R_\omega(T)^{n+1}.$$

Using the fact that

$$\frac{1}{2\pi i} \oint_{\partial D} (z - \omega)^{n-k-1}dz = \delta_{nk}$$

then yields the claim. $\qquad\square$

## D    PROOF OF LEMMA 2.3

We want to prove the following:

**Lemma D.1.** For holomorphic $g$ and generic $T$ we have $\|g(T)\|_{op} \leqslant |g(\infty)| + \frac{1}{2\pi} \oint_{\partial D} |g(z)| \gamma_T(z) d|z|$. Furthermore we have for any $T$ with $\gamma_T(\omega) \leqslant C$, that $\|g(T)\|_{op} \leqslant \|g\|_{\mathscr{F}_{\omega,C}^{hol}}$ as long as $g \in \mathscr{F}_{C,\omega}$.

*Proof.* We first note

$$\left\| g(\infty) \cdot Id + \frac{1}{2\pi i} \oint_{\partial D} g(z) \cdot (zId - T)^{-1} dz \right\|_{op} \leqslant \|g(\infty) \cdot Id\|_{op} + \left\| \frac{1}{2\pi i} \oint_{\partial D} g(z) \cdot (zId - T)^{-1} dz \right\|_{op}$$

$$\leqslant |g(\infty)| + \frac{1}{2\pi} \oint_{\partial D} |g(z)| \left\| \cdot (zId - T)^{-1} \right\|_{op} d|z|.$$

The first claim thus follows together with $\|R_z(T)\|_{op} \leqslant \gamma_T(z)$. The second claim can be derived as follows:

$$\|g(T)\|_{op} = \left\| \sum_{k=0}^{\infty} b_k^g (T - \omega)^{-k} \right\|_{op} \leqslant \sum_{k=0}^{\infty} |b_k^g| \left\| (T-\omega)^{-k} \right\|_{op} \leqslant \sum_{k=0}^{\infty} |b_k^g| \gamma_T(\omega)^k \leqslant \sum_{k=0}^{\infty} |b_k^g| C^k.$$

$\square$

## E    PROOF OF THEOREM 3.1 AND TIGHTNESS OF RESULTS

. We want to prove the following:

**Theorem E.1.** With the notation of Section 2 let $\Phi_N : \mathscr{L}_{\text{in}} \to \mathscr{L}_{\text{out}}$ be the map associated to an $N$-layer GCN. We have

$$\|\Phi_N(f) - \Phi_N(h)\|_{\mathscr{L}_{\text{out}}} \leqslant \left( \prod_{n=1}^{N} L_n R_n B_n \right) \cdot \|f - h\|_{\mathscr{L}_{\text{in}}}$$

with $B_n := \sqrt{\sup_{\lambda \in \sigma(T_n)} \sum_{j \in K_{n-1}} \sum_{i \in K_n} |g_{ij}^n(\lambda)|^2}$ if $T_n$ is normal. For general $T_n$ we have for all $\{g_{ij}\}$ entire, holomorphic and in $\mathscr{F}_{\omega,C}$ respectively:

$$B_n := \begin{cases} \sum_{k=0}^{\infty} \sqrt{\sum_{j \in K_{n-1}} \sum_{i \in K_n} |(a_{ij}^{g_n})_k|^2} \cdot \|T_n\|_{op}^k \\ \sqrt{\sum_{j \in K_{n-1}} \sum_{i \in K_n} \|g_{ij}^n(\infty)\|^2} + \frac{1}{2\pi} \oint_{\Gamma} \gamma_T(z) \sqrt{\sum_{j \in K_{n-1}} \sum_{i \in K_n} |g_{ij}^n(z)|^2} d|z| \\ \sqrt{\sum_{j \in K_{n-1}} \sum_{i \in K_n} \|g_{ij}^n\|_{\omega,C}^2} \end{cases}$$

*Proof.* Given input signals $f, h^n \in \mathscr{L}_{\text{in}}$, let us – sticking to the notation introduced in Section 2 – denote the intermediate signal representations in the intermediate layers $\mathscr{L}_n$ by $f^n, h^n \in \mathscr{L}_n$. With the update rule described in Section 2 and the norm induced on each $\mathscr{L}_n$ as described in Appendix A, we then have

$$\|f^{n+1} - h^{n+1}\|_{\mathscr{L}_{n+1}}^2$$

$$= \sum_{i=1}^{K_{n+1}} \left\| \rho_{n+1} \left( \sum_{j=1}^{K_n} g_{ij}^{n+1}(T_{n+1}) P_{n+1}(f_j^n) \right) - \rho_{n+1} \left( \sum_{j=1}^{K_n} g_{ij}^{n+1}(T_{n+1}) P_{n+1}(h_j^n) \right) \right\|_{\ell^2(G_{n+1})}^2$$

$$\leqslant L_{n+1}^2 \sum_{i=1}^{K_{n+1}} \left\| \sum_{j=1}^{K_n} g_{ij}^{n+1}(T_{n+1}) P_{n+1}(f_j^n) - \sum_{j=1}^{K_n} g_{ij}^{n+1}(T_{n+1}) P_{n+1}(h_j^n) \right\|_{\ell^2(G_{n+1})}^2$$

$$= L_{n+1}^2 \sum_{i=1}^{K_{n+1}} \left\| \sum_{j=1}^{K_n} g_{ij}^{n+1}(T_{n+1}) \left[ P_{n+1}(f_j^n) - P_{n+1}(h_j^n) \right] \right\|_{\ell^2(G_{n+1})}^2 .$$

We next note

$$\sum_{i=1}^{K_{n+1}} \left\| \sum_{j=1}^{K_n} g_{ij}^{n+1}(T_{n+1}) \left[ P_{n+1}(f_j^n) - P_{n+1}(h_j^n) \right] \right\|_{\ell^2(G_{n+1})}^2$$

$$\leqslant \sum_{i=1}^{K_{n+1}} \left( \sum_{j=1}^{K_n} \|g_{ij}^{n+1}(T_{n+1})\|_{op} \| \left[ P_{n+1}(f_j^n) - P_{n+1}(h_j^n) \right] \|_{\ell^2(G_{n+1})} \right)^2$$

$$\leqslant \left( \sum_{i=1}^{K_{n+1}} \sum_{j=1}^{K_n} \|g_{ij}^{n+1}(T_{n+1})\|_{op}^2 \right) \sum_{j=1}^{K_n} \|\| \left[ P_{n+1}(f_j^n) - P_{n+1}(h_j^n) \right] \|_{\ell^2(G_{n+1})}^2$$

$$\leqslant R_{n+1}^2 \left( \sum_{i=1}^{K_{n+1}} \sum_{j=1}^{K_n} \|g_{ij}^{n+1}(T_{n+1})\|_{op}^2 \right) \|\|f^n - h_j^n\|_{\mathscr{L}_n}^2$$

where the second to last step is an application of the Cauchy Schwarz inequality.

Proceeding inductively and using our previously established estimates, this proves the claim for all settings in which $T_n$ is nor normal (using an additional application of the triangle inequality for the case of holomorphic filters).

To prove the claim for normal $T_n$ as well, we note that in this setting we have (writing $(\phi_\alpha, \lambda_\alpha)_{\alpha=1}^{|G|}$ for a normalozed eigenvalue-eigenvector sequence of $T_{n+1}$) that we have

$$\sum_{i=1}^{K_{n+1}} \left\| \sum_{j=1}^{K_n} g_{ij}^{n+1}(T_{n+1}) \left[ P_{n+1}(f_j^n) - P_{n+1}(h_j^n) \right] \right\|_{\ell^2(G_{n+1})}^2$$

$$= \sum_{i=1}^{K_{n+1}} \left\| \sum_{j=1}^{K_n} \sum_\alpha g_{ij}^{n+1}(\lambda_\alpha) \langle \phi_\alpha, \left[ P_{n+1}(f_j^n) - P_{n+1}(h_j^n) \right] \rangle_{\ell^2(G_{n+1})} \phi_\alpha \right\|_{\ell^2(G_{n+1})}^2$$

$$= \sum_{i=1}^{K_{n+1}} \sum_{j=1}^{K_n} \sum_\alpha |g_{ij}^{n+1}(\lambda_\alpha)|^2 |\langle \phi_\alpha, \left[ P_{n+1}(f_j^n) - P_{n+1}(h_j^n) \right] \rangle_{\ell^2(G_{n+1})}|^2$$

$$\leqslant \sum_\alpha \left( \sum_{i,j} |g_{ij}(\lambda_\alpha)|^2 \right) \sum_{j=1}^{K_n} |\langle \phi_\alpha, \left[ P_{n+1}(f_j^n) - P_{n+1}(h_j^n) \right] \rangle_{\ell^2(G_{n+1})}|^2$$

$$\leqslant B_{n+1} R_{n+1} \|\|f^n - h_j^n\|_{\mathscr{L}_n}^2.$$

Here we applied Cauchy Schwarz once more in the second to last step and bounded

$$\left( \sum_{i,j} |g_{ij}(\lambda_\alpha)|^2 \right) \leqslant \left( \sup_{\lambda \in \sigma(T)} \sum_{i,j} |g_{ij}(\lambda)|^2 \right).$$

$\square$

To see that these bounds are not necessarily tight, we may simply note that if we have a simple one-layer Network as depicted in Fig. 9 below, the stability can be tightened to

$$\|\Phi_N(f) - \Phi_N(h)\|_{\mathscr{L}_{out}} \leqslant LRB \cdot \|f - h\|_{\mathscr{L}_{in}}$$

with with $B_n := \max_{i=a,b}(\sup_{\lambda \in \sigma(T)} |g_i(\lambda)|)$ as opposed to with $B_n := \sqrt{\sup_{\lambda \in \sigma(T)} \sum_{i=a,b} |g_i(\lambda)|^2}$ if $T$ is normal; as an easy calculation shows.

## F    Proof of Lemma 4.2

We want to prove the following:

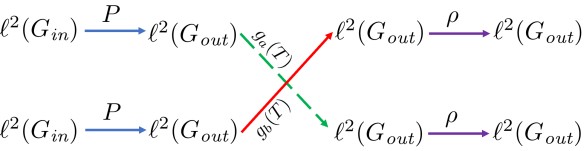

Figure 9: Sparsely connected Layer

**Lemma F.1.** Let $T, \widetilde{T}$ be operators on on $\ell^2(G)$ , $\ell^2(\widetilde{G})$ with $\|T\|_{op}, \|\widetilde{T}\|_{op} \leqslant C$. Let $J : \ell^2(G) \to \ell^2(\widetilde{G})$ be arbitrary but linear. With $K_g = \sum_{k=1}^{\infty} |a_k^g| k C^{k-1}$ for $g$ entire and $K_g = \frac{1}{2\pi} \oint_{\partial D} \frac{1}{z} \gamma_T(z) \gamma_{\widetilde{T}}(z) |g(z)| d|z|$ for $g$ holomorphic, we have

$$\|g(T)J - Jg(\widetilde{T})\|_{op} \leqslant K_g \cdot \|JT - \widetilde{T}J\|_{op}$$

*Proof.* Let us first verify the claim for entire $g$. We first note that

$$\widetilde{T}^k J - JT^k = \widetilde{T}^{k-1}(\widetilde{T}J - JT) + (\widetilde{T}^{k-1}J - JT^{k-1})T$$
$$= \widetilde{T}^{k-1}(\widetilde{T}J - JT) + \widetilde{T}^{k-2}(\widetilde{T}J - JT)T + (\widetilde{T}^{k-2}J - JT^{k-2})T^2.$$

Thus, with $\|T\|_{op}, \|\widetilde{T}\|_{op} \leqslant C$ we find

$$\|\widetilde{T}^k J - JT^k\|_{op} \leqslant kC^{k-1}\|\widetilde{T}J - JT\|_{op}.$$

The claim now follows from applying the triangle inequality.
Now let us prove the bound for holomorphic $g$. We first note the following:

$$\frac{1}{\widetilde{T} - z}(\widetilde{T}J - JT)\frac{1}{T - z}$$
$$= \frac{1}{\widetilde{T} - z}\widetilde{T}J\frac{1}{T - z} - \frac{1}{\widetilde{T} - z}JT\frac{1}{T - z}$$
$$= \left[\frac{1}{\widetilde{T} - z}(\widetilde{T} - z)J + \frac{z}{\widetilde{T} - z}\right]\frac{1}{T - z} - \frac{1}{\widetilde{T} - z}\left[\frac{1}{T - z}(T - z)J + \frac{z}{T - z}\right]$$
$$= z\left(J\frac{1}{T - z} - \frac{1}{\widetilde{T} - z}J\right).$$

Thus we have

$$\|g(\widetilde{T})J - Jg(T)\|_{op} \leqslant \frac{1}{2\pi}\oint_{\partial D}\frac{1}{|z|}\|R_z(T)\|_{op}\|R_z(\widetilde{T})\|_{op}|g(z)|d|z| \leqslant \frac{1}{2\pi}\oint_{\partial D}\frac{1}{|z|}\gamma_T(z)\gamma_{\widetilde{T}}(z)|g(z)|d|z|.$$

□

# G  PROOF OF THEOREM 4.3

We prove the following generalization of Theorem 4.3:

**Theorem G.1.** Let $\Phi_N, \widetilde{\Phi}_N$ be the maps associated to $N$-layer graph convolutional networks with the same non-linearities and functional calculus filters, but based on different graph signal spaces $\ell^2(G), \ell^2(\widetilde{G})$, characteristic operators $T_n, \widetilde{T}_n$ and connecting operators $P_n, \widetilde{P}_n$. Assume $B_n, \widetilde{B}_n \leqslant B$ as well as $R_n, \widetilde{R}_n \leqslant R$ and $L_n \leqslant L$ for some $B, R, L > 0$ and all $n \geqslant 0$. Assume that there are identification operators $J_n : \ell^2(G_n) \to \ell^2(\widetilde{G}_n)$ $(0 \leqslant n \leqslant N)$ almost commuting with non-linearities and connecting operators in the sense of $\|\widetilde{P}_n J_{n-1} f - J_n P_n f\|_{\ell^2(\widetilde{G}_n)} \leqslant \delta_2 \|f\|_{\ell^2(G_n)}$ and $\|\rho_n(J_n f) - J_n \rho_n(f)\|_{\ell^2(\widetilde{G}_n)} \leqslant \delta_1 \|f\|_{\ell^2(G_n)}$. Depending on whether normal or arbitrary characteristic operators are used, define $D_n^2 := \sum_{j \in K_{n-1}} \sum_{i \in K_n} D_{g_{ij}^n}^2$ or $D_n^2 := \sum_{j \in K_{n-1}} \sum_{i \in K_n} K_{g_{ij}^n}^2$. Choose $D$ such that $D_n \leqslant D$ for all $n$. Finally assume that $\|J_n T_n - \widetilde{T}_n J_n\|_* \leqslant \delta$ and with $* = F$ if both

operators are normal and $* = op$ otherwise. Then we have for all $f \in \mathscr{L}_{\text{in}}$ and with $\mathscr{J}_N$ the operator that the $K_N$ copies of $J_N$ induced through concatenation that

$$\|\widetilde{\Phi}(J_0 f) - \mathscr{J}_N \Phi(f)\|_{\widetilde{\mathscr{Z}}_{\text{out}}} \leqslant N \cdot [RLD\delta + \delta_1 BR + \delta_2 BL] \cdot (BRL)^{N-1} \cdot \|f\|_{\mathscr{L}_{\text{in}}}.$$

*Proof.* For simplicity in notation, let us denote the hidden representation of $J_0 f$ in $\widetilde{\mathscr{L}}_n$ by $\widetilde{f}^n$. We then note the following

$$\|\mathscr{J}_{n+1} f^{n+1} - \widetilde{f}^{n+1}\|_{\widetilde{\mathscr{Z}}_{n+1}}$$

$$= \left( \sum_{i=1}^{K_{n+1}} \left\| J_{n+1}\rho_{n+1}\left( \sum_{j=1}^{K_n} g_{ij}^{n+1}(T_{n+1})P_{n+1}(f_j^n) \right) - \rho_{n+1}\left( \sum_{j=1}^{K_n} g_{ij}^{n+1}(T_{n+1})\widetilde{P}_{n+1}(\widetilde{f}_j^n) \right) \right\|_{\ell^2(G_{n+1})}^2 \right)^{\frac{1}{2}}$$

$$\leqslant \left( \sum_{i=1}^{K_{n+1}} \left\| J_{n+1}\rho_{n+1}\left( \sum_{j=1}^{K_n} g_{ij}^{n+1}(T_{n+1})P_{n+1}(f_j^n) \right) - \rho_{n+1}\left( J_{n+1} \sum_{j=1}^{K_n} g_{ij}^{n+1}(T_{n+1})P_{n+1}(f_j^n) \right) \right\|_{\ell^2(G_{n+1})}^2 \right)^{\frac{1}{2}}$$

$$+ L \left( \sum_{i=1}^{K_{n+1}} \left\| J_{n+1} \sum_{j=1}^{K_n} g_{ij}^{n+1}(T_{n+1})P_{n+1}(f_j^n) - \sum_{j=1}^{K_n} g_{ij}^{n+1}(T_{n+1})\widetilde{P}_{n+1}(\widetilde{f}_j^n) \right\|_{\ell^2(G_{n+1})}^2 \right)^{\frac{1}{2}}$$

We can bound the first term by $\delta_1 B \cdot R \cdot (BRL)^n \cdot \|f\|_{\mathscr{L}_{\text{in}}}$. For the second term we find

$$L \left( \sum_{i=1}^{K_{n+1}} \left\| J_{n+1} \sum_{j=1}^{K_n} g_{ij}^{n+1}(T_{n+1})P_{n+1}(f_j^n) - \sum_{j=1}^{K_n} g_{ij}^{n+1}(T_{n+1})\widetilde{P}_{n+1}(\widetilde{f}_j^n) \right\|_{\ell^2(G_{n+1})}^2 \right)^{\frac{1}{2}}$$

$$\leqslant L \left( \sum_{i=1}^{K_{n+1}} \left\| \sum_{j=1}^{K_n} (J_{n+1}g_{ij}^{n+1}(T_{n+1}) - g_{ij}^{n+1}(\widetilde{T}_{n+1})J_{n+1})P_{n+1}(f_j^n) \right\|_{\ell^2(G_{n+1})}^2 \right)^{\frac{1}{2}}$$

$$+ LB \left( \sum_{j=1}^{K_n} \left\| J_{n+1}P_{n+1}(f_j^n) - \widetilde{P}_{n+1}(\widetilde{f}_j^n) \right\|_{\ell^2(G_{n+1})}^2 \right)^{\frac{1}{2}}$$

Arguing as in the proof of 3.1 we can bound the first term by $LD \cdot \delta R \cdot (BRL)^n \|f\|_{\mathscr{L}_{\text{in}}}$. For the second term we find,

$$LB \left( \sum_{j=1}^{K_n} \left\| J_{n+1}P_{n+1}(f_j^n) - \widetilde{P}_{n+1}(\widetilde{f}_j^n) \right\|_{\ell^2(G_{n+1})}^2 \right)^{\frac{1}{2}}$$

$$\leqslant LB\delta_2(BRL)^n + \|\mathscr{J}_n f^n - \widetilde{f}^n\|_{\widetilde{\mathscr{Z}}_n}$$

arguing as above. Iterating from $n = N$ to $n = 0$ then yields the claim. $\qquad\square$

## H    TRANSFERABILITY: GENERAL CONSIDERATIONS

We first prove the statement made at the beginning of Section 5 that

$$\|(\omega Id - \Delta_{\delta_b})^{-1} - (\omega Id - \Delta_{\delta_a})^{-1}\|_{op} = \mathcal{O}(\delta_a \cdot \delta_b).$$

To this end denote the increasing sequence of eigenvalues (counted without multiplicity) of $\Delta_1$ by $\{\lambda_i\}_{i=0}^M$. Recall that $\lambda_0 = 0$ Denote the sequence of projections on the corresponding eigenspaces by $\{P_i\}_{i=0}^M$. We have for the resolvent that

$$\frac{1}{\omega Id - \Delta_\delta} = \frac{1}{\omega Id - \delta \cdot \Delta_1} = \sum_{i=0}^M \frac{1}{\omega - \frac{1}{\delta}\lambda_i} P_i.$$

Thus we have for $\delta_a, \delta_b$ small enough that

$$\left\|\frac{1}{\omega Id - \Delta_{\delta_a}} - \frac{1}{\omega Id - \Delta_{\delta_b}}\right\|_{op} = \left|\frac{1}{\omega - \frac{1}{\delta_a}\lambda_1} - \frac{1}{\omega - \frac{1}{\delta_b}\lambda_1}\right| = \left|\lambda_1 \frac{\frac{1}{\delta_a} - \frac{1}{\delta_b}}{(\omega - \frac{1}{\delta_a}\lambda_1)(\omega - \frac{1}{\delta_b}\lambda_1)}\right|$$

$$= \lambda_1 \frac{1}{|(\omega - \frac{1}{\delta_a}\lambda_1)(\omega - \frac{1}{\delta_b}\lambda_1)|} = \mathcal{O}(\delta_a \cdot \delta_b).$$

Next we note the convergence $(\omega Id - \Delta_\delta)^{-1} \to P_0 \cdot (\omega - 0)^{-1}$. But this is obvious, since for $\lambda_i \neq 0$ we have

$$\frac{1}{\omega - \frac{\lambda_i}{\delta}} \to 0$$

as $\delta \to 0$.

# I PROOFS OF LEMMA 5.3 AND THEOREM 5.4

**Lemma I.1.** Let $T$ and $\widetilde{T}$ be characteristic operators on $\ell^2(G)$ and $\ell^2(\widetilde{G})$ be respectively. If these operators are $\omega$-$\delta$-close with identification operator $J$, and $\|R_\omega\|_{op}, \overline{R}_\omega\|_{op} \leqslant C$ we have

$$\|Jg(T) - g(\widetilde{T})J\|_{op} \leqslant K_g \cdot \|(\widetilde{R}_\omega J - JR_\omega)\|_{op}$$

with $K_g = \oint_{\partial D}(1 + |z - \omega|\gamma_T(z))(1 + |z - \omega|\gamma_{\widetilde{T}}(z))|g(z)|d|z|$ if $g$ is holomorphic and $K_g = \|g\|_{\mathscr{F}_{\omega,C}^{hol}}$ if $g \in \mathscr{F}_{\omega,C}^{hol}$. If $T$ and $\widetilde{T}$ are normal as well as doubly $\omega$-$\delta$-close and $g \in \mathscr{F}_{\omega,C}^{cont}$, we have $K_g = \|g\|_{\mathscr{F}_{\omega,C}^{cont}}$.

*Proof.* We first deal with the statement concerning holomorphic $g$. To this end we note that Lemma 4.5.9 of Post (2012) proves

$$\|\widetilde{R}_z J - JR_z\|_{op} \leqslant (1 + |z - \omega|\gamma_T(z))(1 + |z - \omega|\gamma_{\widetilde{T}}(z)) \cdot \|\widetilde{R}_\omega J - JR_\omega\|_{op}.$$

The claim then follows from

$$\|Jg(T) - g(\widetilde{T})J\|_{op} \leqslant \frac{1}{2\pi} \oint_{\partial D} |g(z)|\|\widetilde{R}_z J - JR_z\|_{op}d|z|.$$

For $g \in \mathscr{F}_{\omega,C}^{hol}$ the claim is proved exactly as in the proof of Lemma 2.3.
For $g \in \mathscr{F}_{\omega,C}^{cont}$ we note that

$$(\widetilde{R}_\omega)^\mu(\widetilde{R}_\omega^*)^\nu J - J(R_\omega)^\mu(R_\omega^*)^\nu = (\widetilde{R}_\omega)^\mu\left[(\widetilde{R}_\omega^*)^\nu J - J(R_\omega^*)^\nu\right] + \left[(\widetilde{R}_\omega)^\mu J - J(R_\omega)^\mu\right](R_\omega^*)^\nu.$$

Together with the result

$$\|\widetilde{T}^k J - JT^k\|_{op} \leqslant kC^{k-1}\|\widetilde{T}J - JT\|_{op}.$$

established in the proof of Lemma 4.2, the claim then follows from the triangle inequality together with the definition of the semi-norm $\|g\|_{\mathscr{F}_{\omega,C}^{cont}}$.

$\square$

As in the previous section, we state a slightly more general version of our main theorem of this section:

**Theorem I.2.** Let $\Phi, \widetilde{\Phi}$ be the maps associated to $N$-layer graph convolutional networks with the same non-linearities and functional calculus filters, but based on different graph signal spaces $\ell^2(G_n), \ell^2(\widetilde{G}_n)$, characteristic operators $T_n, \widetilde{T}_n$ and connecting operators $P_n, \widetilde{P}_n$. Assume $B_n, \widetilde{B}_n \leqslant B$ as well as $R_n, \widetilde{R}_n \leqslant R$ and $L_n \leqslant L$ for some $B, R, L > 0$ and all $n \geqslant 0$. Assume that there are identification operators $J_n : \ell^2(G_n) \to \ell^2(\widetilde{G}_n)$ $(0 \leqslant n \leqslant N)$ almost commuting with non-linearities and connecting operators in the sense of $\|\widetilde{P}_n J_{n-1}f - J_n P_n f\|_{\ell^2(\widetilde{G}_n)} \leqslant \delta_2\|f\|_{\ell^2(G_n)}$ and $\|\rho_n(J_n f) - J_n\rho_n(f)\|_{\ell^2(\widetilde{G}_n)}\delta_1\|f\|_{\ell^2(G_n)}$. define $D_n^2 := \sum_{j\in K_{n-1}}\sum_{i\in K_n} K_{g_{ij}^n}^2$ with $K_{g_{ij}^n}$ as in

Lemma 5.3. Choose $D$ such that $D_n \leqslant D$ for all $n$. Finally assume that $\|J_n(\omega Id - T_n)^{-1} - (\omega Id - \widetilde{T}_n)^{-1}J_n\|_{op} \leqslant \delta$. If filters in $\mathscr{F}^{cont}_{\omega,C}$ are used, assume additionally that $\|J_n((\omega Id - T_n)^{-1})^* - ((\omega Id - \widetilde{T}_n)^{-1})^* J_n\|_{op} \leqslant \delta$. Then we have for all $f \in \mathscr{L}_{\text{in}}$ and with $\mathscr{J}_N$ the operator that the $K_N$ copies of $J_N$ induced through concatenation that

$$\|\widetilde{\Phi}(J_0 f) - \mathscr{J}_N \Phi(f)\|_{\widetilde{\mathscr{L}}_{\text{out}}} \leqslant N \cdot [RLD\delta + \delta_1 BR + \delta_2 BL] \cdot (BRL)^{N-1} \cdot \|f\|_{\mathscr{L}_{\text{in}}}.$$

*Proof.* The proof proceeds in complete analogy to the one of Theorem 4.3. ◻

## J COLLAPSING STRONG EDGES: PROOFS AND FURTHER DETAILS

We utilize the notation introduced in Section 5.2. Beyond this, we denote the positive semi-definite form induced by the energy functional $E_{\widetilde{G}}$ by

$$E_{\widetilde{G}}(u,v) := \langle u, \Delta_G v\rangle_{\ell^2(\widetilde{G})}.$$

We further use the notation $E_{\widetilde{G}}(u) := E_{\widetilde{G}}(u,u)$. With

$$\begin{aligned}
E_{\widetilde{G}} = &\sum_{\substack{\alpha \in \widetilde{G}_{Greek} \\ \beta \in \widetilde{G}_{Greek}}} \widetilde{W}_{\alpha\beta}|u(\alpha) - u(\beta)|^2 \\
&+ \sum_{\substack{a \in \widetilde{G}_{Latin} \\ b \in \widetilde{G}_{Latin}}} \widetilde{W}_{ab}|u(a) - u(b)|^2 \\
&+ \sum_{\substack{a \in \widetilde{G}_{Latin} \\ \beta \in \widetilde{G}_{Greek}}} \widetilde{W}_{a\beta}|u(a) - u(\beta)|^2 \\
&+ \sum_{\substack{\alpha \in \widetilde{G}_{Greek} \\ b \in \widetilde{G}_{Latin}}} \widetilde{W}_{\alpha b}|u(\alpha) - u(b)|^2 \\
&+ \sum_{\alpha \in \widetilde{G}_{Greek}} \widetilde{W}_{\alpha\star}|u(\alpha) - u(\star)|^2 \\
&+ \sum_{\beta \in \widetilde{G}_{Greek}} \widetilde{W}_{\star\beta}|u(\star) - u(\beta)|^2 \\
&+ \sum_{a \in \widetilde{G}_{Latin}} \widetilde{W}_{a\star}|u(a) - u(\star)|^2 \\
&+ \sum_{b \in \widetilde{G}_{Latin}} \widetilde{W}_{\star b}|u(\star) - u(b)|^2
\end{aligned} \tag{9}$$

Similar considerations apply when $\widetilde{G}$ is replaced by $G$.

Let us next solve the convex optimization program (5) introduced in Definition 5.5, restated here for convenience:

**Definition J.1.** For each $g \in G$, define the signal $\psi_g^\delta \in \ell^2(\widetilde{G})$ as the unique solution to the convex optimization program

$$\min E_{\widetilde{G}}(u) \quad \textit{subject to } u(h) = \delta_{hg} \textit{ for all } h \in \widetilde{G}_{Latin} \bigcup \{\star\}.$$

As a first step we note that all entries of $\psi_g$ are real and non-negative, which follows since each summand in (9) is non-increasing under the map $u \mapsto |u|$ due to the reverse triangle $||a|-|b|| \leqslant |a-b|$.

To find the explicit form of $\psi_g$, fix $g \in \widetilde{G}_{Latin} \bigcup \{\star\}$ and denote by $\chi_g \in \ell^2(\widetilde{G})$ the signal defined by setting it to $\chi^\eta(h) = \delta_{hg}$ for $h \in \widetilde{G}_{Latin} \bigcup \{\star\}$ and $\eta_g(\alpha) = \eta_g^\alpha$ with $\{\eta_g^\alpha\}_{\alpha \in \widetilde{G}_{Greek}}$ a set of $|\widetilde{G}_{Greek}|$

free parameters in $\mathbb{R}_{\leqslant 0}$. We then have

$$E_{\widetilde{G}}(\chi_g) = 2 \sum_{a \in \widetilde{G}_{Latin}} \widetilde{W}_{ag} + 2 \sum_{\alpha \in \widetilde{G}_{Greek}} \widetilde{W}_{\alpha g} |1 - \eta_g^\alpha|^2 + 2 \sum_{\substack{\alpha \in \widetilde{G}_{Greek} \\ b \in \widetilde{G}_{Latin} \bigcup \{\star\}}} \widetilde{W}_{\alpha b} |\eta_g^\alpha|^2$$
$$+ \sum_{\alpha, \beta \in \widetilde{G}_{Greek}} \widetilde{W}_{\alpha\beta} |\eta_g^\alpha - \eta_g^\beta|^2.$$

By definition, $\chi_g$ depends smoothly on the parameters $\{\eta_g^\alpha\}_{\alpha \in \widetilde{G}_{Greek}}$. Finding the minimizer of the convex optimization program (5) is then equivalent to finding the values $\{\eta_g^\alpha\}_{\alpha \in \widetilde{G}_{Greek}}$ at which we have

$$\frac{\partial E_{\widetilde{G}}(\chi_g)}{\partial \eta_g^\alpha} = 0.$$

We note

$$\frac{1}{4} \frac{\partial E_{\widetilde{G}}(\chi_g)}{\partial \eta_g^\xi} = \left( \widetilde{W}_{g\xi} + \sum_{\substack{a \in \widetilde{G}_{Latin} \bigcup \{\star\} \\ a \neq g}} \widetilde{W}_{g\xi} + \sum_{\alpha \in \widetilde{G}_{Greek}} \widetilde{W}_{\alpha g} \right) \eta_\xi^g - \sum_{\alpha \in \widetilde{G}_{Greek}} \widetilde{W}_{\alpha g} \eta_\alpha^g - \widetilde{W}_{g\xi}$$

Collecting these equations for all parameters into a matrix equation, we find that the 'Greek entries' of the vector $\psi_g$ are given explicitly by

$$\begin{pmatrix} \psi_g(\alpha) \\ \psi_g(\beta) \\ \vdots \end{pmatrix} = \begin{pmatrix} \tilde{d}_\alpha & -\widetilde{W}_{\alpha\beta} & \cdots \\ -\widetilde{W}_{\beta\alpha} & \tilde{d}_\beta & \vdots \\ \vdots & \cdots & \ddots \end{pmatrix}^{-1} \cdot \begin{pmatrix} \widetilde{W}_{g\alpha} \\ \widetilde{W}_{g\beta} \\ \vdots \end{pmatrix}, \tag{10}$$

with degrees in $\widetilde{G}$ denoted by $\tilde{d}_\alpha$. Let us denote the restriction of $\psi_g^\delta$ to Greek entries, thought of as a vector in $\mathbb{C}^{|\widetilde{G}_{Greek}|}$ by $\vec{\eta}_g^\delta$.

Given the degree $\tilde{d}_\alpha$ corresponding to a Greek index, we decompose it as

$$\tilde{d}_\alpha = \tilde{d}_\alpha^r + \widetilde{W}_{\alpha\star} + V_\alpha$$

with $\tilde{d}_\alpha^r$ accounting for edges from $\alpha$ to other greek vertices

$$\tilde{d}_\alpha^r = \sum_{\beta \in \widetilde{G}_{Greek}} \widetilde{W}_{\alpha\beta} = \frac{1}{\delta} \sum_{\beta \in \widetilde{G}_{Greek}} \omega_{\alpha\beta},$$

and $V_\alpha$ accounting for edges from $\alpha$ to Latin vertices

$$V_\alpha = \sum_{a \in \widetilde{G}_{Latin}} \widetilde{W}_{a\alpha}.$$

Recall that we also may write

$$\widetilde{W}_{\alpha\star} = \frac{1}{\delta} \omega_{\alpha\star}.$$

We may then write

$$\begin{pmatrix} \tilde{d}_\alpha & -\widetilde{W}_{\alpha\beta} & \cdots \\ -\widetilde{W}_{\beta\alpha} & \tilde{d}_\beta & \vdots \\ \vdots & \cdots & \ddots \end{pmatrix} = \begin{pmatrix} \tilde{d}_\alpha^r & -\widetilde{W}_{\alpha\beta} & \cdots \\ -\widetilde{W}_{\beta\alpha} & \tilde{d}_\beta^r & \vdots \\ \vdots & \cdots & \ddots \end{pmatrix} + \frac{1}{\delta} \begin{pmatrix} \omega_{\alpha\star} & 0 & \cdots \\ 0 & \omega_{\beta\star} & \vdots \\ \vdots & \cdots & \ddots \end{pmatrix} + \begin{pmatrix} V_\alpha & 0 & \cdots \\ 0 & V_\beta & \vdots \\ \vdots & \cdots & \ddots \end{pmatrix}$$
$$=: \frac{1}{\delta} \mathscr{L} + \frac{1}{\delta} diag(\vec{\omega}_\star) + V,$$

where we made the obvious definitions for the matrices $\mathscr{L}$ and $V$ and denoted by $\vec{\omega}_\star$ the vector with entries $\omega_{\alpha\star}$. Let us also use the notation

$$h := \mathscr{L} + diag(\omega_\star).$$

Next we want to establish that $h$ is invertible. For this we first note that that $\mathscr{L}$ is the graph Laplacian of the subgraph $\widetilde{G}_{Greek}$; which we assume to be connected. Hence $\mathscr{L}$ is positive semi-definite with the eigenspace corresponding to the eigenvalue zero being spanned by (entry-wise) constant vectors. Since all entries of $\omega_\star$ are non-negative, the operator $h$ is also positive semi-definite. Since we assume that the vertex $\star$ is connected to at least one other vertex in $\widetilde{G}_{Greek}$, there is at least one entry in $\vec{\omega}_\star$ that is strictly greater than zero. We show that this already implies that $h$ is in fact also positive definite and hence invertible. Indeed, for any $\vec{v} \in \mathbb{C}^{|\widetilde{G}_{Greek}|}$ we have

$$\langle \vec{v}, \mathscr{L} \cdot \vec{v} \rangle_{\mathbb{C}^{|\widetilde{G}_{Greek}|}} = \langle \vec{v}, h \cdot \vec{v} \rangle_{\mathbb{C}^{|\widetilde{G}_{Greek}|}} + \langle \vec{v}, diag(\vec{\omega}_\star) \cdot \vec{v} \rangle_{\mathbb{C}^{|\widetilde{G}_{Greek}|}}.$$

Both terms on the right hand side are non-negative. If $\vec{v}$ is a constant (non-zero) vector, the first term vanishes, but since at least one entry of $\omega_\star$ is strictly positive, with all others being non-negative, the second term on the right hand side is strictly positive. If $\vec{v}$ is non-constant, the first term on the right hand side is larger than zero. Hence $h$ is positive definite and thus invertible. Similarly one proves that (for any $\delta \geqslant 0$) the operator $h + \delta V$ is positive definite and hence invertible. Thus we now know that the operator

$$\frac{1}{\delta}(h + \delta V) = \begin{pmatrix} \widetilde{d}_\alpha & -\widetilde{W}_{\alpha\beta} & \cdots \\ -\widetilde{W}_{\beta\alpha} & \widetilde{d}_\beta & \vdots \\ \vdots & \cdots & \ddots \end{pmatrix}$$

utilized in (10) is indeed invertible. We note (again with the restriction of $\psi_g^\delta$ to Greek entries thought of as a vector in $\mathbb{C}^{|\widetilde{G}_{Greek}|}$ denoted by $\vec{\eta}_g^\delta$) that we may equivalently write (10) as

$$(h + \delta V)^{-1} \vec{\eta}_g^\delta = \delta \vec{\widetilde{W}}_g \tag{11}$$

and

$$\vec{\widetilde{W}}_g := \begin{pmatrix} \widetilde{W}_{g\alpha} \\ \widetilde{W}_{g\beta} \\ \vdots \end{pmatrix}$$

thought of as an element of $\mathbb{C}^{|\widetilde{G}_{Greek}|}$. To proceed, we now first focus on the case $g = \star$, for which we may write (11) equivalently as

$$(h + \delta V)^{-1} \vec{\eta}_\star^\delta = \vec{\omega}_\star. \tag{12}$$

Since $\vec{\omega}_\star$ is independent of $\delta$, we may take the limit $\delta \to 0$ and arrive at

$$(\mathscr{L} + diag(\vec{\omega}_\star)) \vec{\eta}_\star^0 = \vec{\omega}_\star$$

which is uniquely solved by $\vec{\eta}_\star^0 = (1, 1, 1, ....) \equiv \mathbb{1}_{Greek}$.
Since we assume $\delta \ll 1$, we can now investigate the solution $\vec{\eta}_g^\delta$ for non-zero $\delta$ through perturbation theory. We write

$$\vec{\eta}_\star^\delta = \mathbb{1}_{\widetilde{G}_{Greek}} - \vec{\zeta}_\star^\delta$$

with $\vec{\zeta}_\star^0 = 0$ and find from (12) – using $h \cdot \mathbb{1}_{Greek} = \vec{\eta}_\star^\delta$ – the defining equation

$$\vec{\zeta}_\star^\delta = \delta(h + \delta V)^{-1} \cdot V \cdot \mathbb{1}_{\widetilde{G}_{Greek}}.$$

From this we obtain the estimate

$$\|\vec{\zeta}_\star^\delta\|_{\ell^2(\widetilde{G}_{Greek})} \leqslant \|(h + \delta V)\|_{op} \cdot \|V \cdot \mathbb{1}_{\widetilde{G}_{Greek}}\|_{\ell^2(\widetilde{G}_{Greek})} \cdot \delta,$$

where we denote by $\ell^2(\widetilde{G}_{Greek})$ the space graph signal space $\mathbb{C}^{|\widetilde{G}_{Greek}|}$ equipped with node weights $\{\widetilde{\mu}_g\}_{g \in \widetilde{G}_{Greek}}$.

We note that both $h$ and $V$ are positive semi-definite and we thus obtain

$$\lambda_{\min}(h) \leqslant \lambda_{\min}(h + \delta V)$$

for the minimal eigenvalues of the respective operators. Hence

$$\|(h + \delta V)^{-1}\|_{op} \leqslant \|h^{-1}\|_{op},$$

and thus also

$$\|\vec{\zeta}_\star^\delta\|_{\ell^2(\widetilde{G}_{Greek})} \leqslant \underbrace{\|h^{-1}\|_{op} \cdot \|V \cdot \mathbb{1}_{\widetilde{G}_{Greek}}\|_{\ell^2(\widetilde{G}_{Greek})}}_{=:K} \cdot \delta. \tag{13}$$

Since $\|h^{-1}\|_{op} = 1/\lambda_{\min}(h)$ we may write

$$K = \frac{\|V \cdot \mathbb{1}_{Greek}\|_{\ell^2(\widetilde{G}_{Greek})}}{\lambda_{\min}(h)}. \tag{14}$$

From (11) we know that for $g \neq \star$ we have $\vec{\eta}_g^\delta = 0$.
We now also want to bound $\|\vec{\eta}_g^\delta\|_{\ell^2(\widetilde{G}_{Greek})}$ in terms of $\delta$. We will do this by establishing the relationship

$$\sum_{g \in \widetilde{G}_{Latin}} \vec{\eta}_g^\delta = \vec{\zeta}_\star^\delta. \tag{15}$$

and then utilizing our estimate on $\|\vec{\zeta}_\star^\delta\|_{\ell^2(\widetilde{G}_{Greek})}$ established above. To prove (15), we will need the concept of **harmonic extensions**:

**Definition J.2.** Denote by $\ell^2(\widetilde{G}_{Latin} \cup \{\star\})$ the graph signal space $\mathbb{C}^{|\widetilde{G}_{Latin} \cup \{\star\}|}$ equipped with the node weights $\{\widetilde{\mu}_g\}_{g \in \widetilde{G}_{Latin} \cup \{\star\}}$. Given an arbitrary signal $\overline{u} \in \ell^2(\widetilde{G}_{Latin} \cup \{\star\})$ a **harmonic extension** of $\overline{u}$ to all of $\ell^2(\widetilde{G})$ is a signal $u \in \ell^2(\widetilde{G})$ satisfying

$$(\Delta_{\widetilde{G}} u)(\alpha) = 0 \ \ \forall \alpha \in \widetilde{G}_{Greek} \ \ and \ \ u(h) = \overline{u}(h) \ \ \forall h \in \widetilde{G}_{Latin} \bigcup \{\star\}.$$

We first note that the concept of harmonic extensions is both well-defined an well-behaved:

**Lemma J.3.** Fix $\overline{u} \in \ell^2(\widetilde{G}_{Latin} \cup \{\star\})$. There exists a unique harmonic extension $u \in \ell^2(\widetilde{G})$ of $\overline{u}$. It is given as the solution to the convex optimization program

$$\min E_{\widetilde{G}}(u) \ \ subject \ to \ \ u(h) = \delta_{hg} \ for \ all \ h \in \widetilde{G}_{Latin} \bigcup \{\star\}.$$

Furthermore if $u$ and $v$ are the harmonic extensions of $\overline{u}$ and $\overline{v}$, then $(u + v)$ is the (unique) harmonic extension of $(\overline{u} + \overline{v})$.

*Proof.* We write a signal $\psi \in \ell^2(\widetilde{G})$ as $\psi = (\overline{\psi}, \eta)$ with $\overline{\psi} \in \ell^2(\widetilde{G}_{Latin} \cup \{\star\})$ and $\eta \in \ell^2(\widetilde{G}_{Greek})$. We then notice

$$\psi = argmin E_{\widetilde{G}}(u) \ \ subject \ to \ \ \psi(h) = \overline{\psi}(h) \ for \ all \ h \in \widetilde{G}_{Latin} \bigcup \{\star\}$$

$$\Leftrightarrow \frac{\partial E_{\widetilde{G}}(\psi)}{\partial \eta_\alpha} = 0 \ \ \forall \alpha \in \widetilde{G}_{Greek} \ \ and \ \ \psi(h) = \overline{\psi}(h) \ for \ all \ h \in \widetilde{G}_{Latin} \bigcup \{\star\}$$

$$\Leftrightarrow \sum_{y \in \widetilde{G}} \widetilde{W}_{\alpha y}(\psi(\alpha) - \psi(y)) = 0 \ \ \forall \alpha \in \widetilde{G}_{Greek} \ \ and \ \ \psi(h) = \overline{\psi}(h) \ for \ all \ h \in \widetilde{G}_{Latin} \bigcup \{\star\}$$

$$\Leftrightarrow (\Delta_{\widetilde{G}} \psi)(\alpha) = 0 \ \forall \alpha \in \widetilde{G}_{Greek} \ \ and \ \ \psi(h) = \overline{\psi}(h) \ for \ all \ h \in \widetilde{G}_{Latin} \bigcup \{\star\}.$$

Here, we treated $\eta_\alpha$ and its complex conjugate as independent variables and used that $E_{\widetilde{G}}(\cdot)$ is a real-valued functional for the first equivalence. As harmonic extensions are thus equivalently characterised as the solutions of convex minimization programs, they are unique.
To prove the last statement, we note that by linearity of the graph Laplacian, $(u + v)$ certainly is a harmonic extension of $(\overline{u} + \overline{v})$. Since harmonic extensions are unique, it is the only one. $\square$

After this preparatory effort, we are now ready to prove (15):

**Lemma J.4.** For any $\delta \geq 0$ the signals $\{\vec{\eta}_g^\delta\}_{g \in \widetilde{G}_{Latin} \bigcup \{\star\}}$ form a partition of unity of $\ell^2(\widetilde{G}_{Greek})$:

$$\sum_{g \in \widetilde{G}_{Latin} \bigcup \{\star\}} \vec{\eta}_g^\delta = \mathbb{1}_{\widetilde{G}_{Greek}} \tag{16}$$

Equivalently we have

$$\sum_{g \in \widetilde{G}_{Latin}} \vec{\eta}_g^\delta = \vec{\zeta}_\star^\delta.$$

As an immediate Corollary we obtain

**Corollary J.5.** For any $\delta \geq 0$ the signals $\{\psi_g^\delta\}_{g \in \widetilde{G}_{Latin} \bigcup \{\star\}}$ form a partition of unity of $\ell^2(\widetilde{G})$:

$$\sum_{g \in \widetilde{G}_{Latin} \bigcup \{\star\}} \vec{\eta}_g^\delta = \mathbb{1}_{\widetilde{G}}. \tag{17}$$

*Proof.* Using the 'boundary conditions' in (5), it is straightforward to verify that (16) is equivalent to (17). From Lemma J.3 we now know that $\psi_g^\delta$, originally characterised as the solution of the problem

$$\min E_{\widetilde{G}}(u) \quad subject\ to\ \ u(h) = \delta_{hg}\ for\ all\ h \in \widetilde{G}_{Latin} \bigcup \{\star\},$$

is equivalently characterised as the harmonic extension of $u(h) = \delta_{hg}$. From the last statement of Lemma J.3, we know that $\sum_{g \in \widetilde{G}_{Latin} \bigcup \{\star\}} \vec{\eta}_g^\delta$ is the unique harmonic extension of

$$\sum_{g \in \widetilde{G}_{Latin} \bigcup \{\star\}} \delta_{hg} = \mathbb{1}_{\widetilde{G}_{\widetilde{G}_{Latin} \bigcup \{\star\}}}.$$

But this – in turn – is the unique solution of the problem

$$\min E_{\widetilde{G}}(u) \quad subject\ to\ \ u(h) = 1\ for\ all\ h \in \widetilde{G}_{Latin} \bigcup \{\star\}.$$

Since we have

$$E_{\widetilde{G}}(\mathbb{1}_{\widetilde{G}}) = 0,$$

which is the lowest possible attainable value of $E_{\widetilde{G}}(\cdot)$, and setting $u = \mathbb{1}_{\widetilde{G}}$ is compatible with the 'boundary condition' $u(h) = 1$ *for all* $h \in \widetilde{G}_{Latin} \bigcup \{\star\}$, we know that is the (unique) harmonic extension of $\mathbb{1}_{\widetilde{G}_{Latin} \bigcup \{\star\}}$. By the last statement of Lemma J.3 we thus have

$$\sum_{g \in \widetilde{G}_{Latin} \bigcup \{\star\}} \vec{\eta}_g^\delta = \mathbb{1}_{\widetilde{G}}.$$

$\square$

Having established that we may write

$$\sum_{g \in \widetilde{G}_{Latin}} \vec{\eta}_g^\delta = \vec{\zeta}_\star^\delta,$$

together with the fact that every entry of each $\vec{\eta}_g^\delta$ is non-negative, we now know that

$$0 \leq \vec{\eta}_g^\delta(\alpha), \vec{\zeta}_\star^\delta \leq 1.$$

Furthermore – using our earlier estimate (13) – we now easily obtain

$$\left\| \sum_{g \in \widetilde{G}_{Latin}} \vec{\eta}_g^\delta \right\|_{\ell^2(\widetilde{G}_{Greek})} \leq K \cdot \delta.$$

Hence – by positivity of the entries – we also have for each individual $g \in \widetilde{G}_{Latin}$ that

$$\left\| \vec{\eta}_g^\delta \right\|_{\ell^2(\widetilde{G}_{Greek})} \leq K \cdot \delta.$$

For the weights $\{\mu_g^\delta\}_{g \in G}$ we then find

$$\widetilde{\mu}_g \leqslant \mu_g^\delta \leqslant \widetilde{\mu}_g + \delta K \sum_{\alpha \in \widetilde{G}_{Greek}} \widetilde{\mu}_\alpha$$

if $g \neq \star$. We also write $\widetilde{\mu}(\widetilde{G}_{Greek}) := \sum_{\alpha \in \widetilde{G}_{Greek}} \widetilde{\mu}_\alpha$. If $g = \star$, we have

$$\widetilde{\mu}_\star^\delta + (1 - \delta)\widetilde{\mu}(\widetilde{G}_{Greek}) \leqslant \mu_\star^\delta \leqslant \widetilde{\mu}_\star^\delta + \widetilde{\mu}(\widetilde{G}_{Greek}).$$

Having set the scene, we are now ready to prove Theorem 5.4. Following Post & Simmer (2017), instead of checking the conditions of Definition 5.1 and Definition 5.2 it is instead sufficient to check the following, with $J \, \widetilde{J}$ as defined in Section 5.2 to establish Theorem 5.6:

**Lemma J.6.** In addition to identification operators $J, \widetilde{J}$, assume that there exist additional operators $J^1 : \ell^2(G) \to \ell^2(\widetilde{G})$ and $\widetilde{J}^1 : \ell^2(\widetilde{G}) \to \ell^2(G)$ so that the following set of equations is satisfied with $\epsilon = \mathcal{O}(\delta^{\frac{1}{2}})$

$$\|Jf\| \leqslant (1 + \epsilon')\|f\|, \quad |\langle Jf, u \rangle - \langle f, \widetilde{J}u \rangle| \leqslant \epsilon'\|f\| \tag{18}$$

$$\|f - \widetilde{J}Jf\| \leqslant \epsilon'\sqrt{\|f\|^2 + E_G(f)}, \quad \|u - J\widetilde{J}u\| \leqslant \epsilon'\sqrt{\|u\|^2 + E_{\widetilde{G}}(u)} \tag{19}$$

$$\|J^1 f - Jf\| \leqslant \epsilon'\sqrt{\|f\|^2 + E_G(f)}, \quad \|\widetilde{J}u - \widetilde{J}^1 u\| \leqslant \epsilon'\sqrt{\|u\|^2 + E_{\widetilde{G}}(u)} \tag{20}$$

$$\|E_{\widetilde{G}}(J^1 f, u) - E_G(f, \widetilde{J}^1 u)\| \leqslant \epsilon' \cdot \sqrt{\|f\|^2 + E_G(f)} \cdot \sqrt{\|u\|^2 + E_{\widetilde{G}}(u)}. \tag{21}$$

Then the (normal) operators $\Delta$ and $\widetilde{\Delta}$ are (doubly) (-1)- ($\epsilon = 12\epsilon'$) -close with identification-operator $J$.

Here, we always have $u \in \ell^2(\widetilde{G})$ and $f \in \ell^2(G)$)

*Proof.* This follows immediately after combining Proposition 4.4.12 with Theorem 4.4.15 of Post (2012). □

We set $J^1 f = Jf$ and $(\widetilde{J}^1 u)(x) = u(x)$ and now determine the individual $\epsilon = \epsilon(\delta)$ values for which these equations are satisfied:

**Left-hand-side of (18):**
For the left hand side of (18) we note (using $2ab \leqslant a^2 + b^2$ and the fact that the $\psi_g$ form a partition of unity):

$$\begin{aligned}
\|Jf\|_{\ell^2(\widetilde{G})}^2 &= \sum_{h,g \in G} \langle \psi_h^\delta, \psi_g^\delta \rangle_{\ell^2(\widetilde{G})} \overline{f}(h)f(g) \\
&\leqslant \frac{1}{2}\sum_{h \in G} |f(h)|^2 \sum_{g \in G}\langle \psi_h^\delta, \psi_g \rangle_{\ell^2(\widetilde{G})} + \frac{1}{2}\sum_{g \in G} |f(g)|^2 \sum_{h \in G}\langle \psi_h^\delta, \psi_g^\delta \rangle_{\ell^2(\widetilde{G})} \\
&= \frac{1}{2}\sum_{h \in G} |f(h)|^2 \langle \psi_h^\delta, \mathbb{1} \rangle_{\ell^2(\widetilde{G})} + \frac{1}{2}\sum_{g \in G} |f(g)|^2 \langle \mathbb{1}, \psi_g^\delta \rangle_{\ell^2(\widetilde{G})} \\
&= \sum_{g \in G} |f(g)|^2 \mu_g^\delta \\
&= \|f\|_{\ell^2(G)}^2.
\end{aligned}$$

Here the second to last inequality follows from the definition of the weights $\mu_g^\delta$. Thus the left hand side of (18) holds with

$$\epsilon = 0.$$

**Right-hand-side of (18):**
The right hand side of (18) holds trivially with

$$\epsilon = 0$$

since we have chosen $J^* = \widetilde{J}$.

**Left-hand-side of (19):**
Now let us check the l.h.s. of (19). We have:

$$(f - \widetilde{J}Jf)(y) = f(y) - \sum_{g \in G} f(g) \frac{\langle \psi_g^\delta, \psi_y^\delta \rangle_{\ell^2(\widetilde{G})}}{\mu_y^\delta}.$$

Using the constant $K$ defined in (14) we have

$$\widetilde{\mu}_g \leqslant \mu_g^\delta \leqslant \widetilde{\mu}_g + \delta K \sum_{\alpha \in \widetilde{G}_{Greek}} \widetilde{\mu}_\alpha$$

if $g \neq \star$. We also write $\widetilde{\mu}(\widetilde{G}_{Greek}) := \sum_{\alpha \in \widetilde{G}_{Greek}} \widetilde{\mu}_\alpha$. If $G = \star$, we have

$$\widetilde{\mu}_\star + (1 - \delta)\widetilde{\mu}(\widetilde{G}_{Greek}) \leqslant \mu_\star^\delta \leqslant \widetilde{\mu}_\star + 1\widetilde{\mu}(\widetilde{G}_{Greek}).$$

We next note

$$\langle \psi_x^\delta, \psi_y^\delta \rangle_{\ell^2(\widetilde{G})} = \widetilde{\mu}_x \delta_{xy} + \langle \vec{\eta}_x^\delta, \vec{\eta}_y^\delta \rangle_{\ell^2(G_{Greek})}$$

with $\widetilde{W}_x$ the vector with entries $\widetilde{W}_x(g) = \widetilde{W}_{xg}$.

Thus for $y \neq \star$ we find

$$|(f - \widetilde{J}Jf)(y)| \leqslant \left(1 - \frac{\widetilde{\mu}_y}{\mu_y^\delta}\right)|f(y)| + \left|\sum_{\substack{g \in G \\ g \neq y}} f(g) \frac{\langle \psi_g^\delta, \psi_y^\delta \rangle_{\ell^2(\widetilde{G})}}{\mu_y^\delta}\right|.$$

We thus find

$$
\begin{aligned}
\|f - \widetilde{J}Jf\|_{\ell^2(G)} &\leqslant \sqrt{\sum_{\substack{y \in G \\ y \neq \star}} \left(\left(1 - \frac{\widetilde{\mu}_y}{\mu_y^\delta}\right)|f(y)| + \left|\sum_{\substack{g \in G \\ g \neq y}} f(g) \frac{\langle \psi_g^\delta, \psi_y^\delta \rangle_{\ell^2(\widetilde{G})}}{\mu_y^\delta}\right|\right)^2} \\
&\quad + \left|f(\star) - \sum_{g \in G} f(g) \frac{\langle \psi_g^\delta, \psi_\star^\delta \rangle_{\ell^2(\widetilde{G})}}{\mu_\star^\delta}\right| \\
&\leqslant \sqrt{\sum_{\substack{y \in G \\ y \neq \star}} \left(\left(1 - \frac{\widetilde{\mu}_y}{\mu_y^\delta}\right)|f(y)|\right)^2} + \sqrt{\sum_{\substack{y \in G \\ y \neq \star}} \left(\left|\sum_{\substack{g \in G \\ g \neq y}} f(g) \frac{\langle \psi_g^\delta, \psi_y^\delta \rangle_{\ell^2(\widetilde{G})}}{\mu_y^\delta}\right|\right)^2} \\
&\quad + \left|f(\star) - \sum_{g \in G} f(g) \frac{\langle \psi_g^\delta, \psi_\star^\delta \rangle_{\ell^2(\widetilde{G})}}{\mu_\star^\delta}\right|
\end{aligned}
$$

To bound the first term of the estimate, we note (for $y \neq \star$) and $\delta$ small enough:

$$\left(1 - \frac{\widetilde{\mu}_y}{\mu_y^\delta}\right) \leqslant \left(1 - \frac{\widetilde{\mu}_y}{\widetilde{\mu}_y + \delta K \widetilde{\mu}(\widetilde{G}_{Greek})}\right) = \frac{\delta K \widetilde{\mu}(\widetilde{G}_{Greek})}{\delta K \widetilde{\mu}_y + \widetilde{\mu}(\widetilde{G}_{Greek})} \leqslant \delta \frac{K \widetilde{\mu}(\widetilde{G}_{Greek})}{\min_{g \in \widetilde{G}_{Latin}} \widetilde{\mu}_g}.$$

We also note (for $y \neq \star$)

$$|f(y)| \leqslant \frac{1}{\min\limits_{g \in \widetilde{G}_{Latin}} \sqrt{\mu_g}} |f(y)| \sqrt{\mu_y} \leqslant \frac{1}{\min\limits_{g \in \widetilde{G}_{Latin}} \sqrt{\widetilde{\mu}_g}} |f(y)| \sqrt{\widetilde{\mu}_y}$$

Thus we find

$$\sqrt{\sum_{\substack{y \in G \\ y \neq \star}} \left( \left( 1 - \frac{\widetilde{\mu}_y}{\mu_y^\delta} \right) |f(y)| \right)^2} \leqslant \delta \left( \frac{K \widetilde{\mu}(\widetilde{G}_{Greek})}{\min\limits_{g \in \widetilde{G}_{Latin}} \widetilde{\mu}_g^{\frac{3}{2}}} \right) \sqrt{\sum_{\substack{y \in G \\ y \neq \star}} |f(y)|^2 \mu_y} \leqslant \delta \left( \frac{K \widetilde{\mu}(\widetilde{G}_{Greek})}{\min\limits_{g \in \widetilde{G}_{Latin}} \widetilde{\mu}_g^{\frac{3}{2}}} \right) \|f\|_{\ell^2(G)}.$$

To estimate the second term, we estimate

$$|f(g)| \leqslant \frac{1}{\min\limits_{g \in \widetilde{G}_{Latin} \cup \{\star\}} \sqrt{\widetilde{\mu}_g}} \|f\|_{\ell^2(G)}$$

to obtain

$$\left| \sum_{\substack{g \in G \\ g \neq y}} f(g) \frac{\langle \psi_g^\delta, \psi_y^\delta \rangle_{\ell^2(\widetilde{G})}}{\mu_y^\delta} \right| \leqslant \left( \frac{1}{\min\limits_{g \in \widetilde{G}_{Latin} \cup \{\star\}} \sqrt{\widetilde{\mu}_g}} \right) \|f(y)\|_{\ell^2(G)} \cdot \left| \sum_{\substack{g \in G \\ g \neq y}} \frac{\langle \psi_g^\delta, \psi_y^\delta \rangle_{\ell^2(\widetilde{G})}}{\mu_y^\delta} \right|$$

$$= \left( \frac{1}{\min\limits_{g \in G_{Latin} \cup \{\star\}} \sqrt{\widetilde{\mu}_g}} \right) \|f(y)\|_{\ell^2(G)} \cdot \left| \sum_{\substack{g \in G \\ g \neq y}} \frac{\langle \vec{\eta}_g^\delta, \vec{\eta}_y^\delta \rangle_{\ell^2(\widetilde{G}_{Greek})}}{\mu_y^\delta} \right|$$

$$\leqslant \left( \frac{1}{\min\limits_{g \in \widetilde{G}_{Latin} \cup \{\star\}} \sqrt{\widetilde{\mu}_g}} \right) \|f(y)\|_{\ell^2(G)} \cdot \left| \sum_{\substack{g \in G \\ g \neq y}} \frac{\langle \vec{\eta}_g^\delta, \vec{\eta}_y^\delta \rangle_{\ell^2(\widetilde{G}_{Greek})}}{\widetilde{\mu}_y} \right|$$

Thus we find (using that $\langle \vec{\eta}_g^\delta, \vec{\eta}_y^\delta \rangle_{\ell^2(\widetilde{G}_{Greek})}$ is a non-negative number and we have $\| \cdot \|_2 \leqslant \| \cdot \|_1$)

$$\sqrt{\sum_{\substack{y \in G \\ y \neq \star}} \left( \left| \sum_{\substack{g \in G \\ g \neq y}} f(g) \frac{\langle \psi_g^\delta, \psi_y^\delta \rangle_{\ell^2(\widetilde{G})}}{\mu_y^\delta} \right| \right)^2} \leqslant \frac{1}{\min\limits_{g \in \widetilde{G}_{Latin} \cup \{\star\}} \sqrt{\widetilde{\mu}_g}} \|f\|_{\ell^2(G)} \cdot \sum_{\substack{y \in G \\ y \neq \star}} \sum_{\substack{g \in G \\ g \neq y}} \frac{\langle \vec{\eta}_g^\delta, \vec{\eta}_y^\delta \rangle_{\ell^2(\widetilde{G}_{Greek})}}{\widetilde{\mu}_y}$$

$$\leqslant \frac{1}{\min\limits_{g \in \widetilde{G}_{Latin} \cup \{\star\}} \widetilde{\mu}_g^{\frac{3}{2}}} \|f\|_{\ell^2(G)} \cdot \sum_{\substack{y \in G \\ y \neq \star}} \sum_{\substack{g \in G \\ g \neq y}} \langle \vec{\eta}_g^\delta, \vec{\eta}_y^\delta \rangle_{\ell^2(\widetilde{G}_{Greek})}$$

$$\leqslant \frac{1}{\min\limits_{g \in \widetilde{G}_{Latin} \cup \{\star\}} \widetilde{\mu}_g^{\frac{3}{2}}} \|f\|_{\ell^2(G)} \cdot \sum_{\substack{y \in G \\ y \neq \star}} \sum_{g \in G} \langle \vec{\eta}_g^\delta, \vec{\eta}_y^\delta \rangle_{\ell^2(\widetilde{G}_{Greek})}$$

$$\leqslant \frac{1}{\min\limits_{g \in \widetilde{G}_{Latin} \cup \{\star\}} \widetilde{\mu}_g^{\frac{3}{2}}} \|f\|_{\ell^2(G)} \cdot \langle \mathbb{1}_{\widetilde{G}_{Greek}}, \vec{\zeta}_\star^\delta \rangle_{\ell^2(\widetilde{G}_{Greek})}$$

$$\leqslant \frac{1}{\min\limits_{g \in \widetilde{G}_{Latin} \cup \{\star\}} \widetilde{\mu}_g^{\frac{3}{2}}} \|f\|_{\ell^2(G)} \cdot \|\mathbb{1}_{\widetilde{G}_{Greek}}\|_{\ell^2(\widetilde{G}_{Greek})} \cdot \|\vec{\zeta}_\star^\delta\|_{\ell^2(\widetilde{G}_{Greek})}$$

$$\leqslant \delta \cdot \left( \frac{K \cdot \sqrt{\widetilde{\mu}(\widetilde{G}_{Greek})}}{\min\limits_{g \in \widetilde{G}_{Latin} \cup \{\star\}} \widetilde{\mu}_g^{\frac{3}{2}}} \right) \|f\|_{\ell^2(G)}$$

Let us thus turn to the remaining term; corresponding to $y = \star$: We have

$$\left| f(\star) - \sum_{g \in G} f(g) \frac{\langle \psi_g^\delta, \psi_\star^\delta \rangle_{\ell^2(\widetilde{G})}}{\mu_\star^\delta} \right| \leqslant \left| 1 - \frac{\langle \psi_\star^\delta, \psi_\star^\delta \rangle_{\ell^2(\widetilde{G})}}{\mu_\star^\delta} \right| |f(\star)| + \left| \sum_{\substack{g \in G \\ g \neq \star}} f(g) \frac{\langle \psi_g^\delta, \psi_\star^\delta \rangle_{\ell^2(\widetilde{G})}}{\mu_\star^\delta} \right| \quad (22)$$

We first deal with the left summand. We note

$$
\begin{aligned}
\left| 1 - \frac{\langle \psi_\star^\delta, \psi_\star^\delta \rangle_{\ell^2(\widetilde{G})}}{\mu_\star^\delta} \right| &= \left| \frac{\mu_\star^\delta - \widetilde{\mu}_\star - \langle \mathbb{1}_{\widetilde{G}_{Greek}} - \vec{\zeta}_\star^\delta, \mathbb{1}_{\widetilde{G}_{Greek}} - \vec{\zeta}_\star^\delta \rangle_{\ell^2(\widetilde{G}_{Greek})}}{\mu_\star^\delta} \right| \\
&\leqslant \left| \frac{\mu_\star^\delta - \widetilde{\mu}_\star - \langle \mathbb{1}_{\widetilde{G}_{Greek}} - \vec{\zeta}_\star^\delta, \mathbb{1}_{\widetilde{G}_{Greek}} - \vec{\zeta}_\star^\delta \rangle_{\ell^2(\widetilde{G}_{Greek})}}{\widetilde{\mu}_\star + \widetilde{\mu}(\widetilde{G}_{Greek}) - \delta K \widetilde{\mu}(\widetilde{G}_{Greek})} \right| \\
&\leqslant \left| \frac{\left( \mu_\star^\delta - \widetilde{\mu}_\star - \langle \mathbb{1}_{\widetilde{G}_{Greek}}, \mathbb{1}_{\widetilde{G}_{Greek}} \rangle_{\ell^2(\widetilde{G}_{Greek})} \right) + \left( \langle \vec{\zeta}_\star^\delta, \vec{\zeta}_\star^\delta \rangle_{\ell^2(\widetilde{G}_{Greek})} - 2 \langle \mathbb{1}_{\widetilde{G}_{Greek}}, \vec{\zeta}_\star^\delta \rangle_{\ell^2(\widetilde{G}_{Greek})} \right)}{\widetilde{\mu}_\star + \widetilde{\mu}(\widetilde{G}_{Greek}) - \delta K \widetilde{\mu}(\widetilde{G}_{Greek})} \right| \\
&\leqslant \frac{(\delta K) + \left| \langle \vec{\zeta}_\star^\delta, \vec{\zeta}_\star^\delta \rangle_{\ell^2(\widetilde{G}_{Greek})} - 2 \langle \mathbb{1}_{\widetilde{G}_{Greek}}, \vec{\zeta}_\star^\delta \rangle_{\ell^2(\widetilde{G}_{Greek})} \right|}{\widetilde{\mu}_\star + \widetilde{\mu}(\widetilde{G}_{Greek}) - \delta K \widetilde{\mu}(\widetilde{G}_{Greek})} \\
&\leqslant \frac{(\delta K) + \delta^2 K^2 + 2 \| \mathbb{1}_{\widetilde{G}_{Greek}} \|_{\ell^2(\widetilde{G}_{Greek})} \cdot \| \vec{\zeta}_\star^\delta \|_{\ell^2(\widetilde{G}_{Greek})}}{\widetilde{\mu}_\star + \widetilde{\mu}(\widetilde{G}_{Greek}) - \delta K \widetilde{\mu}(\widetilde{G}_{Greek})} \\
&\leqslant \frac{(\delta K) + \left| \langle \vec{\zeta}_\star^\delta, \vec{\zeta}_\star^\delta \rangle_{\ell^2(\widetilde{G}_{Greek})} - 2 \langle \mathbb{1}_{\widetilde{G}_{Greek}}, \vec{\zeta}_\star^\delta \rangle_{\ell^2(\widetilde{G}_{Greek})} \right|}{\widetilde{\mu}_\star + \widetilde{\mu}(\widetilde{G}_{Greek}) - \delta K \widetilde{\mu}(\widetilde{G}_{Greek})} \\
&\leqslant \frac{(\delta K) + \delta^2 K^2 + 2 \sqrt{\widetilde{\mu}(\widetilde{G}_{Greek})} K \delta}{\widetilde{\mu}_\star + \widetilde{\mu}(\widetilde{G}_{Greek}) - \delta K \widetilde{\mu}(\widetilde{G}_{Greek})} \\
&\leqslant \frac{(\delta K) + \delta^2 K^2 + 2 \sqrt{\widetilde{\mu}(\widetilde{G}_{Greek})} K \delta}{\widetilde{\mu}_\star}
\end{aligned}
$$

Thus, under the assumption $\delta \leqslant 1$ (implying $\delta^2 \leqslant \delta$), we have

$$\left| 1 - \frac{\langle \psi_\star^\delta, \psi_\star^\delta \rangle_{\ell^2(\widetilde{G})}}{\mu_\star^\delta} \right| \leqslant \frac{K + K^2 + 2 \sqrt{\widetilde{\mu}(\widetilde{G}_{Greek})} K}{\widetilde{\mu}_\star} \cdot \delta.$$

This implies that we have

$$\left| f(\star) - \sum_{g \in G} f(g) \frac{\langle \psi_g^\delta, \psi_\star^\delta \rangle_{\ell^2(\widetilde{G})}}{\mu_\star^\delta} \right| \leqslant \delta \cdot \frac{K + K^2 + 2 \sqrt{\widetilde{\mu}(\widetilde{G}_{Greek})} K}{\widetilde{\mu}_\star^{\frac{3}{2}}} \cdot \| f \|_{\ell^2(G)}.$$

For the right-hand-side summand of the estimate in (22) we note

$$\left| \sum_{\substack{g \in G \\ g \neq \star}} f(g) \frac{\langle \psi_g^\delta, \psi_\star^\delta \rangle_{\ell^2(\widetilde{G})}}{\mu_\star^\delta} \right| = \left| \sum_{\substack{g \in G \\ g \neq \star}} f(g) \frac{\langle \vec{\eta}_g^\delta, \vec{\eta}_\star^\delta \rangle_{\ell^2(\widetilde{G}_{Greek})}}{\mu_\star^\delta} \right|$$

$$\leqslant \frac{1}{\min\limits_{g \in \widetilde{G}_{Latin} \cup \{\star\}} \widetilde{\mu}_g^{\frac{3}{2}}} \|f\|_{\ell^2(G)} \sum_{\substack{g \in G \\ g \neq \star}} \langle \vec{\eta}_g^\delta, \vec{\eta}_\star^\delta \rangle_{\ell^2(\widetilde{G}_{Greek})}$$

$$\leqslant \frac{1}{\min\limits_{g \in \widetilde{G}_{Latin} \cup \{\star\}} \widetilde{\mu}_g^{\frac{3}{2}}} \|f\|_{\ell^2(G)} \sum_{g \in G} \langle \vec{\eta}_g^\delta, \vec{\eta}_\star^\delta \rangle_{\ell^2(\widetilde{G}_{Greek})}$$

$$= \frac{1}{\min\limits_{g \in \widetilde{G}_{Latin} \cup \{\star\}} \widetilde{\mu}_g^{\frac{3}{2}}} \|f\|_{\ell^2(G)} \langle \mathbb{1}_{\widetilde{G}_{Greek}}, \vec{\zeta}_\star^\delta \rangle_{\ell^2(\widetilde{G}_{Greek})}$$

$$\delta \cdot \left( \frac{K \cdot \sqrt{\widetilde{\mu}(\widetilde{G}_{Greek})}}{\min\limits_{g \in \widetilde{G}_{Latin} \cup \{\star\}} \widetilde{\mu}_g^{\frac{3}{2}}} \right) \|f\|_{\ell^2(G)}.$$

Putting it all together, we find for $\delta \leqslant 1$ that

$$\|f - \widetilde{J}Jf\|_{\ell^2(G)} \leqslant \delta \cdot K^A \cdot \|f\|_{\ell^2(G)}$$

with

$$K^A := \left( \frac{K\widetilde{\mu}(\widetilde{G}_{Greek})}{\min\limits_{g \in \widetilde{G}_{Latin}} \widetilde{\mu}_g^{\frac{3}{2}}} \right) + 2 \left( \frac{K \cdot \sqrt{\widetilde{\mu}(\widetilde{G}_{Greek})}}{\min\limits_{g \in \widetilde{G}_{Latin} \cup \{\star\}} \widetilde{\mu}_g^{\frac{3}{2}}} \right) + \frac{K + K^2 + 2\sqrt{\widetilde{\mu}(\widetilde{G}_{Greek})}K}{\widetilde{\mu}_\star^{\frac{3}{2}}}.$$

Thus the left hand side of (19) holds with

$$\epsilon = K^A \cdot \delta.$$

**Right-hand-side of (19):**
Hence let us now check the right hand side of (19). We note

$$(u - J\widetilde{J}u) = u - \sum_{x \in G} \frac{\langle \psi_x^\delta, u \rangle_{\ell^2(\widetilde{G})}}{\mu_x^\delta} \psi_x^\delta.$$

Let us denote by $M$ the matrix representation

$$M^\delta = Id - \widetilde{J}J = Id - \sum_{x \in G} \frac{\langle \psi_x^\delta, \cdot \rangle_{\ell^2(\widetilde{G})}}{\mu_x^\delta} \psi_x^\delta.$$

We use the triangle inequality to arrive at

$$\left\| (u - J\widetilde{J}u) \right\|_{\ell^2(\widetilde{G})} \leqslant \left\| M^0 \cdot u \right\|_{\ell^2(\widetilde{G})} + \left\| M^\delta - M^0 \right\|_{op} \cdot \|u\|_{\ell^2(\widetilde{G})}. \tag{23}$$

Using the fact that for $g \neq \star$ we have $\vec{\eta}_g^\delta \to \vec{0}$ an $\vec{\eta}_\star^0 = \mathbb{1}_{\widetilde{G}_{Greek}}$ we find in the $(\delta \to 0)$-limit that

$$M^0 = \begin{pmatrix} 0_{|\widetilde{G}_{Latin}| \times |\widetilde{G}_{Latin}|} & 0_{|\widetilde{G}_{Latin}| \times |\widetilde{G}_{Greek} \cup \{\star\}|} \\ 0_{|\widetilde{G}_{Greek} \cup \{\star\}| \times |\widetilde{G}_{Latin}|} & \underline{M}^0 \end{pmatrix}$$

with

$$
\underline{M}^0 = \begin{pmatrix} 1 & & \\ & \ddots & \\ & & 1 \end{pmatrix} - \frac{1}{\widetilde{\mu}(\widetilde{G}_{Greek}) + \widetilde{\mu}_\star} \begin{pmatrix} \widetilde{\mu}_\star & \widetilde{\mu}_\alpha & \widetilde{\mu}_\beta & \cdots \\ \widetilde{\mu}_\star & \widetilde{\mu}_\alpha & \widetilde{\mu}_\beta & \cdots \\ \vdots & \vdots & \vdots & \end{pmatrix}
$$

acting on $\ell^2(\widetilde{G}_{Greek} \cup \{\star\})$. For any element $v \in \ell^2(\widetilde{G})$, let us denote its restriction to $\widetilde{G}_{Greek} \cup \{\star\}$ by $\underline{v} \in \ell^2(\widetilde{G}_{Greek} \cup \{\star\})$.
We thus find

$$
\left\| \underline{M}^0 \cdot \underline{u} \right\|^2_{\ell^2(\widetilde{G}_{Greek} \cup \{\star\})} = \left\langle \underline{M}^0 \cdot \underline{u}, \underline{M}^0 \cdot \underline{u} \right\rangle_{\ell^2(\widetilde{G}_{Greek} \cup \{\star\})}
$$

$$
= \sum_{i \in \widetilde{G}_{Greek} \cup \{\star\}} \sum_{j \in \widetilde{G}_{Greek} \cup \{\star\}} \underline{u}(i)\underline{u}(j) \sum_{a,b \in \widetilde{G}_{Greek} \cup \{\star\}} \left[ \delta_{ia} - \frac{\widetilde{\mu}_i}{\widetilde{\mu}(\widetilde{G}_{Greek}) + \widetilde{\mu}_\star} \right] \cdot \widetilde{\mu}_a \delta_{ab} \cdot \left[ \delta_{bj} - \frac{\widetilde{\mu}_j}{\widetilde{\mu}(\widetilde{G}_{Greek}) + \widetilde{\mu}_\star} \right]
$$

$$
= \sum_{i \in \widetilde{G}_{Greek} \cup \{\star\}} \sum_{j \in \widetilde{G}_{Greek} \cup \{\star\}} \underline{u}(i)\underline{u}(j) \sum_{a \in \widetilde{G}_{Greek} \cup \{\star\}} \left[ \delta_{ia} - \frac{\widetilde{\mu}_i}{\widetilde{\mu}(\widetilde{G}_{Greek}) + \widetilde{\mu}_\star} \right] \cdot \left[ \widetilde{\mu}_a \delta_{aj} - \frac{\widetilde{\mu}_a \widetilde{\mu}_j}{\widetilde{\mu}(\widetilde{G}_{Greek}) + \widetilde{\mu}_\star} \right]
$$

$$
= \sum_{i \in \widetilde{G}_{Greek} \cup \{\star\}} \sum_{j \in \widetilde{G}_{Greek} \cup \{\star\}} \underline{u}(i)\underline{u}(j) \times \ldots
$$

$$
\ldots \times \sum_{a \in \widetilde{G}_{Greek} \cup \{\star\}} \left[ \widetilde{\mu}_a \widetilde{\mu}_\star \delta_{ia} \delta_{aj} - \frac{\delta_{ia} \widetilde{\mu}_a \widetilde{\mu}_j}{\widetilde{\mu}(\widetilde{G}_{Greek}) + \widetilde{\mu}_\star} - \frac{\delta_{ij} \widetilde{\mu}_i \widetilde{\mu}_j}{\widetilde{\mu}(\widetilde{G}_{Greek}) + \widetilde{\mu}_\star} + \frac{\widetilde{\mu}_i \widetilde{\mu}_a \widetilde{\mu}_j}{(\widetilde{\mu}(\widetilde{G}_{Greek}) + \widetilde{\mu}_\star)^2} \right]
$$

$$
= \sum_{i \in \widetilde{G}_{Greek} \cup \{\star\}} \sum_{j \in \widetilde{G}_{Greek} \cup \{\star\}} \underline{u}(i)\underline{u}(j) \left[ \widetilde{\mu}_i \delta_{ij} - \frac{\widetilde{\mu}_i \widetilde{\mu}_j}{\widetilde{\mu}(\widetilde{G}_{Greek}) + \widetilde{\mu}_\star} \right]
$$

$$
= \sum_{i,j \in \widetilde{G}_{Greek} \cup \{\star\}} \left( \frac{\widetilde{\mu}_i \widetilde{\mu}_j}{\widetilde{\mu}(\widetilde{G}_{Greek}) + \widetilde{\mu}_\star} \right) |\underline{u}(i) - \underline{u}(j)|^2.
$$

To proceed, we prove the following Lemma:

**Lemma J.7.** Let $i, j \in \widetilde{G}_{Greek} \cup \{\star\}$. Denote by $C_{\widetilde{G}_{Greek} \cup \{\star\}}(i,j)$ the minimum number of edges for which $\omega_{ij} \gtrless 0$ needed to connect $i$ and $j$ by a path. Set

$$
C_{\widetilde{G}_{Greek} \cup \{\star\}} := \max_{i \neq j \in \widetilde{G}_{Greek} \cup \{\star\}} C_{\widetilde{G}_{Greek} \cup \{\star\}}(i,j).
$$

Furthermore set

$$
\Omega := \min_{i \neq j \in \widetilde{G}_{Greek} \cup \{\star\}} \omega_{ij}.
$$

We have

$$
|\underline{u}(i) - \underline{u}(j)| \leqslant \delta^{\frac{1}{2}} \left( \frac{C_{\widetilde{G}_{Greek} \cup \{\star\}}}{\sqrt{\Omega}} \right) \sqrt{E_{\widetilde{G}}(u)}.
$$

We call $C_{\widetilde{G}_{Greek} \cup \{\star\}}$ the **connectivity constant** of the sub-graph $\widetilde{G}_{Greek} \cup \{\star\}$ and note that it is well-defined since we assume $\widetilde{G}_{Greek} \cup \{\star\}$ to be connected.

*Proof.* Fix $i$ and $j$. Let $\{i, g_1, ..., g_n, j\}$ be the vertices traversed by a path of minimal length determining $C_{\widetilde{G}_{Greek} \cup \{\star\}}(i,j)$. We then have

$$
\begin{aligned}
&|\underline{u}(i) - \underline{u}(j)| \\
&\leqslant |\underline{u}(i) - \underline{u}(g_1)| + |\underline{u}(g_1) - \underline{u}(g_2)| + ... + |\underline{u}(g_n) - \underline{u}(j)| \\
&\leqslant \delta^{\frac{1}{2}} \frac{1}{\sqrt{\Omega}} \left( \sqrt{\widetilde{W}_{ig_1} |\underline{u}(i) - \underline{u}(g_1)|^2} + \sqrt{\widetilde{W}_{g_1 g_2} |\underline{u}(g_1) - \underline{u}(g_2)|^2} + ... + \sqrt{\widetilde{W}_{g_n j} |\underline{u}(g_n) - \underline{u}(j)|^2} \right) \\
&\leqslant \delta^{\frac{1}{2}} \frac{1}{\sqrt{\Omega}} \left( \sqrt{E_{\widetilde{G}}(u)} + \sqrt{E_{\widetilde{G}}(u)} + ... + \sqrt{E_{\widetilde{G}}(u)} \right) \\
&= \delta^{\frac{1}{2}} \frac{C_{\widetilde{G}_{Greek} \cup \{\star\}}(i,j)}{\sqrt{\Omega}} \sqrt{E_{\widetilde{G}}(u)} \\
&\leqslant \delta^{\frac{1}{2}} \frac{C_{\widetilde{G}_{Greek} \cup \{\star\}}}{\sqrt{\Omega}} \sqrt{E_{\widetilde{G}}(u)}.
\end{aligned}
$$

$\square$

With the help of this Lemma we then find

$$
\begin{aligned}
\left\| M^0 \cdot u \right\|_{\ell^2(\widetilde{G})} &\leqslant \delta^{\frac{1}{2}} \frac{C_{\widetilde{G}_{Greek} \cup \{\star\}}}{\sqrt{\Omega}} \sqrt{E_{\widetilde{G}}(u)} \cdot \sqrt{\sum_{i,j \in \widetilde{G}_{Greek} \cup \{\star\}} \left( \frac{\widetilde{\mu}_i \widetilde{\mu}_j}{\widetilde{\mu}(\widetilde{G}_{Greek}) + \widetilde{\mu}_\star} \right)} \\
&= \delta^{\frac{1}{2}} \cdot \left( \frac{C_{\widetilde{G}_{Greek} \cup \{\star\}} \cdot \sqrt{\widetilde{\mu}(\widetilde{G}_{Greek}) + \widetilde{\mu}_\star}}{\sqrt{\Omega}} \right) \cdot \sqrt{E_{\widetilde{G}}(u)}.
\end{aligned}
$$

To derive a bound for $\left\| M^\delta - M^0 \right\|_{op}$ in the second term of the estimate (23), we write

$$
M^\delta - M^0 = \begin{pmatrix} B & A \\ A^\dagger & D \end{pmatrix}.
$$

Here we denote by

$$
A^\dagger : \ell^2(\widetilde{G}_{Latin}) \longrightarrow \ell^2(\widetilde{G}_{Greek} \cup \{\star\})
$$

the adjoint of the operator

$$
A : \ell^2(\widetilde{G}_{Greek} \cup \{\star\}) \longrightarrow \ell^2(\widetilde{G}_{Latin}).
$$

Clearly $\|A\|_{op} = \|A^\dagger\|_{op}$ so that we have

$$
\left\| M^\delta - M^0 \right\|_{op} \leqslant \|B\|_{op} + 2 \|A\|_{op} + \|D\|_{op}. \tag{24}
$$

To bound $\|B\|_{op}$ we note that $B$ is diagonal and we have

$$
B = \begin{pmatrix} \widetilde{\mu}_a \left( \frac{1}{\mu_a^\delta} - \frac{1}{\mu_a^0} \right) & & \\ & \widetilde{\mu}_b \left( \frac{1}{\mu_b^\delta} - \frac{1}{\mu_b^0} \right) & \\ & & \ddots \end{pmatrix}
$$

so that

$$
\begin{aligned}
\|B\|_{op} &\leqslant \left[ \max_{a \in \widetilde{G}_{Latin}} \widetilde{\mu}_a \left| \frac{1}{\mu_a^{\delta}} - \frac{1}{\mu_a^{0}} \right| \right] \\
&= \left[ \max_{a \in \widetilde{G}_{Latin}} \widetilde{\mu}_a \left| \frac{1}{\mu_a^{\delta}} - \frac{1}{\mu_a^{0}} \right| \right] \\
&= \left[ \max_{a \in \widetilde{G}_{Latin}} \widetilde{\mu}_a \left| \frac{\mu_a^{\delta} - \mu_a^{0}}{\mu_a^{\delta} \cdot \mu_a^{0}} \right| \right] \\
&\leqslant \left[ \max_{a \in \widetilde{G}_{Latin}} \widetilde{\mu}_a \left| \frac{\mu_a^{\delta} - \mu_a^{0}}{\widetilde{\mu}_a^{2}} \right| \right] \\
&\leqslant \left[ \max_{a \in \widetilde{G}_{Latin}} \widetilde{\mu}_a \left| \frac{K \delta \widetilde{\mu}(\widetilde{G}_{Greek})}{\widetilde{\mu}_a^{2}} \right| \right] \\
&\leqslant \delta \cdot \left[ \frac{K \cdot \widetilde{\mu}(\widetilde{G}_{Greek})}{\displaystyle\min_{a \in \widetilde{G}_{Latin}} \mu_a} \right].
\end{aligned}
$$

To estimate $\|A\|_{op}$ we note

$$
A = \begin{pmatrix}
0 & \frac{\vec{\eta}_a^{\delta}(\alpha)}{\mu_a^{\delta}} & \frac{\vec{\eta}_a^{\delta}(\beta)}{\mu_a^{\delta}} & \cdots \\
0 & \frac{\vec{\eta}_b^{\delta}(\alpha)}{\mu_b^{\delta}} & \frac{\vec{\eta}_a^{\delta}(\beta)}{\mu_b^{\delta}} & \cdots \\
0 & \frac{\vec{\eta}_c^{\delta}(\alpha)}{\mu_c^{\delta}} & \frac{\vec{\eta}_c^{\delta}(\beta)}{\mu_c^{\delta}} & \cdots \\
\vdots & \vdots & \vdots &
\end{pmatrix}.
$$

We can consider the map

$$
A : \ell^2(\widetilde{G}_{Greek} \cup \{\star\}) \longrightarrow \ell^2(\widetilde{G}_{Latin}).
$$

as a composition of maps

$$
A : \ell^2(\widetilde{G}_{Greek} \cup \{\star\}) \xrightarrow{Id} \mathbb{C}^{|\widetilde{G}_{Greek} \cup \{\star\}|} \xrightarrow{A} \mathbb{C}^{|\widetilde{G}_{Latin}|} \xrightarrow{Id} \ell^2(\widetilde{G}_{Latin}).
$$

For the map $Id : \ell^2(\widetilde{G}_{Greek} \cup \{\star\}) \to \mathbb{C}^{|\widetilde{G}_{Greek} \cup \{\star\}|}$ we find $\|Id\|_{op} = \left( \min_{g \in \widetilde{G}_{Greek} \cup \{\star\}} \widetilde{\mu}_g \right)^{-1}$. Similarly

we find for the map $Id : \ell^2(\widetilde{G}_{Latin}) \to \mathbb{C}^{|\widetilde{G}_{Latin}|}$ that $\|Id\|_{op} = \left( \max_{g \in \widetilde{G}_{Latin}} \widetilde{\mu}_g \right)$. To bound the operator

norm of the map $A : \mathbb{C}^{|\widetilde{G}_{Greek} \cup \{\star\}|} \to \mathbb{C}^{|\widetilde{G}_{Latin}|}$, we use that the operator-norm is smaller than the

maximal column-sum times $\sqrt{|\widetilde{G}_{Greek} \cup \{\star\}|}$. Hence for $A$ as a map from $\mathbb{C}^{|\widetilde{G}_{Greek} \cup \{\star\}|}$ to $\mathbb{C}^{|\widetilde{G}_{Latin}|}$ we

find

$$\|A\|_{op} \leqslant \sqrt{|\widetilde{G}_{Greek} \cup \{\star\}|} \cdot \left(\frac{1}{\min\limits_{g \in \widetilde{G}_{Latin}} \mu_g^\delta}\right) \cdot \max_{\alpha \in \widetilde{G}_{Greek}} \left[\sum_{a \in \widetilde{G}_{Latin}} \vec{\eta}_a^\delta(\alpha)\right]$$

$$= \sqrt{|\widetilde{G}_{Greek} \cup \{\star\}|} \cdot \left(\frac{1}{\min\limits_{g \in \widetilde{G}_{Latin}} \mu_g^\delta}\right) \cdot \max_{\alpha \in \widetilde{G}_{Greek}} \left[\vec{\zeta}_\star^\delta(\alpha)\right]$$

$$= \delta \cdot K \cdot \sqrt{|\widetilde{G}_{Greek} \cup \{\star\}|} \cdot \left(\frac{1}{\min\limits_{g \in \widetilde{G}_{Latin}} \mu_g^\delta \cdot \min\limits_{\alpha \in \widetilde{G}_{Greek}} \sqrt{\widetilde{\mu}_\alpha}}\right)$$

$$\leqslant \delta \cdot K \cdot \sqrt{|\widetilde{G}_{Greek} \cup \{\star\}|} \cdot \left(\frac{1}{\min\limits_{g \in \widetilde{G}_{Latin}} \widetilde{\mu}_g \cdot \max\limits_{\alpha \in \widetilde{G}_{Greek}} \sqrt{\widetilde{\mu}_\alpha}}\right).$$

Here we estimated

$$\max_{\alpha \in \widetilde{G}_{Greek}} \left[\vec{\zeta}_\star^\delta(\alpha)\right] \leqslant \frac{1}{\min\limits_{\alpha \in \widetilde{G}_{Greek}} \sqrt{\widetilde{\mu}_\alpha}} \|\vec{\zeta}_\star^\delta\|_{\ell^2(\widetilde{G}_{Greek})}.$$

In total, we find for the operator-norm of

$$A : \ell^2(\widetilde{G}_{Greek} \cup \{\star\}) \longrightarrow \ell^2(\widetilde{G}_{Latin}).$$

that

$$\|A\|_{op} \leqslant \delta \cdot K \cdot \sqrt{|\widetilde{G}_{Greek} \cup \{\star\}|} \cdot \left(\frac{\max\limits_{g \in \widetilde{G}_{Latin}} \widetilde{\mu}_g}{\min\limits_{g \in \widetilde{G}_{Latin}} \widetilde{\mu}_g \cdot \max\limits_{\alpha \in \widetilde{G}_{Greek} \cup \{\star\}} \widetilde{\mu}_\alpha^{\frac{3}{2}}}\right).$$

Thus let us now investigate $\|D\|_{op}$. As before. let us denote by $\underline{u} \in \ell^2(\widetilde{G}_{Greek} \cup \{\star\})$ the restriction of an element $u \in \ell^2(\widetilde{G}$ to $\widetilde{G}_{Greek} \cup \{\star\}$. We have

$$\|D\|_{op} = \left\|\sum_{x \in \widetilde{G}_{Latin} \cup \{\star\}} \frac{\langle \underline{\psi}_x^\delta, \cdot \rangle_{\ell^2(\widetilde{G}_{Greek} \cup \{\star\})}}{\mu_x^\delta} \underline{\psi}_x^\delta - \sum_{x \in \widetilde{G}_{Latin} \cup \{\star\}} \frac{\langle \underline{\psi}_x^0, \cdot \rangle_{\ell^2(\widetilde{G}_{Greek} \cup \{\star\})}}{\mu_x^0} \underline{\psi}_x^0\right\|$$

$$\leqslant \left\|\sum_{x \in \widetilde{G}_{Latin}} \frac{\langle \underline{\psi}_x^\delta, \cdot \rangle_{\ell^2(\widetilde{G}_{Greek} \cup \{\star\})}}{\mu_x^\delta} \underline{\psi}_x^\delta - \sum_{x \in \widetilde{G}_{Latin}} \frac{\langle \underline{\psi}_x^0, \cdot \rangle_{\ell^2(\widetilde{G}_{Greek} \cup \{\star\})}}{\mu_x^0} \underline{\psi}_x^0\right\|$$

$$+ \left\|\frac{\langle \underline{\psi}_\star^\delta, \cdot \rangle_{\ell^2(\widetilde{G}_{Greek} \cup \{\star\})}}{\mu_\star^\delta} \underline{\psi}_\star^\delta - \frac{\langle \underline{\psi}_\star^0, \cdot \rangle_{\ell^2(\widetilde{G}_{Greek} \cup \{\star\})}}{\mu_\star^0} \underline{\psi}_\star^0\right\|$$

$$= \left\|\sum_{x \in \widetilde{G}_{Latin}} \frac{\langle \underline{\psi}_x^\delta, \cdot \rangle_{\ell^2(\widetilde{G}_{Greek} \cup \{\star\})}}{\mu_x^\delta} \underline{\psi}_x^\delta\right\| + \left\|\frac{\langle \underline{\psi}_\star^\delta, \cdot \rangle_{\ell^2(\widetilde{G}_{Greek} \cup \{\star\})}}{\mu_\star^\delta} \underline{\psi}_\star^\delta - \frac{\langle \underline{\psi}_\star^0, \cdot \rangle_{\ell^2(\widetilde{G}_{Greek} \cup \{\star\})}}{\mu_\star^0} \underline{\psi}_\star^0\right\|.$$

We note for the matrix representation of the first term, that (with $\alpha, \beta \in \widetilde{G}_{Greek} \cup \{\star\}$) we have

$$\left(\sum_{x \in \widetilde{G}_{Latin}} \frac{\langle \underline{\psi}_x^\delta, \cdot \rangle_{\ell^2(\widetilde{G}_{Greek} \cup \{\star\})}}{\mu_x^\delta} \underline{\psi}_x^\delta\right)_{\alpha\beta} = \left(\sum_{x \in \widetilde{G}_{Latin}} \frac{1}{\mu_x^\delta} \vec{\eta}_x^\delta(\alpha) \vec{\eta}_x^\delta(\beta) \widetilde{\mu}_\beta\right).$$

Using the 'maximal row sum trick' complementary to the 'maximal column sum trick' already used for $A$ above and recalling the definition of the weights

$$\mu_g^\delta := \sum_{h \in \widetilde{G}} \psi_g^\delta(h) \cdot \widetilde{\mu}_h$$

we find

$$\left\| \sum_{x \in \widetilde{G}_{Latin}} \frac{\langle \psi_x^\delta, \cdot \rangle_{\ell^2(\widetilde{G}_{Greek} \cup \{\star\})}}{\mu_x^\delta} \underline{\psi_x^\delta} \right\|$$

$$\leq \sqrt{|\widetilde{G}_{Greek} \cup \{\star\}|} \cdot \frac{\max\limits_{x \in \widetilde{G}_{Greek} \cup \{\star\}} \sqrt{\widetilde{\mu}_x}}{\min\limits_{x \in \widetilde{G}_{Greek} \cup \{\star\}} \sqrt{\widetilde{\mu}_x}} \cdot \max_{\beta \in \widetilde{G}_{Greek} \cup \{\star\}} \left( \sum_{\alpha \in \widetilde{G}_{Greek} \cup \{\star\}} \left( \sum_{x \in \widetilde{G}_{Latin}} \frac{1}{\mu_x^\delta} \vec{\eta}_x^\delta(\alpha) \vec{\eta}_x^\delta(\beta) \widetilde{\mu}_\beta \right) \right)$$

$$\leq \sqrt{|\widetilde{G}_{Greek} \cup \{\star\}|} \cdot \frac{\max\limits_{x \in \widetilde{G}_{Greek} \cup \{\star\}} \sqrt{\widetilde{\mu}_x}}{\min\limits_{y \in \widetilde{G}_{Greek} \cup \{\star\}} \sqrt{\widetilde{\mu}_y}} \cdot \max_{\alpha \in \widetilde{G}_{Greek} \cup \{\star\}} \left( \sum_{x \in \widetilde{G}_{Latin}} \frac{1}{\mu_x^\delta} \vec{\eta}_x^\delta(\alpha) \right)$$

$$\leq \sqrt{|\widetilde{G}_{Greek} \cup \{\star\}|} \cdot \frac{\max\limits_{x \in \widetilde{G}_{Greek} \cup \{\star\}} \sqrt{\widetilde{\mu}_x}}{\min\limits_{y \in \widetilde{G}_{Greek} \cup \{\star\}} \sqrt{\widetilde{\mu}_y}} \cdot \max_{\alpha \in \widetilde{G}_{Greek} \cup \{\star\}} \left( \sum_{x \in \widetilde{G}_{Latin}} \vec{\eta}_x^\delta(\alpha) \right)$$

$$\leq \sqrt{|\widetilde{G}_{Greek} \cup \{\star\}|} \cdot \frac{\max\limits_{x \in \widetilde{G}_{Greek} \cup \{\star\}} \sqrt{\widetilde{\mu}_x}}{\min\limits_{y \in \widetilde{G}_{Greek} \cup \{\star\}} \sqrt{\widetilde{\mu}_y}} \cdot \max_{\alpha \in \widetilde{G}_{Greek} \cup \{\star\}} \vec{\zeta}_\star^\delta(\alpha)$$

$$\leq \sqrt{|\widetilde{G}_{Greek} \cup \{\star\}|} \cdot \frac{\max\limits_{x \in \widetilde{G}_{Greek} \cup \{\star\}} \sqrt{\widetilde{\mu}_x}}{\min\limits_{y \in \widetilde{G}_{Greek} \cup \{\star\}} \widetilde{\mu}_y^{\frac{3}{2}}} \cdot \max_{\alpha \in \widetilde{G}_{Greek} \cup \{\star\}} \|\vec{\zeta}_\star^\delta\|_{\ell^2(\widetilde{G}_{Greek})}$$

$$\leq \sqrt{|\widetilde{G}_{Greek} \cup \{\star\}|} \cdot \frac{\max\limits_{x \in \widetilde{G}_{Greek} \cup \{\star\}} \sqrt{\widetilde{\mu}_x}}{\min\limits_{y \in \widetilde{G}_{Greek} \cup \{\star\}} \widetilde{\mu}_y^{\frac{3}{2}}} \cdot K \cdot \delta.$$

It remains to bound the second term. We find (using $\left\| \underline{\psi}_\star^\delta \right\|_{\ell^2(\widetilde{G}_{Greek} \cup \{\star\})} \leqslant \left\| \underline{\psi}_\star^0 \right\|_{\ell^2(\widetilde{G}_{Greek} \cup \{\star\})}$):

$$
\left\| \frac{\langle \underline{\psi}_\star^\delta, \underline{u} \rangle_{\ell^2(\widetilde{G}_{Greek} \cup \{\star\})}}{\mu_\star^\delta} \underline{\psi}_\star^\delta - \frac{\langle \underline{\psi}_\star^0, \underline{u} \rangle_{\ell^2(\widetilde{G}_{Greek} \cup \{\star\})}}{\mu_\star^0} \underline{\psi}_\star^0 \right\|_{\ell^2(\widetilde{G}_{Greek} \cup \{\star\})}
$$

$$
\leqslant \left\| \left( \frac{1}{\mu_\star^\delta} - \frac{1}{\mu_\star^0} \right) \langle \underline{\psi}_\star^\delta, \underline{u} \rangle_{\ell^2(\widetilde{G}_{Greek} \cup \{\star\})} \underline{\psi}_\star^\delta \right\|_{\ell^2(\widetilde{G}_{Greek} \cup \{\star\})}
$$

$$
+ \frac{1}{\mu_\star^0} \left\| \langle \underline{\psi}_\star^\delta, \underline{u} \rangle_{\ell^2(\widetilde{G}_{Greek} \cup \{\star\})} \underline{\psi}_\star^\delta - \langle \underline{\psi}_\star^0, \underline{u} \rangle_{\ell^2(\widetilde{G}_{Greek} \cup \{\star\})} \underline{\psi}_\star^0 \right\|_{\ell^2(\widetilde{G}_{Greek} \cup \{\star\})}
$$

$$
\leqslant \left| \frac{1}{\mu_\star^\delta} - \frac{1}{\mu_\star^0} \right| \cdot \left\| \underline{\psi}_\star^\delta \right\|^2_{\ell^2(\widetilde{G}_{Greek} \cup \{\star\})} \cdot \left\| \underline{u} \right\|_{\ell^2(\widetilde{G}_{Greek} \cup \{\star\})}
$$

$$
+ \frac{1}{\mu_\star^0} \left\| \left( \langle \underline{\psi}_\star^\delta, \underline{u} \rangle_{\ell^2(\widetilde{G}_{Greek} \cup \{\star\})} - \langle \underline{\psi}_\star^0, \underline{u} \rangle_{\ell^2(\widetilde{G}_{Greek} \cup \{\star\})} \right) \underline{\psi}_\star^0 + \langle \underline{\psi}_\star^\delta, \underline{u} \rangle_{\ell^2(\widetilde{G}_{Greek} \cup \{\star\})} \left( \underline{\psi}_\star^\delta - \underline{\psi}_\star^0 \right) \right\|_{\ell^2(\widetilde{G}_{Greek} \cup \{\star\})}
$$

$$
\leqslant \left| \frac{1}{\mu_\star^\delta} - \frac{1}{\mu_\star^0} \right| \cdot \left\| \underline{\psi}_\star^0 \right\|^2_{\ell^2(\widetilde{G}_{Greek} \cup \{\star\})} \cdot \left\| \underline{u} \right\|_{\ell^2(\widetilde{G}_{Greek} \cup \{\star\})}
$$

$$
+ 2 \frac{1}{\mu_\star^0} \left\| \underline{\psi}_\star^\delta - \underline{\psi}_\star^0 \right\|_{\ell^2(\widetilde{G}_{Greek} \cup \{\star\})} \cdot \left\| \underline{\psi}_\star^0 \right\|_{\ell^2(\widetilde{G}_{Greek} \cup \{\star\})} \cdot \left\| \underline{u} \right\|_{\ell^2(\widetilde{G}_{Greek} \cup \{\star\})}
$$

$$
\leqslant \left( \frac{\delta \cdot K \cdot \widetilde{\mu}(\widetilde{G}_{Greek})}{\left( \widetilde{\mu}_\star + \widetilde{\mu}(\widetilde{G}_{Greek}) \right) \widetilde{\mu}_\star} \right) \cdot \left( \widetilde{\mu}_\star + \widetilde{\mu}(\widetilde{G}_{Greek}) \right) \cdot \left\| \underline{u} \right\|_{\ell^2(\widetilde{G}_{Greek} \cup \{\star\})}
$$

$$
+ 2 \frac{1}{\widetilde{\mu}_\star + \widetilde{\mu}(\widetilde{G}_{Greek})} \cdot \left( \delta \cdot K \cdot \widetilde{\mu}(\widetilde{G}_{Greek}) \right) \cdot \sqrt{\widetilde{\mu}_\star + \widetilde{\mu}(\widetilde{G}_{Greek})} \cdot \left\| \underline{u} \right\|_{\ell^2(\widetilde{G}_{Greek} \cup \{\star\})}
$$

Thus we find

$$
\|D\|_{op} \leqslant \delta \cdot K \cdot \widetilde{\mu}(\widetilde{G}_{Greek}) \cdot \left( \frac{1}{\widetilde{\mu}_\star} + 2 \frac{1}{\sqrt{\widetilde{\mu}_\star + \widetilde{\mu}(\widetilde{G}_{Greek})}} \right)
$$

In total, using (23) and (24), we find

$$
\left\| (u - J\widetilde{J}u) \right\|_{\ell^2(\widetilde{G})}
$$

$$
\leqslant \delta^{\frac{1}{2}} \cdot \left( \frac{C_{\widetilde{G}_{Greek} \cup \{\star\}} \cdot \sqrt{\widetilde{\mu}(\widetilde{G}_{Greek}) + \widetilde{\mu}_\star}}{\sqrt{\Omega}} \right) \cdot \sqrt{E_{\widetilde{G}}(u)}
$$

$$
+ \delta \cdot \left[ \frac{K \cdot \widetilde{\mu}(\widetilde{G}_{Greek})}{\min_{a \in \widetilde{G}_{Latin}} \mu_a} \right] \cdot \|u\|_{\ell^2(\widetilde{G})} + 2 \cdot \delta \cdot K \cdot \sqrt{|\widetilde{G}_{Greek} \cup \{\star\}|} \cdot \left( \frac{\max_{g \in \widetilde{G}_{Latin}} \widetilde{\mu}_g}{\min_{g \in \widetilde{G}_{Latin}} \widetilde{\mu}_g \cdot \max_{\alpha \in \widetilde{G}_{Greek} \cup \{\star\}} \widetilde{\mu}_\alpha^{\frac{3}{2}}} \right) \cdot \|u\|_{\ell^2(\widetilde{G})}
$$

$$
+ \delta \cdot K \cdot \widetilde{\mu}(\widetilde{G}_{Greek}) \cdot \left( \frac{1}{\widetilde{\mu}_\star} + 2 \frac{1}{\sqrt{\widetilde{\mu}_\star + \widetilde{\mu}(\widetilde{G}_{Greek})}} \right) \cdot \|u\|_{\ell^2(\widetilde{G})}
$$

and may hence set

$$\epsilon = \delta^{\frac{1}{2}} \cdot \left( \frac{C_{\widetilde{G}_{Greek} \cup \{\star\}} \cdot \sqrt{\widetilde{\mu}(\widetilde{G}_{Greek}) + \widetilde{\mu}_\star}}{\sqrt{\Omega}} \right)$$

$$+ \delta \cdot \left[ \frac{K \cdot \widetilde{\mu}(\widetilde{G}_{Greek})}{\min_{a \in \widetilde{G}_{Latin}} \mu_a} \right] + 2 \cdot \delta \cdot K \cdot \sqrt{|\widetilde{G}_{Greek} \cup \{\star\}|} \cdot \left( \frac{\max_{g \in \widetilde{G}_{Latin}} \widetilde{\mu}_g}{\min_{g \in \widetilde{G}_{Latin}} \widetilde{\mu}_g \cdot \max_{\alpha \in \widetilde{G}_{Greek} \cup \{\star\}} \widetilde{\mu}_\alpha^{\frac{3}{2}}} \right)$$

$$+ \delta \cdot K \cdot \widetilde{\mu}(\widetilde{G}_{Greek}) \cdot \left( \frac{1}{\widetilde{\mu}_\star} + 2 \frac{1}{\sqrt{\widetilde{\mu}_\star + \widetilde{\mu}(\widetilde{G}_{Greek})}} \right)$$

**Left-hand-side of (20):**

The left hand side of (20) is true with $\epsilon = 0$ by definition.

**Right-hand-side of (20):**

Let us thus check the right hand side of (20):

We have

$$(\widetilde{J}u - \widetilde{J}^1 u)(x) = \frac{1}{\mu_x} \langle \psi_x^\delta, u \rangle_{\ell^2(\widetilde{G})} - u(x).$$

We note

$$\|(\widetilde{J}u - \widetilde{J}^1 u)\|_{\ell^2(G)} \leqslant \left| \frac{1}{\mu_\star^\delta} \langle u, \psi_\star^\delta \rangle - u(\star) \right| \sqrt{\mu_\star^\delta} + \sqrt{\sum_{\substack{x \in G \\ g \neq \star}} \left| \frac{1}{\mu_x} \langle \psi_x^\delta, u \rangle_{\ell^2(\widetilde{G})} - u(x) \right|^2 \mu_x^\delta}. \quad (25)$$

We first deal with the left hand term of the estimate and note that for $x = \star$ we have

$$\mu_\star^\delta \leqslant \mu_\star^0 = \widetilde{\mu}_\star + \widetilde{\mu}(\widetilde{G}_{Greek})$$

and in the limit $\delta \to 0$ that

$$\left| \frac{1}{\mu_\star^\delta} \langle \psi_\star^\delta, u \rangle_{\ell^2(\widetilde{G})} - u(\star) \right| \longrightarrow \frac{1}{\widetilde{\mu}_\star + \widetilde{\mu}(\widetilde{G}_{Greek})} \left| \left[ \sum_{g \in \widetilde{G}_{Greek} \cup \{\star\}} u(g) \right] - u(\star) \right|$$

$$= \frac{1}{\widetilde{\mu}_\star + \widetilde{\mu}(\widetilde{G}_{Greek})} \left| \sum_{g \in \widetilde{G}_{Greek} \cup \{\star\}} u(g) - u(\star) \right|$$

$$\leqslant \frac{1}{\widetilde{\mu}_\star + \widetilde{\mu}(\widetilde{G}_{Greek})} \sum_{g \in \widetilde{G}_{Greek} \cup \{\star\}} |u(g) - u(\star)|$$

$$\leqslant \frac{1}{\widetilde{\mu}_\star + \widetilde{\mu}(\widetilde{G}_{Greek})} \sum_{g \in \widetilde{G}_{Greek} \cup \{\star\}} \delta^{\frac{1}{2}} \left( \frac{C_{\widetilde{G}_{Greek} \cup \{\star\}}}{\sqrt{\Omega}} \right) \sqrt{E_{\widetilde{G}}(u)}$$

$$\leqslant \delta^{\frac{1}{2}} \cdot \frac{|\widetilde{G}_{Greek} \cup \{\star\}|}{\widetilde{\mu}_\star + \widetilde{\mu}(\widetilde{G}_{Greek})} \left( \frac{C_{\widetilde{G}_{Greek} \cup \{\star\}}}{\sqrt{\Omega}} \right) \sqrt{E_{\widetilde{G}}(u)}$$

Here we applied Lemma J.7. Comparing the $\delta > 0$ and $\delta = 0$ terms, we find

$$\left| \frac{1}{\mu_\star^\delta} \langle \psi_\star^\delta, u \rangle_{\ell^2(\widetilde{G})} - \frac{1}{\mu^0_\star} \langle \psi_\star^0, u \rangle_{\ell^2(\widetilde{G})} \right|$$

$$\leqslant \frac{1}{\mu_\star^\delta} \left| \langle \psi_\star^\delta - \psi_\star^0, u \rangle_{\ell^2(\widetilde{G})} \right| + \left| \frac{1}{\mu_\star^\delta} - \frac{1}{\mu_\star^0} \right| \cdot \left| \langle \psi_\star^0, u \rangle_{\ell^2(\widetilde{G})} \right|$$

$$\leqslant \frac{1}{\widetilde{\mu}_\star} \|u\|_{\ell^2(\widetilde{G})} \cdot \|\vec{\zeta}_\star^\delta\|_{\ell^2(\widetilde{G}_{Greek})} + \left| \frac{1}{\mu_\star^\delta} - \frac{1}{\mu_\star^0} \right| \cdot \left( \widetilde{\mu}_\star + \widetilde{\mu}(\widetilde{G}_{Greek}) \right) \|u\|_{\ell^2(\widetilde{G}}$$

$$\leqslant \frac{K\delta}{\widetilde{\mu}_\star} \|u\|_{\ell^2(\widetilde{G})} + \left( \frac{K\delta}{\widetilde{\mu}_\star \left( \widetilde{\mu}_\star + \widetilde{\mu}(\widetilde{G}_{Greek}) \right)} \right) \cdot \left( \widetilde{\mu}_\star + \widetilde{\mu}(\widetilde{G}_{Greek}) \right) \|u\|_{\ell^2(\widetilde{G}}$$

$$= \delta \frac{2K}{\widetilde{\mu}_\star} \|u\|_{\ell^2(\widetilde{G})}.$$

Thus we have

$$\left| \frac{1}{\mu_\star^\delta} \langle u, \psi_\star^\delta \rangle - u(\star) \right| \sqrt{\mu_\star^\delta} \leqslant \delta^{\frac{1}{2}} \cdot \frac{|\widetilde{G}_{Greek} \cup \{\star\}|}{\sqrt{\widetilde{\mu}_\star + \widetilde{\mu}(\widetilde{G}_{Greek})}} \left( \frac{C_{\widetilde{G}_{Greek} \cup \{\star\}}}{\sqrt{\Omega}} \right) \sqrt{E_{\widetilde{G}}(u)}$$

$$+ \delta \frac{2K}{\widetilde{\mu}_\star} \|u\|_{\ell^2(\widetilde{G})}.$$

For the remaining term in (25) we note

$$\sqrt{\sum_{x \in \widetilde{G}_{Latin}} \left| \frac{1}{\mu_x} \langle \psi_x^\delta, u \rangle_{\ell^2(\widetilde{G})} - u(x) \right|^2}$$

$$\leqslant \sqrt{\sum_{x \in \widetilde{G}_{Latin}} \left| 1 - \frac{\widetilde{\mu}_x}{\mu_x^\delta} \right|^2 \cdot |u(x)|^2 \mu_x^\delta + \sum_{x \in \widetilde{G}_{Latin}} \left| \langle \underline{\psi}_x^\delta, \underline{u} \rangle_{\ell^2(\widetilde{G}_{Greek} \cup \{\star\})} \right| \sqrt{\mu_x^\delta}}$$

$$\leqslant \frac{K\delta}{\widetilde{\mu}_\star} \cdot \|u\|_{\ell^2(\widetilde{G})} + \sum_{x \in \widetilde{G}_{Latin}} \langle \underline{\psi}_x^\delta, |\underline{u}| \rangle_{\ell^2(\widetilde{G}_{Greek} \cup \{\star\})} \sqrt{\mu_x^\delta}$$

$$\leqslant \frac{K\delta}{\widetilde{\mu}_\star} \cdot \|u\|_{\ell^2(\widetilde{G})} + \|\vec{\zeta}_\star^\delta\|_{\ell^2(\widetilde{G}_{Greek})} \cdot \left[ \max_{x \in \widetilde{G}_{Latin}} \sqrt{\mu_x^\delta} \right] \|\underline{u}\|_{\ell^2(\widetilde{G})}$$

$$\leqslant \frac{K\delta}{\widetilde{\mu}_\star} \cdot \|u\|_{\ell^2(\widetilde{G})} + \delta K \widetilde{\mu}(\widetilde{G}_{Greek}) \cdot \left[ \max_{x \in \widetilde{G}_{Latin}} \sqrt{\widetilde{\mu}_x + \delta K \widetilde{\mu}(\widetilde{G}_{Greek})} \right] \|\underline{u}\|_{\ell^2(\widetilde{G})}$$

$$\leqslant \frac{K\delta}{\widetilde{\mu}_\star} \cdot \|u\|_{\ell^2(\widetilde{G})} + \delta K \widetilde{\mu}(\widetilde{G}_{Greek}) \cdot \left[ \sqrt{\max_{x \in \widetilde{G}_{Latin}} \widetilde{\mu}_x} + \sqrt{\delta K \widetilde{\mu}(\widetilde{G}_{Greek})} \right] \|\underline{u}\|_{\ell^2(\widetilde{G})}.$$

**Equation (21):**

It finally only remains to prove the energy differences of (21) and establish

$$|E_{\widetilde{G}}(J^1 f, u) - E_G(f, \widetilde{J}^1 u)| \leqslant \epsilon \cdot \sqrt{\|f\|^2 + E_G(f)} \cdot \sqrt{\|u\|^2 + E_{\widetilde{G}}(u)}.$$

We note that the (unique) operator associated to the energy $E_G$ via

$$E_G(g, f) = \langle g, \Delta_G f \rangle_{\ell^2(G)}$$

is given by

$$(\Delta_G f)(x) = \frac{1}{\mu_x} \sum_{y \sim_G x} W_{xy}(f(x) - f(y)).$$

Here the notation "$y \sim_G x$" signifies that nodes $x$ and $y$ are connected **within** $G$ through edges with **positive** edge-weights $W_{xy} > 0$.
Similarly the operator associated to $E_{\widetilde{G}}$ via

$$E_{\widetilde{G}}(v, u) = \langle v, \Delta_{\widetilde{G}} u \rangle_{\ell^2(\widetilde{G})}$$

is given by

$$(\Delta_{\widetilde{G}} u)(x) = \frac{1}{\widetilde{\mu}_x} \sum_{y \sim_{\widetilde{G}} x} \widetilde{W}_{xy}(u(x) - u(y))$$

with the equivalence relation $\sim_{\widetilde{G}}$ precisely signifying that $\widetilde{W}_{xy} > 0$.
As before. let us denote by $\underline{u} \in \ell^2(\widetilde{G}_{Greek} \cup \{\star\})$ the restriction of an element $u \in \ell^2(\widetilde{G}$ to $\widetilde{G}_{Greek} \cup \{\star\}$.
We note

$$E_G(\underline{\psi_x}, \underline{u}) = \langle \underline{\psi_x}, \Delta_G \underline{u} \rangle_{\ell^2(G)} = \sum_{y \sim_G x} W_{xy}(u(x) - u(y))$$

on the smaller graph $G$. For the graph $\widetilde{G}$ we find

$$E_{\widetilde{G}}(\psi_x, u) = \sum_{y \sim_{\widetilde{G}} x} \widetilde{W}_{xy}(u(x) - u(y))$$
$$+ \sum_{\alpha \in \widetilde{G}_{Greek}} \vec{\eta}_x^\delta(\alpha) \sum_{y \sim_{\widetilde{G}} \alpha} \widetilde{W}_{\alpha y}(u(\alpha) - u(y)).$$

Remembering that we have

$$J^1 f = Jf = \sum_{x \in G} f(x)\psi_x \quad and \quad (\widetilde{J}^1 u)(x) = u(x),$$

we note

$$\left| E_{\widetilde{G}}(J^1 f, u) - E_G(f, \widetilde{J}^1 u) \right| \leqslant \left| \sum_{x \in \widetilde{G}_{Latin} \cup \{\star\}} \overline{f}(x) \left[ E_{\widetilde{G}}(\psi_x, u) - E_G(\underline{\psi_x}, \underline{u}) \right] \right|$$

$$\leqslant \left( \frac{1}{\sqrt{\min_{x \in \widetilde{G}_{Latin} \cup \{\star\}} \widetilde{\mu}_x}} \right) \cdot \|f\|_{\ell^2(G)} \cdot \sum_{x \in \widetilde{G}_{Latin} \cup \{\star\}} \left| E_{\widetilde{G}}(\psi_x, u) - E_G(\underline{\psi_x}, \underline{u}) \right|$$

Let us first bound the terms corresponding to $x \neq \star$: We have

$$E_G(\underline{\psi_x}, \underline{u}) = \sum_{\substack{y \sim_G x \\ y \neq \star}} W_{xy}(u(x) - u(y)) + W_{x\star}(u(x) - u(\star))$$
$$= \sum_{\substack{y \sim_G x \\ y \neq \star}} \widetilde{W}_{xy}(u(x) - u(y)) + W_{x\star}(u(x) - u(\star)),$$

as well as

$$
\begin{aligned}
E_{\widetilde{G}}(\psi_x, u) &= \sum_{y \sim_{\widetilde{G}} x} \widetilde{W}_{xy}(u(x) - u(y)) + \sum_{\alpha \in \widetilde{G}_{Greek}} \vec{\eta}^{\delta}_x(\alpha) \sum_{y \sim_{\widetilde{G}} x} W_{\alpha y}(u(\alpha) - u(y)) \\
&= \sum_{\substack{y \sim_G x \\ y \neq \star}} \widetilde{W}_{xy}(u(x) - u(y)) \\
&\quad + \widetilde{W}_{x\star}(u(x) - u(\star)) \\
&\quad + \sum_{\alpha \in \widetilde{G}_{Greek}} \widetilde{W}_{x\alpha}(u(x) - u(\alpha)) \\
&\quad + \sum_{\alpha \in \widetilde{G}_{Greek}} \vec{\eta}^{\delta}_x(\alpha) \sum_{y \sim_{\widetilde{G}} x} \widetilde{W}_{\alpha y}(u(\alpha) - u(y)).
\end{aligned}
$$

Hence (for $x \neq \star$)

$$
\begin{aligned}
E_G(\underline{\psi_x}, \underline{u}) - E_{\widetilde{G}}(\psi_x, u) &= W_{x\star}(u(x) - u(\star)) - \widetilde{W}_{x\star}(u(x) - u(\star)) \\
&\quad - \sum_{\alpha \in \widetilde{G}_{Greek}} \widetilde{W}_{x\alpha}(u(x) - u(\alpha)) \\
&\quad - \sum_{\alpha \in \widetilde{G}_{Greek}} \vec{\eta}^{\delta}_x(\alpha) \sum_{y \sim_{\widetilde{G}} x} W_{\alpha y}(u(\alpha) - u(y)) \\
&= \left( \sum_{\alpha \in \widetilde{G}_{Greek}} \widetilde{W}_{x\alpha} \right)(u(x) - u(\star)) \\
&\quad - \sum_{\alpha \in \widetilde{G}_{Greek}} \widetilde{W}_{x\alpha}(u(x) - u(\alpha)) \\
&\quad - \sum_{\alpha \in \widetilde{G}_{Greek}} \vec{\eta}^{\delta}_x(\alpha) \sum_{y \sim_{\widetilde{G}} \alpha} W_{\alpha y}(u(\alpha) - u(y)) \\
&= \underbrace{\left( \sum_{\alpha \in \widetilde{G}_{Greek}} \widetilde{W}_{x\alpha}(u(\alpha) - u(\star)) \right)}_{=:I_x} \\
&\quad - \underbrace{\left( \sum_{\alpha \in \widetilde{G}_{Greek}} \vec{\eta}^{\delta}_x(\alpha) \sum_{y \sim_{\widetilde{G}} \alpha} \widetilde{W}_{\alpha y}(u(\alpha) - u(y)) \right)}_{=:II_x}.
\end{aligned}
\tag{26}
$$

For $I_x$ we find – using Lemma J.7 – that

$$
|I_x| \leqslant \left( \sum_{\alpha \in \widetilde{G}_{Greek}} \widetilde{W}_{x\alpha} \right) \cdot \delta^{\frac{1}{2}} \left( \frac{C_{\widetilde{G}_{Greek} \cup \{\star\}}}{\sqrt{\Omega}} \right) \sqrt{E_{\widetilde{G}}(u)}
$$

and hence

$$
\sum_{\substack{x \in G \\ x \neq \star}} |I_x| \leqslant \left( \sum_{\substack{x \in G \\ x \neq \star}} \sum_{\alpha \in \widetilde{G}_{Greek}} \widetilde{W}_{x\alpha} \right) \cdot \delta^{\frac{1}{2}} \left( \frac{C_{\widetilde{G}_{Greek} \cup \{\star\}}}{\sqrt{\Omega}} \right) \sqrt{E_{\widetilde{G}}(u)}.
$$

To bound $|II_x|$ we note

$$
\left| \sum_{\alpha \in \widetilde{G}_{Greek}} \vec{\eta}_x^\delta(\alpha) \sum_{y \sim_{\widetilde{G}} \alpha} \widetilde{W}_{\alpha y}(u(\alpha) - u(y)) \right| = \left| \sum_{\alpha \in \widetilde{G}_{Greek}} \vec{\eta}_x^\delta(\alpha) \sum_{y \sim_{\widetilde{G}} \alpha} \sqrt{\widetilde{W}_{\alpha y}} \sqrt{\widetilde{W}_{\alpha y}}(u(\alpha) - u(y)) \right|
$$

$$
= \left| \sum_{\alpha \in \widetilde{G}_{Greek}} \vec{\eta}_x^\delta(\alpha) \left[ \sum_{y \sim_{\widetilde{G}} \alpha} \widetilde{W}_{y\alpha} \right]^{\frac{1}{2}} \cdot \left[ \sum_{y \sim_{\widetilde{G}} \alpha} \widetilde{W}_{y\alpha} |u(\alpha) - u(y)|^2 \right]^{\frac{1}{2}} \right|
$$

$$
\leqslant \sum_{\alpha \in \widetilde{G}_{Greek}} \vec{\eta}_x^\delta(\alpha) \cdot \left[ \sum_{y \sim_{\widetilde{G}} \alpha} \widetilde{W}_{y\alpha} \right]^{\frac{1}{2}} \cdot \sqrt{E_{\widetilde{G}}(u)}.
$$

Thus we find – using Cauchy-Schwarz – that

$$
\sum_{\substack{x \in G \\ x \neq \star}} |II_x| \leqslant \sum_{\substack{x \in G \\ x \neq \star}} \sum_{\alpha \in \widetilde{G}_{Greek}} \vec{\eta}_x^\delta(\alpha) \cdot \left[ \sum_{y \sim_{\widetilde{G}} \alpha} \widetilde{W}_{y\alpha} \right]^{\frac{1}{2}} \cdot \sqrt{E_{\widetilde{G}}(u)}
$$

$$
= \sum_{\alpha \in \widetilde{G}_{Greek}} \vec{\zeta}_\star^\delta(\alpha) \cdot \left[ \sum_{y \sim_{\widetilde{G}} \alpha} \widetilde{W}_{y\alpha} \right]^{\frac{1}{2}} \cdot \sqrt{E_{\widetilde{G}}(u)}
$$

$$
\leqslant \frac{1}{\min\limits_{\alpha \in \widetilde{G}_{Greek}} \sqrt{\widetilde{\mu}_\alpha}} \cdot \|\vec{\zeta}_\star^\delta\|_{\ell^2(\widetilde{G}_{Greek})} \cdot \left[ \sum_{\alpha \in \widetilde{G}_{Greek}} \sum_{y \sim_{\widetilde{G}} \alpha} \widetilde{W}_{y\alpha} \right]^{\frac{1}{2}} \cdot \sqrt{E_{\widetilde{G}}(u)}
$$

$$
\leqslant \frac{1}{\min\limits_{\alpha \in \widetilde{G}_{Greek}} \sqrt{\widetilde{\mu}_\alpha}} \cdot K\delta \cdot \left[ \sum_{\alpha \in \widetilde{G}_{Greek}} \sum_{y \sim_{\widetilde{G}} \alpha} \widetilde{W}_{y\alpha} \right]^{\frac{1}{2}} \cdot \sqrt{E_{\widetilde{G}}(u)}
$$

$$
\leqslant \frac{1}{\min\limits_{\alpha \in \widetilde{G}_{Greek}} \sqrt{\widetilde{\mu}_\alpha}} \cdot K\delta \cdot \sqrt{\sum_{\alpha \in \widetilde{G}_{Greek}} \widetilde{d}_\alpha} \cdot \sqrt{E_{\widetilde{G}}(u)}.
$$

Here we denoted by $\widetilde{d}_\alpha$ the degree of the node $\alpha$. We further note

$$
\sum_{\alpha \in \widetilde{G}_{Greek}} \widetilde{d}_\alpha = \sum_{\alpha \in \widetilde{G}_{Greek}} \sum_{y \in \widetilde{G}_{Latin}} \widetilde{W}_{\alpha y} + \frac{1}{\delta} \sum_{\alpha \in \widetilde{G}_{Greek}} \sum_{y \in \widetilde{G}_{Greek} \cup \{\star\}} \omega_{\alpha y}.
$$

Writing

$$
\widetilde{d}_{int}^1 := \sum_{\alpha \in \widetilde{G}_{Greek}} \sum_{y \in \widetilde{G}_{Greek} \cup \{\star\}} \omega_{\alpha y}
$$

for the sum of 'internal' degrees of greek nodes within *Greek* $\cup \{\star\}$ at $\delta = 1$ and

$$
d_{external} := \sum_{\alpha \in \widetilde{G}_{Greek}} \sum_{y \in \widetilde{G}_{Latin}} \widetilde{W}_{\alpha y}
$$

for the 'total connection strength' between the Greek and Latin sector, we thus find

$$
\sum_{\substack{x \in G \\ x \neq \star}} |II_x| \leqslant [\sqrt{\widetilde{d}_{int}^1} \cdot \sqrt{\delta} + \sqrt{d_{external}} \cdot \delta] \frac{K}{\min\limits_{\alpha \in \widetilde{G}_{Greek}} \sqrt{\widetilde{\mu}_\alpha}} \cdot \sqrt{E_{\widetilde{G}}(u)}.
$$

It remains to bound the $x = \star$ term in (26). To this end we note

$$
E_G(\underline{\psi}_\star, \underline{u}) = \sum_{y \sim_G \star} W_{\star y}(u(\star) - u(y))
$$

and

$$E_{\widetilde{G}}(\psi_\star, u) = \sum_{y \sim \widetilde{G}\star} \widetilde{W}_{\star y}(u(\star) - u(y))$$
$$+ \sum_{\alpha \in \widetilde{G}_{Greek}} \vec{\zeta}_\star^\delta(\alpha) \sum_{y \sim \widetilde{G}\alpha} \widetilde{W}_{y\alpha}(u(\alpha) - u(y)).$$

For the difference of the energy forms we thus find

$$E_G(\underline{\psi_\star}, \underline{u}) - E_{\widetilde{G}}(\psi_\star, u) = \sum_{y \sim G\star} W_{\star y}(u(\star) - u(y))$$
$$- \sum_{y \sim \widetilde{G}\star} \widetilde{W}_{\star y}(u(\star) - u(y)) - \sum_{\alpha \in \widetilde{G}_{Greek}} \vec{\eta}_\star^\delta(\alpha) \sum_{y \sim \widetilde{G}\alpha} \widetilde{W}_{y\alpha}(u(\alpha) - u(y))$$
$$= \sum_{y \sim G\star} W_{\star y}(u(\star) - u(y))$$
$$- \sum_{y \sim \widetilde{G}\star} \widetilde{W}_{\star y}(u(\star) - u(y)) - \sum_{\alpha \in \widetilde{G}_{Greek}} \vec{\eta}_\star^\delta(\alpha) \sum_{y \sim \widetilde{G}\alpha} \widetilde{W}_{y\alpha}(u(\alpha) - u(y))$$
$$+ \sum_{\alpha \in \widetilde{G}_{Greek}} \sum_{y \sim \widetilde{G}\alpha} \widetilde{W}_{y\alpha}(u(\alpha) - u(y)) - \sum_{\alpha \in \widetilde{G}_{Greek}} \sum_{y \sim \widetilde{G}\alpha} \widetilde{W}_{y\alpha}(u(\alpha) - u(y)).$$

We have

$$\sum_{\alpha \in \widetilde{G}_{Greek}} \sum_{y \sim \widetilde{G}\alpha} \widetilde{W}_{y\alpha}(u(\alpha) - u(y)) = \sum_{\alpha \in \widetilde{G}_{Greek}} \widetilde{W}_{\star\alpha}(u(\alpha) - u(\star)) + \sum_{\substack{y \sim \widetilde{G}\alpha \\ y \in \widetilde{G}_{Latin}}} \widetilde{W}_{y\alpha}(u(\alpha) - u(y))$$
$$+ \underbrace{\sum_{\alpha \in \widetilde{G}_{Greek}} \sum_{\substack{y \sim \widetilde{G}\alpha \\ y \in \widetilde{G}_{Greek}}} \widetilde{W}_{y\alpha}(u(\alpha) - u(y))}_{=0}.$$

with the last term vanishing by symmetry. This implies

$$
\begin{aligned}
E_G(\underline{\psi_\star}, \underline{u}) - E_{\widetilde{G}}(\psi_\star, u) &= \sum_{\alpha \in \widetilde{G}_{Greek}} (1 - \vec{\eta}_\star^\delta(\alpha)) \sum_{y \sim_{\widetilde{G}} \alpha} \widetilde{W}_{y\alpha}(u(\alpha) - u(y)) \\
&+ \sum_{\substack{y \sim_G \star \\ y \in \widetilde{G}_{Latin}}} \left( \sum_{\alpha \in \widetilde{G}_{Greek} \cup \{\star\}} \widetilde{W}_{y\alpha} \right) (u(\star) - u(y)) \\
&- \sum_{y \sim_{\widetilde{G}} \star} \widetilde{W}_{\star y}(u(\star) - u(y)) \\
&- \sum_{\alpha \in \widetilde{G}_{Greek}} \widetilde{W}_{\star \alpha}(u(\alpha) - u(\star)) - \sum_{\alpha \in \widetilde{G}_{Greek}} \sum_{\substack{y \sim_{\widetilde{G}} \alpha \\ y \in \widetilde{G}_{Latin}}} \widetilde{W}_{\alpha y}(u(\alpha) - u(y)) \\
&= \sum_{\alpha \in \widetilde{G}_{Greek}} (1 - \vec{\eta}_\star^\delta(\alpha)) \sum_{y \sim_{\widetilde{G}} \alpha} \widetilde{W}_{y\alpha}(u(\alpha) - u(y)) \\
&+ \sum_{\substack{y \sim_{\widetilde{G}} \star \\ y \in \widetilde{G}_{Latin}}} \left( \sum_{\alpha \in \widetilde{G}_{Greek} \cup \{\star\}} \widetilde{W}_{y\alpha} \right) (u(\star) - u(y)) \\
&- \sum_{\substack{y \sim_G \star \\ y \in \widetilde{G}_{Greek}^\star}} \widetilde{W}_{\star y}(u(\star) - u(y)) \\
&- \sum_{\alpha \in \widetilde{G}_{Greek}} \widetilde{W}_{\star \alpha}(u(\alpha) - u(\star)) - \sum_{\alpha \in \widetilde{G}_{Greek}} \sum_{\substack{y \sim_{\widetilde{G}} \alpha \\ y \in \widetilde{G}_{Latin}}} \widetilde{W}_{\alpha y}(u(\alpha) - u(y)) \\
&= \sum_{\alpha \in \widetilde{G}_{Greek}} (1 - \vec{\eta}_\star^\delta(\alpha)) \sum_{y \sim_{\widetilde{G}} \alpha} \widetilde{W}_{y\alpha}(u(\alpha) - u(y)) \\
&+ \sum_{\substack{y \sim_G \star \\ y \in \widetilde{G}_{Latin}}} \left( \sum_{\alpha \in \widetilde{G}_{Greek} \cup \{\star\}} \widetilde{W}_{y\alpha} \right) (u(\star) - u(y)) \\
&- \sum_{\alpha \in \widetilde{G}_{Greek}} \sum_{\substack{y \sim_{\widetilde{G}} \alpha \\ y \in \widetilde{G}_{Latin}}} \widetilde{W}_{\alpha y}(u(\alpha) - u(y)).
\end{aligned}
$$

Continuing, we find

$$
\begin{aligned}
E_G(\underline{\psi_\star}, \underline{u}) - E_{\widetilde{G}}(\psi_\star, u) &= \sum_{\alpha \in \widetilde{G}_{Greek}} (1 - \vec{\eta}_\star^\delta(\alpha)) \sum_{y \sim_{\widetilde{G}} \alpha} \widetilde{W}_{y\alpha}(u(\alpha) - u(y)) \\
&+ \sum_{\alpha \in \widetilde{G}_{Greek}} \sum_{\substack{y \sim_G \star \\ y \in \widetilde{G}_{Latin}}} \widetilde{W}_{y\alpha}(u(\star) - u(y)) \\
&- \sum_{\alpha \in \widetilde{G}_{Greek}} \sum_{\substack{y \sim_{\widetilde{G}} \alpha \\ y \in \widetilde{G}_{Latin}}} \widetilde{W}_{\alpha y}(u(\alpha) - u(y)) \\
&= \sum_{\alpha \in \widetilde{G}_{Greek}} (1 - \vec{\eta}_\star^\delta(\alpha)) \sum_{y \sim_{\widetilde{G}} \alpha} \widetilde{W}_{y\alpha}(u(\alpha) - u(y)) \\
&+ \sum_{\alpha \in \widetilde{G}_{Greek}} \sum_{\substack{y \sim_{\widetilde{G}} \alpha \\ y \in \widetilde{G}_{Latin}}} \widetilde{W}_{y\alpha}(u(\star) - u(y)) \\
&- \sum_{\alpha \in \widetilde{G}_{Greek}} \sum_{\substack{y \sim_{\widetilde{G}} \alpha \\ y \in \widetilde{G}_{Latin}}} \widetilde{W}_{y\alpha}(u(\alpha) - u(y)).
\end{aligned}
$$

This – in turn – we can write as

$$E_G(\underline{\psi}_\star, \underline{u}) - E_{\widetilde{G}}(\psi_\star, u) = I + II$$

with

$$I := \sum_{\alpha \in \widetilde{G}_{Greek}} \vec{\zeta}_\star^\delta(\alpha) \sum_{y \sim_{\widetilde{G}} \alpha} \widetilde{W}_{y\alpha}(u(\alpha) - u(y)),$$

and

$$II := \sum_{\alpha \in \widetilde{G}_{Greek}} \sum_{\substack{y \sim_{\widetilde{G}} \alpha \\ y \in \widetilde{G}_{Latin}}} \widetilde{W}_{y\alpha}(u(\star) - u(\alpha)).$$

For the first term, we find

$$|I| \leqslant \frac{\|\vec{\zeta}_\star^\delta\|}{\min_{\alpha \in \widetilde{G}_{Greek}} \sqrt{\widetilde{\mu}_\alpha}} \cdot \sqrt{\sum_{\alpha \in \widetilde{G}_{Greek}} \widetilde{d}_\alpha} \cdot \sqrt{E_{\widetilde{G}}(u)}$$

$$\leqslant [\sqrt{\widetilde{d}_{int}^1} \cdot \sqrt{\delta} + \sqrt{d_{external}} \cdot \delta] \frac{K}{\min_{\alpha \in \widetilde{G}_{Greek}} \sqrt{\widetilde{\mu}_\alpha}} \cdot \sqrt{E_{\widetilde{G}}(u)}.$$

For the second term we note

$$|II| \leqslant \sum_{\alpha \in \widetilde{G}_{Greek}} \sum_{\substack{y \sim_{\widetilde{G}} \alpha \\ y \in \widetilde{G}_{Latin}}} \widetilde{W}_{y\alpha}|u(\star) - u(\alpha)|$$

$$\leqslant \sqrt{\delta} \cdot \sum_{\alpha \in \widetilde{G}_{Greek}} \sum_{\substack{y \sim_{\widetilde{G}} \alpha \\ y \in \widetilde{G}_{Latin}}} \widetilde{W}_{y\alpha} \left( \frac{C_{\widetilde{G}_{Greek} \cup \{\star\}}}{\sqrt{\Omega}} \right) \sqrt{E_{\widetilde{G}}(u)}$$

$$= \sqrt{\delta} \cdot d_{external} \cdot \left( \frac{C_{\widetilde{G}_{Greek} \cup \{\star\}}}{\sqrt{\Omega}} \right) \sqrt{E_{\widetilde{G}}(u)}.$$

## K  PROOF OF THEOREM 5.7

We prove the following theorem:

**Theorem K.1.** In the setting of Theorem 5.6 denote by $T$ ($\widetilde{T}$) adjacency matrices or normalized graph Laplacians on $\ell^2(G)$ ($\ell^2(G)$). There are no functions $\eta_1, \eta_2 : [0,1] \to \mathbb{R}_{\geqslant 0}$ with $\eta_i(\delta) \to 0$ as $\delta \to 0$ ($i = 1, 2$), families of identification operators $J^\delta, \widetilde{J}^\delta$ and $\omega \in \mathbb{C}$ so that $J^\delta$ and $\widetilde{J}^\delta$ are $\eta_1(\delta)$-quasi-unitary with respect to $\widetilde{T}, T$ and $\omega$ while the operators $\widetilde{T}$ and $T$ remain $\omega$-$\eta_2(\delta)$ close.

*Proof.* We prove these two result through contradiction on a graph with two vertices and one edge with weight $1/\delta$, which we collapse.
First fix $T$ ($\widetilde{T}$) to be the adjacency matrices

$$\widetilde{W} = \begin{pmatrix} 0 & \frac{1}{\delta} \\ \frac{1}{\delta} & 0 \end{pmatrix}$$

and

$$W = 0.$$

The eigenvectors and eigenvalues of $\widetilde{W}$ are given by $\{-\frac{1}{\delta}, \frac{1}{\delta}\}$ and

$$v_- = \begin{pmatrix} 1 \\ -1 \end{pmatrix} \quad and \quad v_+ = \begin{pmatrix} 1 \\ 1 \end{pmatrix}.$$

Denote the orthogonal projections onto the corresponding eigenspaces by $\{P_-, P_+\}$. Take the function $g$ to be defined as

$$g(\lambda) := 1 - \frac{i}{i - \lambda}.$$

Then since $g(0) = 0$ we have

$$g(W) = 0.$$

Furthermore we have

$$\begin{aligned}
g(\widetilde{W}) &= \left[1 - \frac{i}{i - \frac{1}{\delta}}\right] P_+ + \left[1 - \frac{i}{i + \frac{1}{\delta}}\right] P_- \\
&= P_+ + P_- - \delta \frac{1}{\delta + i} P_+ - \delta \frac{1}{\delta - i} P_- \\
&= Id - \delta \frac{1}{\delta + i} P_+ - \delta \frac{1}{\delta - i} P_- \\
&= Id \left[1 - \delta \frac{1}{\delta + i}\right] + \left[\delta \frac{1}{\delta + i} - \delta \frac{1}{\delta - i}\right] P_- \\
&= Id \left[1 - \delta \frac{1}{\delta + i}\right] - \left[\delta \frac{2i}{\delta^2 + 1}\right] P_-
\end{aligned}$$

We are interested in

$$\left\|g(\widetilde{W}) J^\delta - J^\delta g(W)\right\|_{op} = \left\|g(\widetilde{W}) J^\delta\right\|_{op} = \left\|J^\delta - \delta \left[\frac{1}{\delta + i} P_+ + \frac{1}{\delta - i} P_-\right] J^\delta\right\|_{op}.$$

Assuming

$$\left\|g(\widetilde{W}) J^\delta - J^\delta g(W)\right\|_{op} = \left\|g(\widetilde{W}) J^\delta\right\|_{op} \leqslant \eta_1(\delta)$$

we also find

$$\left|\|J^\delta\|_{op} \left(\frac{i}{\delta + i}\right) - \|J^\delta P_-\|_{op} \left(\frac{\delta 2i}{\delta^2 + 1}\right)\right| \leqslant \eta_1(\delta).$$

Thus also

$$\|J^\delta\|_{op} \left(\frac{i}{\delta + i}\right) \leqslant \eta_1(\delta) + \|J^\delta P_-\|_{op} \left(\frac{\delta 2i}{\delta^2 + 1}\right).$$

Taking the limit and using the condition $\|J^\delta\|_{op} \leqslant 2$, we find that $\|J^\delta\| \to 0$ as $\delta \to 0$. Since we demand

$$\|(J - \widetilde{J}^*)\|_{op} \leqslant \eta_2(\delta)$$

with

$$\lim_{\delta \to 0} \eta_2(\delta) = 0,$$

we also find $\|\widetilde{J}\|_{op} = \|\widetilde{J}^*\|_{op} \to 0$. Next we note that we have

$$R_\omega = \frac{1}{\omega}$$

and demand

$$\|(Id - \widetilde{J}^\delta J^\delta) R_\omega\|_{op} \to 0.$$

However

$$\|(Id - \widetilde{J}^\delta J^\delta) R_\omega\|_{op} = \frac{1}{|\omega|} \|Id - \widetilde{J}^\delta J^\delta\|_{op} \geqslant \frac{1}{|\omega|} (1 - \|\widetilde{J}^*\|_{op} \|J\|_{op}) \to \frac{1}{|\omega|} > 0.$$

Thus we have our contradiction.

Hence let us now choose $T$ ($\widetilde{T}$) as the normalized graph Laplacians associated to the adjacency matrices $W$ ($\widetilde{W}$) from above. We thus have

$$\mathscr{L} = 0$$

and

$$\widetilde{\mathscr{L}} = \begin{pmatrix} 1 & -1 \\ -1 & 1 \end{pmatrix}.$$

The eigenvectors and eigenvalues of $\widetilde{\mathscr{L}}$ are given by $\{0, 2\}$ and

$$v_0 = \begin{pmatrix} 1 \\ 1 \end{pmatrix} \quad and \quad v_2 = \begin{pmatrix} 1 \\ -1 \end{pmatrix}.$$

Denote the orthogonal projections onto the corresponding eigenspaces by $\{P_0, P_2\}$. Then

$$\widetilde{\mathscr{L}} = 2P_2.$$

Chose a function $g$ such that $g(0) = 0$ and without loss of generality assume $g(2) = 1$. Then

$$0 \longleftarrow \left\| g(\widetilde{\mathscr{L}})J^\delta - J^\delta g(\mathscr{L}) \right\|_{op} = \left\| P_2 J^\delta \right\|_{op}. \tag{27}$$

Next we consider the demand

$$\|(Id - J^\delta \widetilde{J}^\delta)\widetilde{R}_\omega u\| \leqslant \eta_3 \cdot \|u\|. \tag{28}$$

Since $(\widetilde{\mathscr{L}} - \omega Id)$ is bijective, (28) is implies

$$\|(Id - J^\delta \widetilde{J}^\delta)v\| \leqslant \eta_3(\delta) \cdot [|\omega|\|v\| + \|\widetilde{\mathscr{L}}\| \cdot \|v\|] = \eta_3(\delta) \cdot [|\omega| + 2] \cdot \|v\|. \tag{29}$$

upon writing

$$u = (\widetilde{\mathscr{L}} - \omega Id)v.$$

We also write

$$v = \begin{pmatrix} v_a \\ v_b \end{pmatrix}.$$

We write

$$\widetilde{J}^\delta = \begin{pmatrix} a^\delta \\ b^\delta \end{pmatrix}^T$$

and

$$J^\delta = \eta_4(\delta) \cdot \begin{pmatrix} 1 \\ -1 \end{pmatrix} + f(\delta) \cdot \begin{pmatrix} 1 \\ 1 \end{pmatrix}.$$

From (27), we know that

$$\lim_{\delta \to 0} \eta_4(\delta) = 0,$$

but we do not yet know the behaviour of $f(\cdot), a^\delta, b^\delta$ as $\delta \to 0$.

With the above notation, we find from (29) that

$$\|(Id - J^\delta \widetilde{J}^\delta)v\| = \left\| \begin{pmatrix} v_a - f(\delta)a^\delta v_a - f(\delta)b^\delta v_b \\ v_b - f(\delta)a^\delta v_a - f(\delta)b^\delta v_b \end{pmatrix} - \eta_4(\delta) \left\langle \begin{pmatrix} v_a \\ v_b \end{pmatrix}, \begin{pmatrix} a^\delta \\ b^\delta \end{pmatrix} \right\rangle \begin{pmatrix} 1 \\ 1 \end{pmatrix} \right\|$$

$$\geqslant \left\| \begin{pmatrix} v_a - f(\delta)a^\delta v_a - f(\delta)b^\delta v_b \\ v_b - f(\delta)a^\delta v_a - f(\delta)b^\delta v_b \end{pmatrix} \right\| - \eta_4(\delta) \cdot 4 \cdot \|v\|.$$

Thus, combining this result with (29), we know that

$$\left\| \begin{pmatrix} v_a - f(\delta)a^\delta v_a - f(\delta)b^\delta v_b \\ v_b - f(\delta)a^\delta v_a - f(\delta)b^\delta v_b \end{pmatrix} \right\| \longrightarrow 0.$$

Thus, since both entries of the above vector need to tend to zero, we need both

$$f(\delta) \cdot a^\delta \to 1 \quad and \quad f(\delta) \cdot b^\delta \to 0$$

as well as

$$f(\delta) \cdot a^\delta \to 0 \quad and \quad f(\delta) \cdot b^\delta \to 1$$

which yields the desired contradiction.

$\square$

## L  PROOF OF THEOREM 5.8

We first note how the graph Laplacian $\Delta_{G_N}$ as we have defined it, is consistent with the underlying positive (in the sense of non-negative eigenvalues) Laplacian

$$" - \Delta_{S^1} = -\frac{\partial^2}{\partial\theta^2}"$$

on the unit circle $S^1$.

To this end, fix $0 < h << 1$. Fix a point $x \in S^1$. For any suitable function $f$ – by means of Taylor expansions – we may write

$$f(x + h) = f(x) + h \cdot [\partial_\theta f](x) + \frac{h^2}{2} \cdot [\Delta_{S^1} f](x) + \mathcal{O}(h^3)$$

$$f(x - h) = f(x) - h \cdot [\partial_\theta f](x) + \frac{h^2}{2} \cdot [\Delta_{S^1} f](x) + \mathcal{O}(h^3).$$

Adding these two terms, we find

$$[-\Delta_{S^1} f](x) = \frac{2f(x) - f(x + h) - f(x - h)}{h^2} + \mathcal{O}(h).$$

This motivates setting our edgeweights on $G_N$ to $1/h^2$ with $h = 2\pi/N$ the distance between evenly spaced nodes on the unit-circle $S^1$.

**Remark L.1.** It should be noted that this consistency property – while given a heuristic to choose weights – does not (immediately) imply 'convergence' of $\Delta_{G_N}$ to $-\Delta_{S^1}$ in the sense needed to e.g. apply Levie et al. (2019a). As our proof of Theorem L proceeds completely without reference to the limit-circle, we do not proceed beyond the above heuristic in investigating in what (relevant) sense $\Delta_{G_N}$ approximates $-\Delta_{S^1}$.

We thus now want to prove the following result:

**Theorem L.2.** In the large graph setting of Section 5.2 choose all node-weights equal to one and $N$ to be odd for definiteness. There exists constants $K_1, K_2 = \mathcal{O}(1)$ so that for each $N \geqslant 1$, there exist identification operators $J, \widetilde{J}$ mapping between $\ell^2(G_N)$ and $\ell^2(G_{N+1})$ so that $J$ and $\widetilde{J}$ are $(K_1/N)$-quasi-unitary with respect to $\Delta_{G_N}, \Delta_{G_{N+1}}$ and $\omega = (-1)$. Furthermore, the operators $\Delta_{G_N}$ and $\Delta_{G_{N+1}}$ are $(-1)$-$(K_2/N)$ close with identification operator $J$.

*Proof.* We first note that the normalized eigenvectors of $G_N$ are given by

$$\phi_k^N(x) = \frac{1}{\sqrt{N}} e^{i \frac{2\pi k}{N} x} \quad 0 \leqslant k < N.$$

The corresponding eigenvalues are easily found to be

$$\lambda_k^N = \frac{N^2}{\pi^2} \sin^2\left(\frac{\pi}{N} \cdot k\right).$$

For definiteness, we have assumed $N$ to be odd, so that $(N + 1)$ is even. We define the identification operator $J : \ell^2(G_N) \to \ell^2(G_{N+1})$ via

$$J(\phi_k^N(x)) = \begin{cases} \phi_k^{N+1} & for \ K < \frac{N}{2} \\ \phi_{k+1}^{N+1} & for \ K < \frac{N}{2} \end{cases}$$

on the orthonormal basis $\{\phi_k^N\}_{0 \leqslant k < N}$ and extend it to all of $\ell^2(G_N)$ via normality. This implies that precisely the eigenspace spanned by $\phi_{\frac{N+1}{2}}^{N+1}$ (corresponding to the eigenvalue $\lambda_{\frac{N+1}{2}}^{N+1} = (N+1)^2/\pi^2$) does not lie in the image of $J$. We set $\widetilde{J}$ to be the adjoint $J^*$ of $J$. Choosing $\omega = 1$, we shall now first check the equations of Definition 5.1. Since $J$ is isometric, we have

$$\|Jf\| = \|f\| \leqslant 2\|f\|$$

as desired. Since $\widetilde{J} = J^*$, we have

$$\|\widetilde{J} - J^*\| = 0.$$

Since $\widetilde{J}J = Id_{\ell^2(G_N)}$, what remains to be checked is the demand

$$\|(Id - J\widetilde{J})\widetilde{R}_{-1}\|_{op} \leqslant K \cdot \frac{1}{N^2}.$$

We have

$$\|(Id - J\widetilde{J})\widetilde{R}_{-1}\|_{op} = 1 \cdot \frac{1}{1 + \lambda_{\frac{N+1}{2}}^{N+1}} = \frac{1}{1 + N^2/\pi^2} \leqslant \frac{\pi^2}{(N+1)^2} \leqslant \pi^2 \cdot \frac{1}{N^2}.$$

Thus let us now check that the conditions of Definition 5.2 are fulfilled. We note that with our identification operator and by symmetry ($\lambda_k^N = \lambda_{N-k}^N$), we have

$$\|JR_{-1} - \widetilde{R}_{-1}J\|_{op} = \max_{0 \leqslant k < \frac{N}{2}} \left| \frac{1}{1 + \frac{N^2}{\pi^2}\sin^2\left(\frac{\pi}{N}k\right)} - \frac{1}{1 + \frac{(N+1)^2}{\pi^2}\sin^2\left(\frac{\pi}{(N+1)}k\right)} \right|.$$

We now need to bound the right hand side uniformly in $k$ as $N \to \infty$. To this end we write $a := 1/N$ (which implies $\frac{N+1}{N} = 1 + a$) and $x = \frac{k}{N}$ (which for our allowed values of $k$ implies $0 \leqslant x < \frac{1}{2}$). With this we have

$$\left| \frac{1}{1 + \frac{N^2}{\pi^2} \sin^2\left(\frac{\pi}{N}k\right)} - \frac{1}{1 + \frac{(N+1)^2}{\pi^2} \sin^2\left(\frac{\pi}{(N+1)}k\right)} \right|$$

$$= (\pi a)^2 \left| \frac{1}{(\pi a)^2 + \sin^2(\pi x)} - \frac{1}{(\pi a)^2 + (1+a)^2 \sin^2\left(\pi x \frac{1}{1+a}\right)} \right|$$

$$= (\pi a)^2 \left| \frac{(1+a)^2 \sin^2\left(\pi x \frac{1}{1+a}\right) - \sin^2(\pi x)}{[(\pi a)^2 + \sin^2(\pi x)] \cdot [(\pi a)^2 + (1+a)^2 \sin^2\left(\pi x \frac{1}{1+a}\right)]} \right|$$

$$= (\pi a)^2 \left| \frac{\sin^2\left(\pi x \frac{1}{1+a}\right) - \sin^2(\pi x) + a \sin^2\left(\pi x \frac{1}{1+a}\right) + a^2 \sin^2\left(\pi x \frac{1}{1+a}\right)}{[(\pi a)^2 + \sin^2(\pi x)] \cdot [(\pi a)^2 + (1+a)^2 \sin^2\left(\pi x \frac{1}{1+a}\right)]} \right|$$

$$= (\pi a)^2 \left| \frac{\sin\left(\pi x \frac{a}{1+a}\right) \cdot \sin\left(\pi x \frac{a+2}{a+1}\right) + a \sin^2\left(\pi x \frac{1}{1+a}\right) + a^2 \sin^2\left(\pi x \frac{1}{1+a}\right)}{[(\pi a)^2 + \sin^2(\pi x)] \cdot [(\pi a)^2 + (1+a)^2 \sin^2\left(\pi x \frac{1}{1+a}\right)]} \right|$$

$$\leqslant (\pi a)^2 \left| \frac{\sin\left(\pi x \frac{a}{1+a}\right) \cdot \sin\left(\pi x \frac{a+2}{a+1}\right) + a \sin^2\left(\pi x \frac{1}{1+a}\right) + a^2 \sin^2\left(\pi x \frac{1}{1+a}\right)}{[\sin^2\left(\pi x \frac{a}{1+a}\right)] \cdot [(\pi a)^2]} \right|$$

$$\leqslant a \left| \frac{\frac{\sin\left(\pi x \frac{a}{1+a}\right)}{a} \cdot \sin\left(\pi x \frac{a+2}{a+1}\right) + \sin^2\left(\pi x \frac{1}{1+a}\right) + a \sin^2\left(\pi x \frac{1}{1+a}\right)}{\sin^2\left(\pi x \frac{1}{1+a}\right)} \right|$$

$$\leqslant 2a + a \left| \frac{\sin\left(\pi x \frac{a+2}{a+1}\right)}{\sin\left(\pi x \frac{1}{1+a}\right)} \right| \cdot \left| \frac{\sin\left(\pi x \frac{a}{1+a}\right)}{a \cdot \sin\left(\pi x \frac{1}{1+a}\right)} \right|.$$

Thus we are done if we can show that the function

$$F(a, x) = \left| \frac{\sin\left(\pi x \frac{a+2}{a+1}\right)}{\sin\left(\pi x \frac{1}{1+a}\right)} \right| \cdot \left| \frac{\sin\left(\pi x \frac{a}{1+a}\right)}{a \cdot \sin\left(\pi x \frac{1}{1+a}\right)} \right|$$

is bounded on the rectangle $[0, 1] \times [0, \frac{1}{2}]$. We change variables $y = \pi x/(1 + a)$ and consider

$$F(a, y) = \left| \frac{\sin(y(a + 2))}{\sin(y)} \right| \cdot \left| \frac{\sin(ya)}{a \cdot \sin(y)} \right|$$

on $[0, 1] \times [0, \frac{\pi}{2}]$ instead. Away from $y = 0$ this is obvious. Close to $y = 0$ we might Taylor expand in numerators and denominators respectively and then (formally) divide them both respectively by $y$ to see that the function $F(a, y)$ is indeed regular at $y = 0$ too and hence on the entire compact set $[0, 1] \times [0, \frac{\pi}{2}]$. As a continuous function, $F$ attains its supremum on this set. Denote it by $K$. Hence we now know

$$\|JR_{-1} - \widetilde{R}_{-1}J\|_{op} \leqslant [2 + K] \cdot a \equiv [2 + K] \cdot \frac{1}{N}.$$

Thus we have established the desired $\mathcal{O}(1/N)$-decay. $\qquad \square$

## M   PROOF OF THEOREM 6.1

**Theorem M.1.** For $p \geqslant 2$ we have in the setting of Theorem 3.1 that $\|\Psi_N^p(f) - \Psi_N^p(h)\|_{\mathbb{R}^{K_{out}}} \leqslant \left(\prod_{n=1}^N L_n R_n B_n\right) \cdot \|f - h\|_{\mathscr{L}_{in}}$. In the setting of Theorem 4.3 or 5.4 and under the additional assumption that the 'final' identification operator $J_N$ satisfies $\left| \|J_N f_i\|_{\ell^k(\widetilde{G}_N)} - \|f_i\|_{\ell^k(G_N)} \right| \leqslant$

$\delta \cdot K \cdot \|f_i\|_{\ell^2(G_N)}$ for all $f_i \in \ell^2(G_N)$, we have $\|\Psi_N^p(f) - \widetilde{\Psi}_N^p(\mathscr{J}_0 f)\|_{\mathbb{R}^{K_{out}}} \leqslant (N \cdot DRL + K \cdot (BRL)) \cdot (BRL)^{N-1} \cdot \|f\|_{\mathscr{L}_{\mathrm{in}}} \cdot \delta$.

*Proof.* To prove the first claim, we note

$$\|\Psi_N^p(f) - \Psi_N^p(g)\|_{\mathbb{R}^{K_{out}}} = \sqrt{\sum_{i \in K_{out}} \left| \|[\Phi_N(f)]_i\|_{\ell^p(G_{out})} - \|[\Phi_N(g)]_i\|_{\ell^p(G_{out})} \right|^2}$$

$$\leqslant \sqrt{\sum_{i \in K_{out}} \left| \|[\Phi_N(f)]_i - [\Phi_N(g)]_i\|_{\ell^p(G_{out})} \right|^2}$$

$$\leqslant \sqrt{\sum_{i \in K_{out}} \left| \|[\Phi_N(f)]_i - [\Phi_N(g)]_i\|_{\ell^2(G_{out})} \right|^2}$$

$$= \|\Phi_N^p(f) - \Phi_N^p(g)\|_{\mathbb{R}^{K_{out}}}$$

where we used the reverse triangle inequality and the fact that $\|\cdot\|_{\ell^p(\widetilde{G}_{out})} \leqslant \|\cdot\|_{\ell^2(\widetilde{G}_{out})}$ for $2 \leqslant p$. To finish the proof we now only need to apply Theorem 3.1.

To prove the second claim we note

$$\|\Psi_N^p(f) - \widetilde{\Psi}_N^p(\mathscr{J}_0 f)\|_{\mathbb{R}^{K_{out}}}$$

$$= \sqrt{\sum_{i \in K_{out}} \left| \|[\Phi_N(f)]_i\|_{\ell^p(G_{out})} - \|[\widetilde{\Phi}_N(\mathscr{J}_0 f)]_i\|_{\ell^p(\widetilde{G}_{out})} \right|^2}$$

$$= \sqrt{\sum_{i \in K_{out}} \left| \|[\Phi_N(f)]_i\|_{\ell^p(G_{out})} - \|[\mathscr{J}_N \Phi_N(f)]_i\|_{\ell^p(G_{out})} + \|[\mathscr{J}_N \Phi_N(f)]_i\|_{\ell^p(G_{out})} - \|[\widetilde{\Phi}_N(\mathscr{J}_0 f)]_i\|_{\ell^p(\widetilde{G}_{out})} \right|^2}$$

$$\leqslant \sqrt{\sum_{i \in K_{out}} \left| \|[\Phi_N(f)]_i\|_{\ell^p(G_{out})} - \|\mathscr{J}_N [\widetilde{\Phi}_N(f)]_i\|_{\ell^p(G_{out})} \right|^2}$$

$$+ \sqrt{\sum_{i \in K_{out}} \left| \|\mathscr{J}_N [\Phi_N(f)]_i\|_{\ell^p(G_{out})} - \|[\widetilde{\Phi}_N(\mathscr{J}_0 f)]_i\|_{\ell^p(\widetilde{G}_{out})} \right|^2}$$

$$\leqslant K \cdot \delta \cdot \|\mathscr{J}_N \Phi(f)\|_{\widetilde{\mathscr{L}}_{\mathrm{out}}} + \|\widetilde{\Phi}(\mathscr{J}_0 f) - \mathscr{J}_N \Phi(f)\|_{\widetilde{\mathscr{L}}_{\mathrm{out}}}$$

and the claim follows as before.

The proof of the third claim proceed in complete analogy. $\qquad\square$

# N ADDITIONAL DETAILS ON EXPERIMENTAL SETUP

**Scaling Operators:** The adjacency matrix fo the given graph is given by

$$A = \begin{pmatrix} 0 & 16 & 7 & 18 & 19 \\ 16 & 0 & 6 & 22 & 3 \\ 7 & 6 & 0 & 1 & 90 \\ 18 & 22 & 1 & 0 & 23 \\ 19 & 3 & 90 & 23 & 0 \end{pmatrix}. \tag{30}$$

**Collapsing Edges:** We consider the setting introduced in Section 5.2 and consider a generic fully connected graph $\widetilde{G}$ with $|\widetilde{G}| = 8$. We consider a splitting into $\widetilde{G} = \widetilde{G}_{Latin} \bigcup \widetilde{G}_{Greek} \bigcup \{\star\}$ with $|\widetilde{G}_{Latin}| = 3$ and $|\widetilde{G}_{Greek}| = 4$. As described in Section 5.2, we assume $\widetilde{W_{ab}}, \widetilde{W}_{a\star} = \mathcal{O}(1), \forall a, b \in \widetilde{G}_{Latin}$ and $\widetilde{W}_{\alpha\beta} = \frac{\omega_{\alpha\beta}}{\delta}$ and $\widetilde{W}_{\alpha\star} = \frac{\omega_{\alpha\star}}{\delta}$ such that $(\omega_{\alpha\beta}, \omega_{\alpha\star} = \mathcal{O}(1)$ for all $\alpha, \beta \in \widetilde{G}_{Greek}$. For completeness and reproducibility, the full adjacency matrix $\widetilde{W}$ can be found in Appendix N. We set

node weight on $\widetilde{G}$ to one and – as discussed – construct a graph $G$ with $|G| = 4$ through 'collapsing strong edges'.

The adjacency matrix of the larger 'un-collapsed' graph $\widetilde{G}$ we consider in Section 7 is given as follows

$$\widetilde{W} = \begin{pmatrix} 0 & 4 & 2 & 10 & 4 & 5 & 6 & 7 \\ 4 & 0 & 17 & 9 & 8 & 9 & 10 & 11 \\ 2 & 17 & 0 & 42 & 12 & 13 & 14 & 15 \\ 10 & 9 & 42 & 0 & 16/\delta & 7/\delta & 18/\delta & 19/\delta \\ 4 & 8 & 12 & 16/\delta & 0 & 6/\delta & 22/\delta & 3/\delta \\ 5 & 9 & 13 & 7/\delta & 6/\delta & 0 & 1/\delta & 90/\delta \\ 6 & 10 & 14 & 18/\delta & 22/\delta & 1/\delta & 0 & 23/\delta \\ 7 & 11 & 15 & 19/\delta & 3/\delta & 90/\delta & 23/\delta & 0 \end{pmatrix} \tag{31}$$

The exceptional vertex $\star$ here carries index "4" ("$\star = 4$"). Node weights are set to unity.

**The Realm of Large Graphs:** We also plot the difference in characteristic operators as opposed to their resolvents:

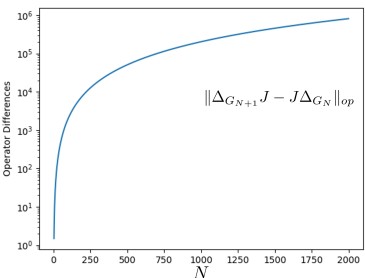

Figure 10: Operator Differences

Their distances does not decay.

**Experiments on Molecules:** The dataset we consider is the QM7 dataset, introduced in Blum & Reymond (2009); Rupp et al. (2012). This dataset contains descriptions of 7165 organic molecules, each with up to seven heavy atoms, with all non-hydrogen atoms being considered heavy. A molecule is represented by its Coulomb matrix $C^{\text{Clmb}}$, whose off-diagonal elements

$$C_{ij}^{\text{Clmb}} = \frac{Z_i Z_j}{|R_i - R_j|}$$

correspond to the Coulomb-repulsion between atoms $i$ and $j$, while diagonal elements encode a polynomial fit of atomic energies to nuclear charge Rupp et al. (2012):

$$C_{ii}^{\text{Clmb}} = \frac{1}{2} Z_i^{2.4}$$

For each atom in any given molecular graph, the individual Cartesian coordinates $R_i$ and the atomic charge $Z_i$ are also accessible individually. To each molecule an atomization energy - calculated via density functional theory - is associated. The objective is to predict this quantity, the performance metric is mean absolute error. Numerically, atomization energies are negative numbers in the range $-600$ to $-2200$. The associated unit is $[kcal/mol]$.

## O    NOTATIONAL CONVENTIONS

We provide a summary of employed notational conventions:

Table 1: Classification Accuracies on Social Network Datasets

| Symbol | Meaning |
|---|---|
| $G$ | a graph or a vertex set |
| $|G|$ | number of nodes in $G$ |
| $\mu_i$ | weight of node $i$ |
| $M$ | weight matrix |
| $\langle \cdot, \cdot \rangle$ | inner product |
| $W$ | adjacency matrix |
| $D$ | degree matrix |
| $\Delta$ | graph Laplacian |
| $\mathscr{L}$ | normalized graph Laplacian |
| $T$ | generic operator |
| $T*$ | adjoint of $T$ |
| $\sigma(T)$ | spectrum (i.e. collection of eigenvalues) of $T$ |
| $\lambda$ | an eigenvalue |
| $g(T)$ | function $g$ applied to operator $T$ |
| $\| \cdot \|_{op}$ | operator norm (i.e. spectral norm) |
| $\| \cdot \|_F$ | Frobenius norm |
| $\omega$ | a complex number |
| $\overline{\omega}$ | complex conjugate of $\omega$ |
| $z$ | a complex number |
| $B_\epsilon(\omega)$ | open ball of radius $\epsilon$ around $\omega$ |
| $a_k^g, b_k^g$ | complex number determined by $g$ and indexed by $k$ |
| $U$ | open set extending to infinity in $\mathbb{C}$ |
| $D$ | a Cauchy domain in $\mathbb{C}$ |
| $\partial D$ | the boundary of $D$ |
| $(\omega Id - T)^{-1}, R_\omega$ | the resolvent of $T$ at $\omega$ |
| $\gamma_T(\cdot)$ | resolvent profile of $T$ |
| $\oint ... dz$ | a complex line integral |
| $\oint ... d|z|$ | the corresponding real line integral |
| $\rho$ | a non-linearity |
| $P$ | a connecting operator |
| $\mathscr{L}$ | (possibly hidden) feature space associated to a GCN |
| $\Phi$ | map associated to a GCN |
| $\epsilon, \delta$ | small numbers |
| $J$ | an identification operator (possibly dependent on some $\epsilon$ or $\delta$) |
| $\widetilde{G}$ | Graph consisting of regular nodes, an exceptional node and a strongly connected sub-graph |
| $\widetilde{G}_{\text{Greek}}$ | nodes in a strongly connected sub-graph |
| $\star$ | exceptional node to which a strongly connected sub-graph is collapsed |
| $\widetilde{G}_{\text{Latin}}$ | regular nodes in $\widetilde{G}$ |
| $E_G(\cdot)$ | Energy form associated to the (undirected) graph $G$ |
| $h$ | distance between nodes on the circle |
| $\| \cdot \|_p$ | the $p$-norm on $\mathbb{R}^d$ |
| $p$ | a natural number |
| $\Psi$ | graph-level feature map associated to a GCN |
| $Z_i$ | atomic charge of atom corresponding to node $i$ |
| $x_i$ | Cartesian position of atom corresponding to node $i$ |
| $\frac{Z_i Z_j}{\|x_i - x_j\|}$ | Coulomb interaction between atoms $i$ and $j$ |
| $\|x_i - x_j\|$ | Euclidean distance between $x_i$ and $x_j$ |

