# OpenReview forum: "Limitless Stability for Graph Convolutional Networks "
_ICLR.cc/2023/Conference — ICLR 2023 poster_

### Official Review · Reviewer_4m2L · 2022-10-25

**Confidence:** 4
**Correctness:** 4
**Technical Novelty And Significance:** 4
**Empirical Novelty And Significance:** 3
**Recommendation:** 8

**Clarity, Quality, Novelty And Reproducibility:**

The paper presents new theoretical results of the stability of GCNs. Overall, the technical novelty is very solid.

**Strength And Weaknesses:**

Strengths:

1. The submission provides new theoretical stability aspects of graph convolutional neural networks. The stability analysis is quite general and hence can potentially be applied to other problem settings of graph learning problems.

2. The theoretical results are verified numerically.

3. In terms of theoretical novelty, this submission develops a general theory of stability for GCNs.

Weakness:
1. It would be nice to list notations so that these theoretical results are more accessible to a general audience.

**Summary Of The Paper:**

This submission provides new stability guarantees and transferability bounds for general graph convolutional networks. The analysis framework can be applied to both directed and undirected graphs based on recent advanced tools. Specifically, node-level and edge-level perturbations with some new properties are presented and discussed. These theoretical results are supported by numerical results.

**Summary Of The Review:**

This submission provides new stability guarantees and transferability bounds for general graph convolutional networks. The analysis framework can be applied to both directed and undirected graphs based on recent advanced tools. Specifically, node-level and edge-level perturbations with some new properties are presented and discussed. These theoretical results are supported by numerical results.

Overall, the authors present new theoretical results on the stability of GCNs models using some new tools. I didn't check the correctness of these proofs line by line. However, these theorems are quite solid works.

---

> ### Author Response · Authors · 2022-11-16
> **We have a included an appendix on Notational Conventions**
>
> We thank the reviewer for his or her careful evaluation, appreciation of the paper and kind comments. We were very happy to read that the  technical novelty of the paper was considered to be very solid.
>
> As we understand it, the sole perceived weakness of the paper and corresponding request was:
>
>  - It would be nice to list notations so that these theoretical results are more accessible to a general audience.
>
> We have now created a corresponding section ('Notational Conventions') .

---

### Official Review · Reviewer_QkKY · 2022-10-25

**Confidence:** 4
**Correctness:** 3
**Technical Novelty And Significance:** 3
**Empirical Novelty And Significance:** 3
**Recommendation:** 6

**Clarity, Quality, Novelty And Reproducibility:**

As mentioned above, the study of stability and transferability of GCNs is well-motivated. The paper provides novel insights into the transferability of GCNs for graphs with strong edges. However, I am not convinced that these results are sufficiently significant. I encourage authors to comment on the motivation behind studying this category of graphs.

I do not agree with the author's assessment that existing works that leverage the theory of graph limits are not applicable for non-asymptomatic settings as such insights can be drawn using triangle inequality. For instance, if a GCN defined on a graph with $n_1$ nodes approaches in some limit to a GCN defined on a limit object, the conditions on transferability between GCNs on $n_1$ and $n_2$ nodes in the non-asymptotic setting can be recovered via triangle inequality.  If the graphs with strong edges lie outside of the scope of theory of such limit objects, it will provide a much more convincing argument for the significance of the transferability results in this paper.


Clarity:

I have the following remarks, addressing which will help improve the clarity of the paper.

a. Please provide the definition of a normal operator.

b. Is $a_{\mu\nu}$ in Defintion 2.1 a scalar?

c. The difference term $JT - {\tilde T} J$ is not properly motivated. Please explain what this difference represents.



**Strength And Weaknesses:**

Strengths:

1. The presented theory is generalizable to generic utilized filters, graph shift operators as well as non-linearities.
2. The paper provides a fresh perspective to the transferability of GCNs for graphs with strongly connected edges.

Weaknesses:
1. While the study of stability and transferability of GCNs is a well motivated research problem, the paper does not provide significantly novel insights into the stability of GCNs. Transferability results are interesting, however, the paper lacks a convincing argument on widespread applicability of these results beyond a narrowly defined category of graphs.
2. Experiments consider fully connected, small graphs (size = 8) and therefore, provide limited insight.
3. Paper clarity can be improved (elaborated in the Clarity, Quality, Novelty And Reproducibility section).


**Summary Of The Paper:**

This paper provides a holistic theoretical framework to evaluate stability and transferability of graph convolutional networks (GCN). As expected, the various theoretical results provide bounds on the changes in GCN outputs in terms of perturbations to the input signal or the structure. A novel perspective on the transferability of GCNs is presented, where transferability is established between GCNs operating on graphs under the assumption that one graph is a collapsed version of the other. Limited numerical results that validate the proposed theory are also provided.

**Summary Of The Review:**

This paper provides some interesting and novel ideas on transferability. However, a major revision is needed to expand on the motivation, significance, and applicability of these results. More comprehensive experimental evaluations will also improve the paper considerably.


**After Author Rebuttal**
I thank the authors for their detailed response and a significant overhaul of the paper, which help address my concerns to some extent and help improve the paper significantly. The authors have also significantly extended the supplementary material with technical and empirical content, however, I did not go through this carefully.  The contributions of the paper in the domain of stability are now better elucidated. The authors provide additional experiments for larger graphs. Furthermore, the authors added an example to the section on transferability that relates their theory to the existing theory of transferability that relies on limit objects. After considering these aspects, I have raised my score to 6.

---

> ### Author Response · Authors · 2022-11-16
> **Followed provided advice dilligently + Explained in detail how transferability approach (contrary to previous approaches) also applies to small and medium sized graphs**
>
> We immensely thank the reviewer for the careful read of our paper. We were especially happy to read that the generality and wide applicability of our result - in terms of utilized characteristic operators, filters and non-linearities - as well as the novelty of our transferability results were appreciated.
>
> We are also very grateful, for the detailed comments, advice and questions we received; which we have followed diligently and answered carefully, as we detail point-for-point below:
>
> - Summary Of The Paper:
> This paper provides a holistic theoretical framework to evaluate stability and transferability of graph convolutional networks (GCN). As expected, the various theoretical results provide bounds on the changes in GCN outputs in terms of perturbations to the input signal or the structure.
>
> Uncovering and quantifying the effects various perturbations have on graph convolutional networks is of course indeed the main focus of GCN-stability research.
>
> One might thus indeed expect from a paper in this area that it provides bounds on the changes in GCN outputs under perturbations to the input signal.
> Unexpectedly, even after extensive literature-searches,  we only encountered one such result studying the stability to input signal perturbations (e.g. the sense of Lipschitz-ness). This result provided results for a very specific architecture evaluated on large random graphs [B].
> To the best of our knowledge, our work is thus the first to provide generally applicable and deterministic results for GCNs without being limited to any statistical setting or specific architecture.
>
> Additionally, our work is also the first to provide a throrough stability analysis for networks on directed graphs. Practically all previous works focused on establishing stability only for undirected graphs.
>
> -  A novel perspective on the transferability of GCNs is presented, where transferability is established between GCNs operating on graphs under the assumption that one graph is a collapsed version of the other.
>
>
>
>
>
> We completely agree that the problem of collapsing sub-graphs is the most important transferability-setting discussed in our paper; especially because it has not been considered before in the literature.
>
> To showcase that the newly introduced transferability framework developed in Section 5.2 however is not limited to this setting, we have added discussions of two additional exemplary transferability settings to our manuscript. We also numerically investigate all three setups in our revised experimental section.
>
> Instead of its applicability to the specific (albeit important) example of collapsing subgraphs, we would argue that the main distinguishing feature of our newly developed approach is its focus on _resolvents_ $1/(\omega Id - T)$ instead of _characteristic_ _operators_ $T$.
> This enables it to capture stability properties that might prove troublesome to be captured by previous approaches.
>
> We have undertaken several steps to make this a lot clearer in our revised manuscript:
>
> Right at the beginning of Section 5, we now introduce a new example, specifically tailored to
> showcase the importance of utilizing resolvents.
> This example is then extended to the setting of collapsing edges, which was already previously discussed in the original version of our submission.
>
> In a second new example, included in Section 5.2, we discuss applicability and performance of our approach to and in the realm of large graphs. To this end, we consider an increasing sequence of graphs approximating the circle in some sense. We then consider transferability between two graphs in this sequence. We find that the difference in resolvents between subsequent graphs decays as $1/N$ with N the graph cardinality. This decay behaviour then persists for filters and entire networks. In contrast, assuming a similar $1/N$-decay behaviour of the difference in _characteristic operators_, the approach of [4] would only yield a decay behaviour of $\sqrt{N}/N$ if no band limitation is imposed, as the stability constant in this approach scales as $\sqrt{\text{number of considered eigenvalues}}*\text{difference in operators}$. Utilizing resolvents, we circumvent this additional $\sqrt{N}$-factor in our approach. What is more, as we numerically showcase, a $1/N$-decay behaviour of the difference in _characteristic_ _operators_ is very much not given. In fact their difference diverges as $N \rightarrow \infty$. This draws into question the applicability of some existing approaches focusing on characteristic operators to this setting, while once again highlighting the merits of utilizing resolvents as we do in our approach.

---

> > ### Author Response · Authors · 2022-11-16
> > **Continuation of comment I**
> >
> > - Limited numerical results that validate the proposed theory are also provided.
> >
> > We completely agree that the scope of the numerical investigations of the previous version of our submission left room for extension.
> > We have now significantly extended our section on numerical results:
> >
> > We first discuss the setting of scaling operators, and numerically showcase how resolvents are able to capture a convergence behaviour, while traditional characteristic operators are unable to do so.
> >
> > We then return to the numerical example already included in the previous version of our submission. Here we showcase again the transferability between a graph and a version of this graph where a sub-graph of strong edges is collapsed. We plot the difference in resolvents  (determining the transferability of filters and networks in our approach) as a function of the characteristic coupling-strength in the collapsed subgraph. We also plot the transferability of monomials in the resolvents, which form the basis for our utilized filters.
> >
> > Then we investigate transferability for large graphs in the circle-discretization setting discussed above. Here we showcase how the transferability error of resolvents decays with the cardinality of considered graphs. We also include a plot of the difference of original characteristic operators, which does not vanish but instead tends to infinity.
> >
> > Finally, we now also test the transferability of an entire network in a real-world setting:
> > In order to do so, we first introduce a small additional theoretical result (Theorem 6.1) guaranteeing that stability and transferability persist even after aggregating node level features into graph level features. In our experiment we then utilise a two-layer GCN with 16 hidden nodes in each layer (mirroring the architecture of [5]) for which we aggregate node-level output to graph level features.
> > With this setup, we consider the transferability of graph-level feature vectors on the QM$7$ dataset. More precisely,  we consider the graph of methane ($5$ Nodes; one Carbon (node weight set to atomic charge $Z_1=6$);  $4$  Hydrogen nodes (node weights set to atomic charges $Z_{i>1}=1$)) and deflect one of the Hydrogen atoms ($i=2$) out of  equilibrium and on a straight line towards the Carbon atom. We thenconsider the transferability of the entire GCN between this graph and an effective description combining Carbon and deflected Hydrogen  into a single node $\star$ (node weight $\mu_{\star}= Z_1 + Z_2 = 7$). At equilibrium  the transferability error of the network is of $\mathcal{O}(1)$. It decreases fast with decreasing Carbon-Hydrogen distance; with  the choice of representation (effective vs. original) quickly becoming insignificant for generated feature vectors.
> >
> >
> >
> >
> >
> >
> > - The presented theory is generalizable to generic utilized filters, graph shift operators as well as non-linearities.
> >
> > We are very happy to read that we were able to convince the reviewer of the wide applicability of our theory.
> >
> >
> > - The paper provides a fresh perspective to the transferability of GCNs for graphs with strongly connected edges.
> >
> > We are very happy to read that the novelty of our approach was appreciated by the reviewer.
> > Above, we have already described in detail how our approach is not limited in applicability to the setting of graphs with strongly connected subgraphs. We also detailed which steps we have undertaken to make this point a lot clearer in our revised manuscript.
> > The setting of strongly connected subgraphs does however remain an important example of a transferability setting. What is more, we are unaware of _any_ previous works studying transferability within this setting. Thus -- to the best of our knowledge --  our work is the first to give explicit stability bounds and now in the revised manuscript  also a statement detailing precisely under which choices of characteristic operators the associated GCNs are stable to such coarse graining procedures (c.f. Theorem 5.7).
> > Should we have overlooked already established results, we would of course be happy to include a corresponding reference, in order to properly cite them and include them in our bibliography.

---

> > > ### Author Response · Authors · 2022-11-16
> > > **Continuation of comment II**
> > >
> > > - Weaknesses:
> > > While the study of stability and transferability of GCNs is a well motivated research problem, the paper does not provide significantly novel insights into the stability of GCNs.
> > >
> > > We agree with the reviewer that both transferability and stability are important aspects of GCNs and believe that results that are as general, widely applicable and precise as possible are of utmost importance in order to understand GCNs and in order to aid in the design of new architectures. It is for this reason that we are confident in arguing that our paper _does_ provide significant new insights into the stability of GCNs, as we explain in detail below. In order to also make this point a lot clearer in our revised submission, we now highlight our novel contributions on this subject already in abstract and introduction:
> > >
> > > First and foremost -- as we have already described above – our manuscript is the first to seriously investigate the Lipschitz-constant of GCNs and bound it in terms of utilized non-linearities and employed filters. What is more, not only do we give an estimate of the stability constant in terms of the magnitudes of the individual filters measured in some norm, but in terms of an interplay of the filters with each other (within the respective layers).
> > > This has immediate implications for how practitioners might want to train models, which we now explicitly discuss directly after the relevant theoretical result (Theorem 3.7) in our revised manuscript.
> > >
> > > Secondly, virtually all previous works on stability of GCNs were only applicable to networks on undirected graphs with their self-adjoint (and hence normal) characteristic operators. In practice however, there is of course a vast number of problems and datasets where the directedness of graphs plays an important role (e.g. citation-networks, communication networks, etc.).
> > > We suspect that this previous limitation in scope of GCN-stability analysis did not come about due to a lack of interest in directed graphs, but rather due to the fact that tools able to cope with the considerably more tricky spectral theory and general behaviour of non-normal operators (such as the characteristic operators on directed graphs) are much harder to come by.
> > > Our work is the first to provide an advanced and thorough stability analysis for filters and networks operating on such directed graphs.
> > > We even succeed in providing single-filter- as well as network level stability results that are independent of (the size of) the utilized characteristic operator.
> > > We achieve these goals by making heavy use of somewhat advanced tools from complex analysis and operator theory such as contour integration and resolvent analysis.
> > > To highlight the importance of these tools more we have now renamed our ‘Preliminaries and Framework’ section to ‘GCNs via Complex Analysis and Operator Theory’.  We hope that this will draw more attention to the so far underappreciated utility of these tools and hope that they might get picked up by the community for further investigations.
> > >
> > > Beyond merely applying tools from complex analysis to investigate stability, we also use them to provide the implementation of filters that were so far only applicable to normal-operators/on undirected graphs to the characteristic operators on directed graphs (c.f. eq. 1 and eq. 2).
> > >
> > > But also when restricting to the setting of directed graphs and their self-adjoint (or more generally normal) characteristic operators, we provide considerably novel results:
> > > Previous stability results on undirected graphs were limited to somewhat narrow classes of filters:
> > > In [1] for example, filters are mandated to satisfy $|h(\cdot)| \leq 1$ and have decaying local Lipschitz constant, in [2] filters have to be functions of the variable $\frac{x+i}{x-i}$ and in [3] filters are assumed to be constant on certains parts of the spectrum/the real line.
> > > We do not make such demands and instead provide results for general and almost arbitrary filters (c.f. Lemma 2.3, Lemma 4.1, Lemma 4.2 and Theorems 3.1 and 4.3) in terms of newly-introduced semi-norms.
> > >
> > >
> > >
> > > Additionally, we would also like to point out that the important fact that scalar-Lipschitz-ness translates directly to operator Lipschitzness without an increase in Lipschitz constant precisely if the Frobenius norm is utilized (c.f. Lemma 4.1) had so far not been appreciated by the community.

---

> > > > ### Author Response · Authors · 2022-11-16
> > > > **Continuation of comment III**
> > > >
> > > > - Transferability results are interesting, however, the paper lacks a convincing argument on widespread applicability of these results beyond a narrowly defined category of graphs.
> > > >
> > > > Above we have discussed how our transferability approach is not limited in applicability to the setting of graphs with strongly connected subgraphs, provided examples of different transferability settings that our framework captures well while other approaches fail or struggle to do so and how the main advantage of our approach is its focus on resolvents $1/(\omega Id - T)$ as opposed to characteristic operators $T$. We also detailed which steps we have undertaken in our revised submission to make this point a lot clearer.
> > > >
> > > > Here we would like to additionally point out that our approach is currently the only one able to deal with general undirected graphs/non-normal operators (Graphon approaches restrict to self-adjoint operators (see e.g. [3]) while approaches discretising topological spaces [4] restrict to diagonalizable characteristic operators).
> > > > We had already stated this point multiple times in our original submission, but made it even more clear in our revised manusript.
> > > >
> > > >
> > > >
> > > >
> > > >
> > > > - I do not agree with the author's assessment that existing works that leverage the theory of graph limits are not applicable for non-asymptomatic settings as such insights can be drawn using triangle inequality.
> > > >
> > > > Clearly the theory of graph limits has an important place in the discussion of transferability properties of GCNs, as many important publications have already established.
> > > > However, in order to leverage the full might of this theory, there clearly needs to exist an underlying limit object in the first place, to which the considered graph sequence then converges in some sense.
> > > > However, such ‘limit objects’ do not always exist, especially when considering directed or small graphs. One of the merits of our approach is, that it does not need such an underlying limit object to be specified or even exist.
> > > >
> > > >
> > > >
> > > > - For instance, if a GCN defined on a graph with $n_1$ nodes approaches in some limit to a GCN defined on a limit object, the conditions on transferability between GCNs on $n_1$ and $n_2$ nodes in the non-asymptotic setting can be recovered via triangle inequality.
> > > >
> > > > We completely agree that IF such a limit object exists for two graphs between which architectures are meant to be transferred, one may in principle apply the triangle inequality to reduce the problem to twice bounding the transferability between a graph and a limit object. However, the accrued error when transferring between graph and limit object – apart from being dependent on network specifications -- is contingent on how well the graph approximates the limit object. This approximation error typically has  power-law decay in the number of nodes of the graph (see e.g. Theorem 33 in [4] or results in [3]).
> > > > Hence we typically need the number of nodes of graphs (between which architecture should be transferred) to be large in order to have a low transferability error.
> > > > Hence in order to achieve a low transferability error, we need to consider graphs with many nodes.
> > > > Thus results leveraging such graph limits -- while being very much applicable in the large-graph regime -- are inapplicable to small (or medium sized, depending on the details of the network architecture) graphs.
> > > >
> > > > This realm of large graphs is the asymptotic setting we refer to.
> > > > We have now stated  this point explicitly already in the introduction to our paper.
> > > >
> > > > - If the graphs with strong edges lie outside of the scope of theory of such limit objects, it will provide a much more convincing argument for the significance of the transferability results in this paper.
> > > >
> > > > Yes, that is precisely correct, with the caveat that it is not so much the strength of edges but rather the number of nodes that is the limiting factor!
> > > > In [3] for example (assuming Lipschitz constants of $\mathcal{O}(1)$), the estimate of the transferability error between  GCNs on $n_1$ and $n_2$ nodes is larger than
> > > > $ LF^{L-1}(\frac{1}{\sqrt{n_1}} + \frac{1}{\sqrt{n_2}})$ (c.f. Theorem 2 of [3]). Here $F$ is the maximum width and $L$ is the depth of the network. On a typical network of depth $2$ with $16$ hidden units in each layer (c.f. [5]), the transferability error on graphs of no more than $8$ nodes as we consider in our edge-collapse experiments thus is estimated by something larger than $\frac{32}{\sqrt{2}}  = \mathcal{O}(10)$. If we consider a three layer network of similar width or the architecture of [6] (where $F=64$), transferability between graphs of $\mathcal{O}(100)$ already becomes difficult.

---

> > > > > ### Author Response · Authors · 2022-11-16
> > > > > **Continuation of comment IV**
> > > > >
> > > > > - Experiments consider fully connected, small graphs (size = 8) and therefore, provide limited insight.
> > > > >
> > > > > We believe it important to also numerically showcase that while other transferability approaches fail on small graphs, ours does indeed succeed. To this end we now provide two experiments on small graphs, one on synthetic data and one on real world data.
> > > > >
> > > > > Following this comment however, we now also have also included an experiment showcasing transferability between large graphs of up to $N = 2000$ nodes within our approach.
> > > > >
> > > > >
> > > > > - Paper clarity can be improved (elaborated in the Clarity, Quality, Novelty And Reproducibility section).
> > > > >
> > > > > We have diligently followed the advice on improving clarity, as detailed below:
> > > > > - Clarity, Quality, Novelty And Reproducibility:  As mentioned above, the study of stability and transferability of GCNs is well-motivated. The paper provides novel insights into the transferability of GCNs for graphs with strong edges.
> > > > > However, I am not convinced that these results are sufficiently significant. I encourage authors to comment on the motivation behind studying this category of graphs.
> > > > >
> > > > >
> > > > > We are happy to do so:
> > > > > The first and most important thing that we would like to point is that our theory and our experiments don’t focus on graphs with strong edges per se, but rather on graphs containing a subcluster whose edges are stronger than the edges surrounding the cluster. We then show that deploying a GCN on this graph, or a graph where the subgraph is collapsed to a single node is increasingly the same, as the characteristic coupling strength within the sub-graph becomes larger.
> > > > >
> > > > > Instead of collapsing the subgraph to a node, one might also think of this procedure as a coarse graining operation, grouping together nodes that are strongly connected into a single entity.
> > > > >
> > > > > As a first example, our developed theory, then applies to networks deployed on original graphs and graphs that arose in a graph-pooling procedure from these original graphs.
> > > > >
> > > > >
> > > > > But more generally, this conceptual setting is applicable whenever there is a hierarchical or multi-scale structure present in the graph, or the system that the graph describes.
> > > > >
> > > > > In a spatial network  -- where nodes are embedded as vectors into some Euclidean space -- for example, one might set edge-weights to the inverse distance between nodes. This would encode the reasonable expectation that the immediate spatial environment of a node in the embedding space is more important than nodes that are far-removed.
> > > > > However, if real world data is used, the resolution of the data-gathering apparatus might be limited: at times it might group together two entities (encoded by nodes in our graph model) into a single larger entity as the distance between them might be too small to be reliably resolved during data collection. At other times it might succeed in resolving the distance.
> > > > > Our results on collapsing edges now show that if and only if the graph Laplacian (as opposed to the adjacency matrix or the normalized graph Laplacian) is used as the characteristic operator in our network, a GCN deployed on this network is insensitive to such data collection errors.
> > > > >
> > > > > One might also consider the hierarchical nature of social networks: On a fundamental level,  one might for example choose edge weights as number of exchanged messages between users (with users encoded as nodes).
> > > > > Instead at the level of individual users, one might also be interested in investigating the network at the level of communities or be interested in the interaction between individual users and communities.
> > > > > Our results show that precisely if the graph Laplacian is utilized,  grouping together users that have exchanged a lot of messages with each other into a single community is an operation under which our GCN is stable.
> > > > >
> > > > > As a final example, our results also have implications for the burgeoning field of GNNs
> > > > > applied to molecular science:
> > > > > For a molecular graph, one typically chooses nodes to represent individual atoms. We might choose edge-weights to be given by the Coulomb interaction between the respective atoms, as is for example done in the QM$7$ dataset. The coulomb interactions between two atoms is given by $C_{ij} = \frac{Z_i Z_j}{|x_i  - x_j|}$ with $Z_i$ the atomic charge and $x_i$ the position of atom $i$. This interaction describes the electrostatic Energy of one of the atoms in the electrostatic field of the other. The closer the charges are to each other, the larger the Coulomb interaction.
> > > > > Now, instead of solely being interested in interactions between individual atoms, one might also be interested in the interactions between (functional) groups of atoms. Our results show, that if one groups together atoms that are close to each other (or equivalently that interact strongly with each other) to a single group, a GCN is stable under this change of scale, if the graph Laplacian is employed in each Layer.
> > > > >
> > > > > Thanks to this comment, we now mention these examples explicitly in our paper.

---

> > > > > > ### Author Response · Authors · 2022-11-16
> > > > > > **Continuation of comment V**
> > > > > >
> > > > > > - Clarity:
> > > > > > I have the following remarks, addressing which will help improve the clarity of the paper.
> > > > > > a.	Please provide the definition of a normal operator.
> > > > > >
> > > > > > Normal operators can be thought of as a generalization of self-adjoint operators:
> > > > > > While a self-adjoint operator $A$ satisfies $A = A^*$ (with $ A^*$ the adjoint), a normal operator need only commute with its adjoint ($AA^* = A^*A$) and not be equal to it. As a consequence of these definitions, self-adjoint operators posses an associated basis of eigenvectors and real-valued eigenvalues. Normal operators posses an assoicated basis of eigenvectors and corresponding complex-valued eigenvalues.
> > > > > > While we had already provided the definition of a normal operator in the appendix in our original submission, we have now included it into the main body of Section 2, following this advice by the reviewer.
> > > > > >
> > > > > > - b.	Is $a_{\mu\nu}$ in Defintion 2.1 a scalar?
> > > > > >
> > > > > > That is precisely correct. We expand our function $g$ into basis-functions indexed by the pair $(\mu\nu)$ of indices. The coefficient $a_{\mu\nu}$ (for a fixed index-pair $(\mu\nu)$) is then a scalar. To be clearer about this point, we now state this explicitly in our newly introduced section on notational conventions.
> > > > > >
> > > > > > - c. The difference term $JT - \widetilde{T}J$ is not properly motivated. Please explain what this difference represents.
> > > > > >
> > > > > > We are happy to oblige:
> > > > > > If $J$ is simply the identity mapping, then $JT - \widetilde{T}J$ reduces to the ordinary difference $T - \widetilde{T}$ which simply measures how different the operators $T $ and $\widetilde{T}$ are from each other.
> > > > > > Now since in general, we allow $T $ and $\widetilde{T}$ to live in different spaces, we can not simply subtract them from each other. The operator $J$ translates between the different spaces. The object $JT  $ first applies the operator $T$ in the original spaces and then transports the result it over to the second (tilde-)space using $J$. The term $ \widetilde{T}J$ does the opposite: it first transports a vector over to the second (tilde-)space and then applies the operator $\widetilde{T}$. The term $JT - \widetilde{T}J$ then measures the difference between first applying the first characteristic operator and then transporting and first transporting and then applying the second characteristic operator.
> > > > > > While we had already included a discussion of this philosophy in our original submission, we have now made it clearer in our revised manuscript.
> > > > > >
> > > > > >
> > > > > > - More comprehensive experimental evaluations will also improve the paper considerably.
> > > > > >
> > > > > > In addition to our original experiments, we included many new experiments  in our revised submission -- as already discussed above. Notably one new experiment is concerned with transferability on large graphs (with $\mathcal{O}(1000)$ nodes) and one experiment is concerned with the transferability of a full network on real world data.
> > > > > >
> > > > > > References:
> > > > > >
> > > > > > [A] https://arxiv.org/abs/2006.01868
> > > > > >
> > > > > > [1]  https://arxiv.org/pdf/1905.04497
> > > > > >
> > > > > > [2] https://arxiv.org/abs/1901.10524
> > > > > >
> > > > > > [3] https://arxiv.org/pdf/2006.03548
> > > > > >
> > > > > > [4] https://arxiv.org/abs/1907.12972
> > > > > >
> > > > > > [5] https://arxiv.org/pdf/1609.02907
> > > > > >
> > > > > > [6] https://arxiv.org/abs/1705.07664

---

### Official Review · Reviewer_T5hc · 2022-10-25

**Confidence:** 2
**Clarity, Quality, Novelty And Reproducibility:** The theory of this paper is rigorous …
**Correctness:** 4
**Technical Novelty And Significance:** 3
**Empirical Novelty And Significance:** 3
**Recommendation:** 6

**Strength And Weaknesses:**

Strength: The paper has a solid theoretical analysis and numerical investigations, numeric results are used to validate the claim.
Weaknesses: Although the theoretical evidence is strong, practical experiments on real applications are missed. How to validate the practicability of the proposed theory？

**Summary Of The Paper:**

This work establishes rigorous, novel and widely applicable stability guarantees and transferability bounds for general graph convolutional networks – without reference to any underlying limit object or statistical distribution. Theoretical results are supported by corresponding numerical investigations.

**Summary Of The Review:**

Except for the lack of practical experimental verification, this is a good theory paper.

---

> ### Author Response · Authors · 2022-11-16
> **We significantly extended our experimental section**
>
> We sincerely thank the reviewer for her or his careful consideration of the paper.
> We were especially happy to read  the sentiment that
> - 'Except for the lack of practical experimental verification, this [submission] is a good theory paper.'
>
>
> In our revised version of the submission, we have now substantially revised and extended our experimental section:
> We now conduct experiments on resolvent- and filter transferability between small graphs, transferability between large graphs and real-world-transferability between graph-level-features generated by a network on QM$7$.
>
>
>
>
> Let us also adress the raised question:
>
> - How to validate the practicability of the proposed theory？
>
>
> In order to test the transferability of an entire network in a real-world setting, we first introduce a small additional theoretical result (Theorem 6.1) guaranteeing that stability and transferability persist even after aggregating node level features into graph level features. In our experiment we then utilise a two-layer GCN with 16 hidden nodes in each layer (mirroring the architecture of [1]) for which we aggregate node-level output to graph level features.
> With this setup, we consider the transferability of graph-level feature vectors on the QM$7$ dataset. More precisely,  we consider the graph of methane ($5$ Nodes; one Carbon (node weight set to atomic charge $Z_1=6$);  $4$  Hydrogen nodes (node weights set to atomic charges $Z_{i>1}=1$)) and deflect one of the Hydrogen atoms ($i=2$) out of  equilibrium and on a straight line towards the Carbon atom. We then consider the transferability of the entire GCN between this graph and an effective description combining Carbon and deflected Hydrogen  into a single node $\star$ (node weight $\mu_{\star}= Z_1 + Z_2 = 7$). At equilibrium  the transferability error of the network is of $\mathcal{O}(1)$. It decreases fast with decreasing Carbon-Hydrogen distance; with  the choice of representation (effective vs. original) quickly becoming insignificant for generated feature vectors.
>
>
> Reference:
>
> [1] https://arxiv.org/abs/1609.02907

---

### Official Review · Reviewer_ekgB · 2022-10-27

**Confidence:** 3
**Correctness:** 3
**Technical Novelty And Significance:** 4
**Empirical Novelty And Significance:** 3
**Recommendation:** 6

**Clarity, Quality, Novelty And Reproducibility:**

___Clarity___
- The paper could reach the TLDR of their goal much faster. The first paragraph can be cut down substantially and then the particular contribution of this paper should be emphasized at most in the second paragraph. RIght now the Introduction feels too much like a (good) Related work section, bugging the reader with detailed literature that only people searching for related work would care about. Instead, I would use the introduction to make it very clear what the work is and why we care, closer to the very last paragraph. Furthermore, all these references could have profited from coming after "preliminaries and framework".
- In section 2 we need intuition on why we care about each new concept, and how things are going to be used.
- Minor details
    - First paragraph page 2 "In ?? stability"
    - Typo "whic" in first paragraph section 3
- Labels in figure 3 are too small. Consider putting less numbers but larger.

__Quality___
    - Although I didn't check the proofs in detail, the analysis seems thorough and the goals non-trivial.
    - The numerical analysis was a bit shallow and the connection to the theoretical bounds could have been explained better.

___Novelty___ although I'm not an expert in theoretical analysis of GNNs this felt novel to me, particularly section 5.

___Reproducibility__ this is mostly math and proofs where in the appendix so it seems pretty reproducible.

**Strength And Weaknesses:**

___Strengths____
    - I found the stability to structural perturbations very interesting and relevant for the community.
    - The analysis is very self-contained and complete at multiple levels, from introducing all the necessary concepts, to analyzing a wide variety of perturbations one may face in practice.
    - Although I didn't check the proofs in detail, the arguments and statements of the bounds are solid.
___Weakness____
    - The paper could do better to first motivate the "Why" (why do we care about what we are going to be presented).
    - Similarly, it is lacking a "So What" on the bounds provided, which are often just left there as final statements, without an analysis that explains whether 1) they are (likely to be) tight and 2) what this implies for practitioners.
    - Although well-written, the paper felt quite dense, even compared to other pure-math ML papers. More examples such as Figure 2 would help.
    - As far as I understood, the assumption on the non-linearities discards the sigmoid and the softmax, which are popular non-linearities. It would be good to acknowledge this directly by name.

**Summary Of The Paper:**

This paper provides theoretical bounds on the stability properties of graph convolutional networks. It does so at multiple levels: input signal perturbations, edge perturbations, and structural perturbations. Input stability refers to the output of the network not changing much if we change the input signal. Similarly, we can consider the change in output if we change the edge weights (edge perturbation) as well as if we collapse nodes connected by strong edges (structural perturbation). Numerical analysis is provided to support some of the claims.

**Summary Of The Review:**

I believe the paper could have been clearer and motivated the problem better. Similarly, the empirical analysis could have been more thorough. However, I believe the problem is important and the theoretical contributions is (AFAIK) novel and note-worthy. Therefore, I recommend to accept the paper.

---

> ### Author Response · Authors · 2022-11-16
> **Shortened Introduction, added a lot more experiments, discussed theoretical results in more depth**
>
> We would like to thank the reviewer for the careful review and the poignant comments on our paper! We were especially happy to read that our work on stability to structural perturbation was found to be interesting and relevant for the community.
>
> Let us address the raised points one by one below:
>
> - Summary Of The Paper: This paper provides theoretical bounds on the stability properties of graph convolutional networks. It does so at multiple levels: input signal perturbations, edge perturbations, and structural perturbations. Input stability refers to the output of the network not changing much if we change the input signal. Similarly, we can consider the change in output if we change the edge weights (edge perturbation) as well as if we collapse nodes connected by strong edges (structural perturbation).
>
> We think that this is a very fair summary of our manuscript as it was originally submitted. The problem of collapsing sub-graphs remains the most important example of structural perturbations discussed in our paper and the one that we put the most emphasis on.   To showcase that the newly developed transferability framework developed in Section 5.2 however is not limited to this setting, we have added discussions of two additional exemplary transferability settings to our manuscript. We also numerically investigate all three setups in our revised experimental section.
>
>
> - Numerical analysis is provided to support some of the claims.
>
> As further detailed below, we have now significantly increased the scope of our experimental section.
>
>
> - Strength And Weaknesses:
> Strengths_ - I found the stability to structural perturbations very interesting and relevant for the community.
>
> It might delight the reviewer that following his or her advice on shortening the introduction (see below), we had space to include an additional result (c.f. Theorem 5.7) which establishes that networks are stable to the collapse of edges if AND ONLY IF the characteristic operator is chosen as the graph Laplacian (as opposed to the adjacency matrix or the normalized graph Laplacian). We have also included a new example showcasing how our approach captures stability on large graphs; and potentially better so than already existing approaches (c.f. Theorem 5.8, Fig. 7 and Fig. 10).
>
> - The analysis is very self-contained and complete at multiple levels, from introducing all the necessary concepts, to analyzing a wide variety of perturbations one may face in practice. - Although I didn't check the proofs in detail, the arguments and statements of the bounds are solid.
>
> We were very happy to read this sentiment.
>
> - Weakness_ - The paper could do better to first motivate the "Why" (why do we care about what we are going to be presented).
>
> Following this advice, we have substantially changed our 'Introduction' section, to provide a better motivation for our work: We describe how previous results on stability only apply to limited classes of filters and are most often inapplicable to GCNs on directed graphs altogether, so that a general theory allowing for arbitrary classes of filters of Graph Shift Operators (GSOs) is necessary. We also discuss the limitations of previous approaches to formalizing stability to structural perturbations (e.g. their  inapplicability to small-to-medium sized social networks and incapapbility of capturing the inherent multi-scale nature of molecular graphs) and how these obstacles are overcome by our approach.
>
>
>  - Similarly, it is lacking a "So What" on the bounds provided, which are often just left there as final statements, without an analysis that explains whether 1) they are (likely to be) tight
>
> Following this comment, we have extended existing discussions of results and added new ones. In particular, we now discuss in detail the implications and prerequisites of and for  Lemma 2.3, Theorem 3.1, Lemma 4.1 and Theorem 4.3. We also discuss under which circumstances (namely sparse connections between layers in the GCN) bounds are not necessarily tight (compare also the new Fig. 9 and surrounding discussion). For theorems in Section 5, we give two example applications.
> We also test Theorem 5.6, Theorem 5.8 as well as the new Theorem 6.1 numerically.

---

> > ### Author Response · Authors · 2022-11-16
> > **Continuation of comment I**
> >
> > -   [...] ... [without an analysis that explains ]  [...] what this implies for practitioners.
> >
> > Following this comment and implied advice, we have now included  discussions of implications for network-training and  loss-function design after Theorem 3.1 for stability to input signal perturbations and after Theorem 4.3 for  stability to edge level perturbations.
> >
> > In the realm of structural perturbations, our newly included Theorem 5.7 now also advises practitioners that in order to obtain stability to collapsing subgraphs, the graph Laplacian (as opposed to the adjacency matrix ot the normalized graph Laplacian) has to be chosen.
> >
> >
> >
> > - Although well-written, the paper felt quite dense, even compared to other pure-math ML papers. More examples such as Figure 2 would help.
> >
> > We have followed this advice diligently and included seven new figures in total. These visualize concepts from the domains needed for the implementation of holomorphic filters, over graphs and feature-aggregation-mechanisms used in experiments, to experimental results.
> >
> >
> >  - As far as I understood, the assumption on the non-linearities discards the sigmoid and the softmax, which are popular non-linearities. It would be good to acknowledge this directly by name.
> >
> > This is a good point. One indeeds needs the fact that the GCN preserves triviality (i.e. maps zero to zero) in order to use the Lipschitz-type result of Theorem 3.1 to draw the conclusion that also $||\Phi(f)||\lesssim  ||f||$. We have now included a corresponding comment about utilizable non-linearities in our revised manuscript.
> >
> >
> > - The paper could reach the TLDR of their goal much faster. The first paragraph can be cut down substantially and then the particular contribution of this paper should be emphasized at most in the second paragraph.
> >
> > We have strictly followed this advice and significantly shortened our introduction, while simultaneously highlighting the importance and contribution of our submission already in the second paragraph.
> > - RIght now the Introduction feels too much like a (good) Related work section, bugging the reader with detailed literature that only people searching for related work would care about. Instead, I would use the introduction to make it very clear what the work is and why we care, closer to the very last paragraph. Furthermore, all these references could have profited from coming after "preliminaries and framework".
> >
> > Following this advice, we have significantly reduced the scope of our related-work paragraph, which we now use to summarize what is known in the literature, where important gaps in knowledge lie and how we fill them. Since this now motivates the scope of our work, we have kept it in the ‘Introduction’ section. Should the reviewer still feel that there would be a different, better place for this paragraph, we would of course be happy to oblige.
> >
> >
> > - In section 2 we need intuition on why we care about each new concept, and how things are going to be used.
> >
> > We have now added such comments, explaining for example how the introduced semi-norms essentially provide the stability constants in the single filter setting.
> >
> > - Minor details
> > o	First paragraph page 2 "In ?? stability"
> > o	Typo "whic" in first paragraph section 3
> >
> > We thank the reviewer for spotting these typos. We have fixed them.
> >
> > - Labels in figure 3 are too small. Consider putting less numbers but larger.
> >
> > We now use a log-scale on all y-axes of our experimental plots, with number of inscriptions thus automatically reduced. Additionally, we enlarged corresponding graphics in our experimental section when compared to our first submission. We believe that now axis-inscriptions should be legible. Should that not be the case (we believe the limiting Figure here would be Fig. 6) we would of course be happy to reduce the number of utilized axis inscriptions.
> >
> > - Quality_ - Although I didn't check the proofs in detail, the analysis seems thorough and the goals non-trivial.
> >
> > We thank the reviewer for this sentiment.

---

> > > ### Author Response · Authors · 2022-11-16
> > > **Continuation of comment II**
> > >
> > > - The numerical analysis was a bit shallow and the connection to the theoretical bounds could have been explained better.
> > >
> > > We have now substantially revised our experimental section:
> > > We now first discuss the setting of scaling characteristic operators (c.f. the beginning of Section 5), and numerically showcase how resolvents are able to capture a convergence behaviour, while traditional characteristic operators are unable to do so.
> > >
> > > We then return to the numerical example already included in the previous version of our submission. Here we showcase again the transferability between a graph and a version of this graph where a sub-graph of strong edges is collapsed. We plot the difference in resolvents  (determining the transferability of filters and networks in our approach) as a function of the characteristic coupling-strength in the collapsed subgraph. We also plot the transferability of monomials in the resolvents, which form the basis for our utilized filters.
> > >
> > > Then we investigate transferability for large graphs in the circle-discretization setting now included as an additional example in Section 5. Here we showcase how the transferability error of resolvents decays with the cardinality of considered graphs. We also include a plot of the difference of original characteristic operators, which does not vanish but instead tends to infinity.
> > >
> > > Finally, we now also test the transferability of an entire network in a real-world setting:
> > > In order to do so, we first introduce a small additional theoretical result (Theorem 6.1) guaranteeing that stability and transferability persist even after aggregating node level features into graph level features. In our experiment we then utilise a two-layer GCN with 16 hidden nodes in each layer (mirroring the architecture of [1]) for which we aggregate node-level output to graph level features.
> > > With this setup, we consider the transferability of graph-level feature vectors on the QM$7$ dataset. More precisely,  we consider the graph of methane ($5$ Nodes; one Carbon (node weight set to atomic charge $Z_1=6$);  $4$  Hydrogen nodes (node weights set to atomic charges $Z_{i>1}=1$)) and deflect one of the Hydrogen atoms ($i=2$) out of  equilibrium and on a straight line towards the Carbon atom. We then consider the transferability of the entire GCN between this graph and an effective description combining Carbon and deflected Hydrogen  into a single node $\star$ (node weight $\mu_{\star}= Z_1 + Z_2 = 7$). At equilibrium  the transferability error of the network is of $\mathcal{O}(1)$. It decreases fast with decreasing Carbon-Hydrogen distance; with  the choice of representation (effective vs. original) quickly becoming insignificant for generated feature vectors.
> > >
> > >
> > > - Novelty although I'm not an expert in theoretical analysis of GNNs this felt novel to me, particularly section 5.
> > >
> > > We were very happy to hear this and do -- of course -- agree:
> > >
> > >  We are for example unaware of previous works investigating the stability to the important coarse-graining procedure of  replacing strongly connected subgraphs  by single nodes or utilizing resolvents to circumvent the problem of uncontrollably growing operator norms of characteristic operators on rescaled graphs or increasing graph sequences.
> > >
> > > The reviewer might also be happy to read that we   have now expanded section 5 even further in scope and clarity, when compared to our original submission (as already mentioned above).
> > >
> > > - _Reproducibility this is mostly math and proofs where in the appendix so it seems pretty reproducible.
> > >
> > > In our revised submission -- in order to increase reproducibility even further -- we have increased the level of detail to which proofs are presented even more. While this has added a good 20 pages to the appendix, we believe it to be worth the effort so that also researchers previously unfamiliar with the topic and utilized tools can follow the proofs with ease.
> > >
> > > We of course also share the code for our additional experiments in the updated supplementary material.
> > >
> > > - Summary Of The Review:
> > > I believe the paper could have been clearer and motivated the problem better. Similarly, the empirical analysis could have been more thorough.
> > >
> > > Following this valid comment we have – as discussed above – significantly increased the scope of our experimental section and provided a better motivation of the studied problems.
> > >
> > > - However, I believe the problem is important and the theoretical contributions is (AFAIK) novel and note-worthy. Therefore, I recommend to accept the paper.
> > >
> > > We wholeheartedly thank the reviewer for this sentiment.
> > >
> > >
> > > Reference:
> > >
> > > [1] https://arxiv.org/abs/1609.02907

---

### Author Response · Authors · 2022-11-18
**Response to all Reviewers**

As the rebuttal phase draws to a close, we would like to thank all reviewers for the time and energy invested into reviewing our paper!

We were encouraged to hear that the reviewers found our results on stability to structural perturbations to be interesting and/or relevant for the community (Reviewers QkKY, ekgB) and that the technical novelty and provided analysis in general was considered to be very solid (Reviewers  ekgB, T5hc, 4m2L).

 Since one of our aims in writing this paper was to provide a  rigorous stability framework that applied to a wide variety of settings – including for the first time also to networks on directed graphs -- we were very happy to read that the wide applicability of the results was appreciated (Reviewers QkKY, 4m2L, T5hc).

Following a common suggestion in the received recommendations, we have significantly increased the scope of our experimental section, which now includes additional experiments and also utilizes real world data, as detailed further in the individual responses below.

We also dilligently followed the received feedback on how to improve the presentation of our material (e.g. by shortening the Introduction, motivating certain concepts more and including a Notations section).

Reviewer QkKY raised the important question of applicability of our studied coarse-graining-example-setting (in which transferability between graphs with strongly connected subgraphs and versions where these subgraphs are replaced by single nodes are studied).
Following this question, we now make this point a lot clearer in our revised paper and list examples of application-settings (such as spatial graphs, social networks, molecular graphs, etc.)  in our revised manuscript.

We now also study one of these settings in greater detail on real world data in our revised experimental section.

Additionally, we now also discuss additional example-applications beyond the coarse-graining setting such as transferability between large graphs.

Below, we have provided detailed responses to received comments and answers to raised questions in the individual responses.

We look forward to the second stage of the discussion period and stand by to address any further concerns, should they arise.

---

### Decision · Program_Chairs · 2023-01-20

**Decision:**

Accept: poster

**Justification For Why Not Higher Score:**

This paper is a (noteworthy)  generalization of existing stability analyses of GCNs --- the analysis however might not necessarily be novel enough to warrant a spotlight.

**Justification For Why Not Lower Score:**

The paper brings substantial contributions to the analysis of the stability of GCNs. The results are novel enough to warrant publication.

**Metareview: Summary, Strengths And Weaknesses:**

This paper studies the problem of stability of Graph Convolutional Networks. In a Euclidean setting, stability refers to exploring the sensitivity of the output of an algorithm to small (trivial) changes in the input. In the graph setting, this sensitivity has to be investigated with respect to both perturbation of the node feature and the adjacency matrix (the latter is sometimes known as “transferability”). Existing analyses are limited (a) in the type of filters that they use and (b) in that they hold asymptotically and not in the finite sample regime. This paper aims to fill that go, and provide a series of results on the stability of GCN to signal, edge and adjacency matrix perturbations. More specifically, the authors use the comparison of GCN to graph filters to derive bounds that do not require the filters to be normal.

The feedback for this paper was good. Overall, the reviewers all agreed on the relevance of the paper, and on the significance of the contributions: the results are nice and generalize some existing analyses. Most of the initial concerns of the reviewers focused on the clarity of the paper --- more specifically, the density of the paper and the presentation of the results. In the initial presentation, the introduction was deemed too dense, and the bounds not appropriately commented upon. The authors were prompt to take this feedback into account and modified their exposition to shorten the introduction, increase the experimental result part and comment on their bounds. The rebuttal seems to have been successful in improving the clarity of the exposition.


**Note From Pc:**

if the above contains the word "oral" or "spotlight" please see: "oral" presentation means -> notable-top-5% and "spotlight" means -> notable-top-25%. As stated in our emails, we are disassociating presentation type from AC recommendations

**Summary Of Ac-Reviewer Meeting:**

Note: a meeting was originally scheduled to discuss the paper. However, after the rebuttal, the reviewers raised their scores, and, considering the positive feedback expressed in their comments, all agreed that meeting would be useless.